# Bit-swapping Oriented Twin-memory Multi-view Clustering in Lifelong Incomplete Scenarios

**Shengju Yu**[1], **Pei Zhang**[1], **Siwei Wang**[2]*, **Suyuan Liu**[1], **Xinhang Wan**[1],
**Zhibin Dong**[1], **Tiejun Li**[1]*, **Xinwang Liu**[1]*

[1]National University of Defense Technology, [2]Intelligent Game and Decision Lab
yu-shengju@foxmail.com, {wangsiwei13,tjli,xinwangliu}@nudt.edu.cn

## Abstract

Although receiving notable improvements, current multi-view clustering (MVC) techniques generally rely on feature library mechanisms to propagate accumulated knowledge from historical views to newly-arrived data, which overlooks the information pertaining to basis embedding within each view. Moreover, the mapping paradigm inevitably alters the values of learned landmarks and built affinities due to the uninterruption nature, accordingly disarraying the hierarchical cluster structures. To mitigate these two issues, we in the paper provide a named BSTM algorithm. Concretely, we firstly synchronize with the distinct dimensions by introducing a group of specialized projectors, and then establish unified anchors for all views collected so far to capture intrinsic patterns. Afterwards, departing from per-view architectures, we devise a shared bipartite graph construction via indicators to quantify similarity, which not only avoids redundant data-recalculations but alleviates the representation distortion caused by fusion. Crucially, there two components are optimized within an integrated framework, and collectively facilitate knowledge transfer upon encountering incoming views. Subsequently, to flexibly do transformation on anchors and meanwhile maintain numerical consistency, we develop a bit-swapping scheme operating exclusively on 0 and 1. It harmonizes anchors on current view and that on previous views through one-hot encoded row and column attributes, and the graph structures are correspondingly reordered to reach a matched configuration. Furthermore, a computationally efficient four-step updating strategy with linear complexity is designed to minimize the associated loss. Extensive experiments organized on publicly-available benchmark datasets with varying missing percentages confirm the superior effectiveness of our BSTM.

## 1 Introduction

As information technology advances, multi-view data, typically referring to the characterization of the same object derived from diverse domains, is becoming increasingly ubiquitous [35, 83, 55, 52, 42]. Naturally, extracting meaningful patterns from these data has sparked substantial research interest [12, 41, 73, 46, 78, 5]. Owing to the superior heterogeneous data integration capability, multi-view clustering (MVC) is generally recognized as an effective methodology for tackling multi-view data and is deployed in various fields like financial analysis, community recommendation, intelligent diagnosis, etc [37, 16, 60, 84, 51, 28]. It seeks to partition samples without requiring any labels into distinct groups such that the intra-group similarity is maximized while the inter-group dissimilarity is maintained, thereby uncovering potential data relationships [91, 47, 9, 19, 82, 80, 8]. To enhance the partition quality, recently researchers have developed numerous innovative methods [53, 58, 43, 92, 79, 85]. For sample, Chen *et al.* [2] learn hybridized representations to build a fused

---

*Corresponding Authors.

sparse similarity measure based on Euclidean rendering, and continuously calibrate a dynamic set with collaborative representation residuals. Li *et al.* [21] establish a common embedding space through sample transformation to refine the knowledge repository, and incorporate the geometric structure of streaming views into the consistency similarity matrix. Wen *et al.* [66] devise a confidence-based neighbor graph for incomplete views to regularize consensus graph learning, and leverage multiple connected sub-graphs to reveal group-wise cluster-aware topological patterns. Lin *et al.* [33] assimilate incomplete feature inference and self-representation recoupling to discover latent consistent cluster partition, and employ the tensor decomposition to explore high-order cross-view correlations.

Although achieving impressive enhancements in clustering effectiveness from multiple perspectives, most studies usually employ feature library mechanisms to disseminate formulated knowledge from prior observations to newly-arrived data, which fails to account the characteristics about basis embedding within each view, whether for historical views or new views. This is not conductive to deriving expressive and comprehensive representations as there generally exist valuable geometric features and essential cluster topologies in the basis embedding. Additionally, due to the inherent non-separable continuum attribute, the mapping paradigm currently adopted unavoidably distorts the learned landmarks and established affinities. Not only does this disrupt the anchor topology structures but disarrays the hierarchical cluster formations, accordingly limiting the clustering ability of models.

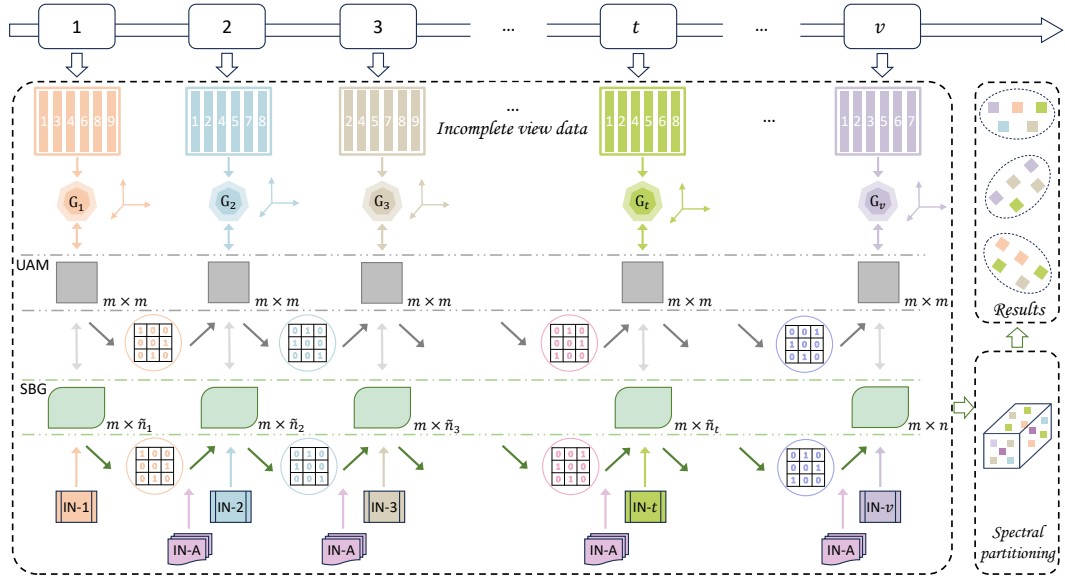

Figure 1: Pipeline of the designed BSTM. It firstly harmonizes diverse dimensions via projectors $\{\mathbf{G}_t\}$, and then builds unified anchor matrix (UAM) for all received views to extract latent patterns. Subsequently, it designs a shared bipartite graph (SBG) paradigm based on constructed indicator matrices IN-$t$ and IN-A to establish similarity. This avoids view-data recalculating and alleviates fusion distortion. Also, UAM and SBG collectively do knowledge transfer. Further, it devises a bit-swapping scheme consisting of 0 and 1 to rearrange anchors and meanwhile maintain numerical consistency. All components are jointly optimized through an integrated learning mode. IN-$t$: The indicator matrix built for the $t$-th view; IN-A: The indicator matrix built for all historical $t-1$ views.

In response to these two challenges, we in the paper develop a named BSTM algorithm, and Fig. 1 presents its overall pipeline. Specifically, we begin with harmonizing disparate dimensions through a set of specialized view-specific projectors, and then construct unified anchors over all arrived (incomplete) views to exploit underlying patterns. Subsequently, rather than conventional per-view architectures, we introduce a shared bipartite graph paradigm based on devised indicator matrices to measure similarity, which not only effectively eliminates view-data recomputing but also mitigates fusion-caused representation distortion. More significantly, both unified anchors and the shared bipartite graph possess memory functions, and collectively transfer knowledge when encountering incoming views. After that, in order to flexibly do transformation on anchors to alleviate mismatching with new views and at the same time preserve numerical consistency, we design a bit-swapping

strategy that operates exclusively on $\{0, 1\}$ space. It introduces a matrix with one-hot encoded row and column properties for all historical views to rearrange anchors according to the characteristics of current view. Correspondingly, the graph structures undergo topological reordering to reach a matched configuration while maintaining value invariance. It is noteworthy that all of these components are co-optimized in one common framework, enabling mutually reinforcing interactions. Thereafter, to minimize the associated loss, we give a computationally-efficient four-step solving scheme with linear complexity. The clustering results are acquired by conducting spectral partitioning on the produced bipartite graph. Finally, to adequately evaluate the effectiveness of presented BSTM, we carry out extensive contrast experiments with diverse missing data ratios on several public datasets. Briefly stated, the contributions of this paper mainly include

- We propose a twin-memory paradigm to facilitate knowledge transfer, which incorporates underlying basis-pattern representations and eliminates redundant view-data recalculating as well as mitigates fusion-induced distortion.
- We introduce a bit-swapping strategy to flexibly transform anchor topologies and graph structures while preserving numerical consistency.
- We optimize all components in an end-to-end manner, and provide a linear-complexity solver to ensure computational efficiency. Extensive experiments validate the effectiveness of our designed method.

## 2 Related Work

To boost clustering performance, MVC research community has developed a series of innovative algorithms from different perspectives [34, 75, 45]. For instance, Wan *et al.* [54] preserve a kernel-induced partition through late fusion for incremental knowledge integration, and substitute average partition with alignment maximization to consolidate prior patterns. On this basis, Yan *et al.* [77] introduce discrete category learning by constrained matrix factorization to progressively uncover latent clustering structures as the optimization progresses. Rather than kernel combinations, Wan *et al.* [56] employ an orthogonal coefficient matrix to integrate view data, and dynamically refine this matrix via an information propagation mechanism that synergistically couples established representations with streaming view observations. Qu *et al.* [49] utilize principal component analysis to standardize the dimension variations, and apply orthogonal permutation matrices to reorganize learned prototypes according to their underlying structural properties. Together with space correlation, Cai *et al.* [1] design view-wise codebooks and embedding codebooks to extract patterns based on spectral clustering paradigm, and aggregate multi-view information via an adaptive fusion mode. Sun *et al.* [50] leverage orthogonal repository and feature dictionary to identify potential cluster centers and uncover shared manifolds, and derive the encoding matrix by maximally transferring knowledge from them. In place of vanilla graph, Yin *et al.* [81] integrate sparse similarity learning with connectivity-preserving learning to simultaneously accomplish noise suppression and intra-cluster structure maintenance after reconstruction. Wang *et al.* [62] harness multi-level consensus representations to maintain heterogeneous distributions and inter-view relationships, and leverage a view-consistency matrix to reveal the similar underlying patterns among views. Under relation consistency learning, Yuan *et al.* [86] utilize a contrastive learning-driven view coherence maximization paradigm to compensate for data incompleteness, and mitigate noisy influence through discriminative prototype learning. Xu *et al.* [76] adopt adaptive feature rendering to bypass data imputation requirements, and extract cluster structures through joint maximization of mutual information and minimization of mean discrepancy.

## 3 Preliminaries

With the newly-arrived $t$-th (incomplete) view data $\mathbf{D}_t \in \mathbb{R}^{d_t \times n_t}$ where $d_t$ denotes the data dimension and $n_t$ denotes the number of observed samples, the basic clustering model can be mathematically described as

$$\min_{\mathbf{A}_p \in \mathbb{R}^{n_p \times n_p}, \mathbf{A} \in \mathbb{R}^{n \times n}} \sum_{p=1}^{t} \|\mathbf{D}_p - \mathbf{D}_p \mathbf{A}_p\|_F^2 + \lambda \Psi \left( \mathbf{C}_p \mathbf{A}_p \mathbf{C}_p^\top, \mathbf{A} \right)$$

$$\text{s.t. } \mathbf{A}_p \geq 0, \mathbf{A}_p^\top \mathbf{1}_{n_p} = \mathbf{1}_{n_p}, [\mathbf{A}_p]_{i,i} = 0, i = 1, \cdots, n_p,$$

$$\mathbf{A} \geq 0, \mathbf{A}^\top \mathbf{1}_n = \mathbf{1}_n, [\mathbf{A}]_{i,i} = 0, i = 1, \cdots, n. \quad (1)$$

$\mathbf{A}_p$ is the affinity generated on view $p$. $\mathbf{C}_p \in \mathbb{R}^{n \times n_p}$ is the built indicator matrix, and consists of $[\mathbf{C}_p]_{i,j} = 1$ if $[\mathbf{w}_p]_j = i$ else $[\mathbf{C}_p]_{i,j} = 0$, $j = 1, \cdots, n_p$ where $\mathbf{w}_p \in \mathbb{R}^{n_p}$ is the given index vector on view $p$. $n$ is the number of samples and greater than or equal to $n_p$. $\mathbf{A}$ is the generated fused affinity. The function $\Psi$ denotes certain fusion strategies, and the hyper-parameter $\lambda$ plays a role in balancing reconstruction error item and fusion item. After obtaining $\mathbf{A}$, the data grouping results can be acquired by conducting spectral clustering on it.

## 4 Methodology

Firstly, we associate a projector $\mathbf{G}_t \in \mathbb{R}^{d_t \times m}$ for new view data $\mathbf{D}_t$ to harmonize its dimension, and then introduce anchor matrix $\mathbf{B}_t \in \mathbb{R}^{m \times m}$ and bipartite graph $\mathbf{Q}_t \in \mathbb{R}^{m \times n_t}$ to exploit potential intrinsic structures where $m$ is the number of anchors. Accordingly, we formulate the loss as

$$\mathcal{L}_0 = \min_{\mathbf{G}_t, \mathbf{B}_t, \mathbf{Q}_t} \|\mathbf{D}_t - \mathbf{G}_t \mathbf{B}_t \mathbf{Q}_t\|_F^2$$
$$\text{s.t. } \mathbf{G}_t^\top \mathbf{G}_t = \mathbf{I}_m, \mathbf{B}_t \mathbf{B}_t^\top = \mathbf{I}_m, \mathbf{Q}_t \geq 0, \mathbf{Q}_t^\top \mathbf{1}_m = \mathbf{1}_{n_t}. \tag{2}$$

The orthogonality aims at enhancing the discrimination, while the non-negativity and column-sum constraints guarantee it satisfies similarity requirements.

Subsequently, rather than recomputing all historical views, we utilize anchor matrix and bipartite graph to collaboratively extract features **across** views. However, due to the incompleteness, the graph size is not compatible for different views. To get out of this dilemma, we establish a tailored indicator matrix $\mathbf{H}_t \in \mathbb{R}^{\widetilde{n}_t \times n_t}$ for current view $t$ where $\widetilde{n}_t$ is the number of samples observed on all $t$ views. Specially, we construct $\widetilde{\mathbf{w}}_t = \mathbf{w}_t \cup \widetilde{\mathbf{w}}_{t-1}$. $\mathbf{w}_t \in \mathbb{R}^{n_t}$ is the given index vector on view $t$ while $\widetilde{\mathbf{w}}_{t-1} \in \mathbb{R}^{\widetilde{n}_{t-1}}$ is the constructed index vector union on all previous $t-1$ views. At initial stage, $\widetilde{\mathbf{w}}_1 = \mathbf{w}_1$. Then, we define $[\mathbf{H}_t]_{i,j}$ as 1 if $[\mathbf{w}_t]_j = [\widetilde{\mathbf{w}}_t]_i$ else as 0; $i = 1, 2, \cdots, \widetilde{n}_t; j = 1, 2, \cdots, n_t$. On this basis, we devise the loss as

$$\mathcal{L}_1 = \min_{\mathbf{G}_t, \widetilde{\mathbf{B}}_t, \widetilde{\mathbf{Q}}_t} \|\mathbf{D}_t - \mathbf{G}_t \widetilde{\mathbf{B}}_t \widetilde{\mathbf{Q}}_t \mathbf{H}_t\|_F^2$$
$$\text{s.t. } \mathbf{G}_t^\top \mathbf{G}_t = \mathbf{I}_m, \widetilde{\mathbf{B}}_t \widetilde{\mathbf{B}}_t^\top = \mathbf{I}_m, \widetilde{\mathbf{Q}}_t \geq 0, \widetilde{\mathbf{Q}}_t^\top \mathbf{1}_m = \mathbf{1}_{\widetilde{n}_t}, \tag{3}$$

where $\widetilde{\mathbf{B}}_t \in \mathbb{R}^{m \times m}$ and $\widetilde{\mathbf{Q}}_t \in \mathbb{R}^{m \times \widetilde{n}_t}$ are the unified anchor matrix and shared bipartite graph respectively, and collectively transfer knowledge from historical $t-1$ views to new view $t$, avoiding view-data recalculating and meanwhile eliminating fusion-caused distortion. In particular, $\widetilde{\mathbf{Q}}_t \mathbf{H}_t \in \mathbb{R}^{m \times n_t}$ can be regarded as the graph measure for view $t$.

Afterwards, due to the unsupervised nature of clustering tasks and the potential heterogeneity in data distributions on different views, anchors on the current $t$-th view could mismatch with that generated on previous $t-1$ views. To remap anchors while maintaining their values, we introduce a bit-swapping transformation which operates exclusively on binary space. If anchors are matched, we optimize the corresponding element as 1 otherwise as 0. Especially, we associate the anchor matrix $\widetilde{\mathbf{B}}_{t-1}$ produced on previous $t-1$ views with a learnable $\widehat{\mathbf{S}}_{t-1} \in \mathbb{R}^{m \times m}$, which is devised to exactly contain a single 1 element in each row and column, to transform anchors. For the anchor transfer branch, consequently, we mathematically define the loss as

$$\mathcal{L}_2 = \min_{\widetilde{\mathbf{B}}_t, \widehat{\mathbf{S}}_{t-1}} \|\widetilde{\mathbf{B}}_t - \widetilde{\mathbf{B}}_{t-1} \widehat{\mathbf{S}}_{t-1}\|_F^2$$
$$\text{s.t. } \widetilde{\mathbf{B}}_t \widetilde{\mathbf{B}}_t^\top = \mathbf{I}_m, \widehat{\mathbf{S}}_{t-1} \in \{0, 1\}, \widehat{\mathbf{S}}_{t-1}^\top \mathbf{1}_m = \mathbf{1}_m, \widehat{\mathbf{S}}_{t-1} \mathbf{1}_m = \mathbf{1}_m. \tag{4}$$

Further, considering that variations in the anchor order will alter the graph structure, in order to make the graph keep pace with anchors, we employ $\widehat{\mathbf{S}}_{t-1}$ to reorganize the graph $\widetilde{\mathbf{Q}}_{t-1} \in \mathbb{R}^{m \times \widetilde{n}_{t-1}}$ that is shared for all previous $t-1$ views. In conjunction with the fact that the rows of $\widetilde{\mathbf{Q}}_{t-1}$ correspond to anchors, accordingly, we can adjust the graph by $\widehat{\mathbf{S}}_{t-1}^\top \widetilde{\mathbf{Q}}_{t-1}$. Next, we need to build the relation between $\widetilde{\mathbf{Q}}_t$ and $\widehat{\mathbf{S}}_{t-1}^\top \widetilde{\mathbf{Q}}_{t-1}$. Due to the presence of missing samples, the sizes of these two parts may encounter incompatibility. To this end, we customize an indicator matrix $\widetilde{\mathbf{E}}_{t-1} \in \mathbb{R}^{\widetilde{n}_t \times \widetilde{n}_{t-1}}$. Specially, the element $[\widetilde{\mathbf{E}}_{t-1}]_{i,j}$ is devised as 1 if $[\widetilde{\mathbf{w}}_t]_i = [\widetilde{\mathbf{w}}_{t-1}]_j$ else as 0.

Then, $\widetilde{\mathbf{Q}}_t \widetilde{\mathbf{E}}_{t-1} \in \mathbb{R}^{m \times \widetilde{n}_{t-1}}$ can be seen as the parts corresponding to the graph on all previous $t-1$ views. Therefore, for the graph transfer branch, we design the loss as

$$
\mathcal{L}_3 = \min_{\widetilde{\mathbf{Q}}_t, \widehat{\mathbf{S}}_{t-1}} \|\widetilde{\mathbf{Q}}_t \widetilde{\mathbf{E}}_{t-1} - \widehat{\mathbf{S}}_{t-1}^\top \widetilde{\mathbf{Q}}_{t-1}\|_F^2
$$
$$
\text{s.t. } \widetilde{\mathbf{Q}}_t \geq 0, \widetilde{\mathbf{Q}}_t^\top \mathbf{1}_m = \mathbf{1}_{\widetilde{n}_t}, \widehat{\mathbf{S}}_{t-1} \in \{0,1\}, \widehat{\mathbf{S}}_{t-1}^\top \mathbf{1}_m = \mathbf{1}_m, \widehat{\mathbf{S}}_{t-1} \mathbf{1}_m = \mathbf{1}_m. \tag{5}
$$

Based on the above exploration, we formulate the final loss as

$$
\mathcal{L} = \mathcal{L}_1 + \lambda \mathcal{L}_2 + \beta \mathcal{L}_3. \tag{6}
$$

## 5 Alternating Optimization

We adopt the idea of alternating optimization to minimize (6).

**Step-1  Updating $\mathbf{G}_t$.** With $\widetilde{\mathbf{B}}_t$, $\widetilde{\mathbf{Q}}_t$ and $\widehat{\mathbf{S}}_{t-1}$ held constant, the objective (6) becomes

$$
\min_{\mathbf{G}_t^\top \mathbf{G}_t = \mathbf{I}_m} \|\mathbf{D}_t - \mathbf{G}_t \widetilde{\mathbf{B}}_t \widetilde{\mathbf{Q}}_t \mathbf{H}_t\|_F^2. \tag{7}
$$

Expanding $F$-norm and deleting irrelevant items, we have the following equivalent problem,

$$
\max_{\mathbf{G}_t^\top \mathbf{G}_t = \mathbf{I}_m} \text{Tr}\left(\mathbf{G}_t^\top \mathbf{D}_t \mathbf{H}_t^\top \widetilde{\mathbf{Q}}_t^\top \widetilde{\mathbf{B}}_t^\top\right). \tag{8}
$$

The optimal solution is $\mathbf{U}\mathbf{V}^\top$, both of which are the singular matrices of $\mathbf{D}_t \mathbf{H}_t^\top \widetilde{\mathbf{Q}}_t^\top \widetilde{\mathbf{B}}_t^\top$.

**Remark 1.** *Due to $\mathbf{D}_t \in \mathbb{R}^{d_t \times n_t}$ and $\mathbf{H}_t \in \mathbb{R}^{\widetilde{n}_t \times n_t}$, directly calculating $\mathbf{D}_t \mathbf{H}_t^\top \widetilde{\mathbf{Q}}_t^\top \widetilde{\mathbf{B}}_t^\top$ will need at least $\mathcal{O}(d_t n_t \widetilde{n}_t)$ computational cost. It is square with respect to the number of samples, and consequently unsuitable for large-scale applications. Instead, we construct a zero matrix with the size of $d_t \times \widetilde{n}_t$, and then assign its columns using the column elements of $\mathbf{D}_t$ via corresponding indexes observed on the t-th view. Accordingly, calculating $\mathbf{D}_t \mathbf{H}_t^\top \widetilde{\mathbf{Q}}_t^\top \widetilde{\mathbf{B}}_t^\top$ needs $\mathcal{O}(d_t \widetilde{n}_t m + d_t m^2)$ cost. Since the data dimension $d_t$ is irrelevant to $n_t$ and $m$ is greatly smaller than the number of samples, the computational overhead about updating $\mathbf{G}_t$ will be $\mathcal{O}(n)$, i.e., linear to the number of samples.*

**Step-2  Updating $\widetilde{\mathbf{B}}_t$.** With $\mathbf{G}_t$, $\widetilde{\mathbf{Q}}_t$ and $\widehat{\mathbf{S}}_{t-1}$ held constant, the objective (6) becomes

$$
\min_{\widetilde{\mathbf{B}}_t \widetilde{\mathbf{B}}_t^\top = \mathbf{I}_m} \|\mathbf{D}_t - \mathbf{G}_t \widetilde{\mathbf{B}}_t \widetilde{\mathbf{Q}}_t \mathbf{H}_t\|_F^2 + \lambda \|\widetilde{\mathbf{B}}_t - \widetilde{\mathbf{B}}_{t-1} \widehat{\mathbf{S}}_{t-1}\|_F^2. \tag{9}
$$

After irrelevant item removing, it is equivalently simplified as

$$
\max_{\widetilde{\mathbf{B}}_t \widetilde{\mathbf{B}}_t^\top = \mathbf{I}_m} \text{Tr}\left(\left(\mathbf{G}_t^\top \mathbf{D}_t \mathbf{H}_t^\top \widetilde{\mathbf{Q}}_t^\top + \lambda \widetilde{\mathbf{B}}_{t-1} \widehat{\mathbf{S}}_{t-1}\right) \widetilde{\mathbf{B}}_t^\top\right). \tag{10}
$$

Accordingly, the optimal solution of $\widetilde{\mathbf{B}}_t$ is $\mathbf{U}\mathbf{V}^\top$, where the matrices $\mathbf{U}$ and $\mathbf{V}$ are the SVD outcomes of $\mathbf{G}_t^\top \mathbf{D}_t \mathbf{H}_t^\top \widetilde{\mathbf{Q}}_t^\top + \lambda \widetilde{\mathbf{B}}_{t-1} \widehat{\mathbf{S}}_{t-1}$.

**Remark 2.** *Rather than directly constructing the term $\mathbf{G}_t^\top \mathbf{D}_t \mathbf{H}_t^\top \widetilde{\mathbf{Q}}_t^\top + \lambda \widetilde{\mathbf{B}}_{t-1} \widehat{\mathbf{S}}_{t-1}$ which needs at least $\mathcal{O}(m n_t \widetilde{n}_t)$ computational cost, note that $\mathbf{D}_t$ and $\mathbf{H}_t$ remain unchanged throughout the entire optimization procedure, we utilize the established $\mathbf{D}_t \mathbf{H}_t^\top$ in updating $\mathbf{G}_t$ to construct this term, which only takes $\mathcal{O}(m d_t \widetilde{n}_t + m^2 \widetilde{n}_t + m^3)$ cost. Therefore, the computational overhead about $\widetilde{\mathbf{B}}_t$ is $\mathcal{O}(n)$.*

**Step-3  Updating $\widetilde{\mathbf{Q}}_t$.** With $\mathbf{G}_t$, $\widetilde{\mathbf{B}}_t$ and $\widehat{\mathbf{S}}_{t-1}$ held constant, the objective (6) becomes

$$
\min_{\widetilde{\mathbf{Q}}_t \geq 0, \widetilde{\mathbf{Q}}_t^\top \mathbf{1}_m = \mathbf{1}_{\widetilde{n}_t}} \|\mathbf{D}_t - \mathbf{G}_t \widetilde{\mathbf{B}}_t \widetilde{\mathbf{Q}}_t \mathbf{H}_t\|_F^2 + \beta \|\widetilde{\mathbf{Q}}_t \widetilde{\mathbf{E}}_{t-1} - \widehat{\mathbf{S}}_{t-1}^\top \widetilde{\mathbf{Q}}_{t-1}\|_F^2. \tag{11}
$$

After expanding, it can be further simplified as

$$
\min_{\widetilde{\mathbf{Q}}_t \geq 0, \widetilde{\mathbf{Q}}_t^\top \mathbf{1}_m = \mathbf{1}_{\widetilde{n}_t}} \text{Tr}\left(\widetilde{\mathbf{Q}}_t^\top \widetilde{\mathbf{B}}_t^\top \mathbf{G}_t^\top \mathbf{G}_t \widetilde{\mathbf{B}}_t \widetilde{\mathbf{Q}}_t \mathbf{H}_t \mathbf{H}_t^\top + \beta \widetilde{\mathbf{Q}}_t^\top \widetilde{\mathbf{Q}}_t \widetilde{\mathbf{E}}_{t-1} \widetilde{\mathbf{E}}_{t-1}^\top\right.
$$
$$
\left. - 2\left(\widetilde{\mathbf{Q}}_t^\top \widetilde{\mathbf{B}}_t^\top \mathbf{G}_t^\top \mathbf{D}_t \mathbf{H}_t^\top + \beta \widetilde{\mathbf{Q}}_t^\top \widehat{\mathbf{S}}_{t-1}^\top \widetilde{\mathbf{Q}}_{t-1} \widetilde{\mathbf{E}}_{t-1}^\top\right)\right). \tag{12}
$$

**Remark 3.** *Due to $\mathbf{H}_t\mathbf{H}_t^\top \in \mathbb{R}^{\widetilde{n}_t \times \widetilde{n}_t}$ and $\widetilde{\mathbf{E}}_{t-1} \in \mathbb{R}^{\widetilde{n}_t \times \widetilde{n}_{t-1}}$, the matrix products $\widetilde{\mathbf{Q}}_t\mathbf{H}_t\mathbf{H}_t^\top$ and $\widetilde{\mathbf{E}}_{t-1}\widetilde{\mathbf{E}}_{t-1}^\top$ will incur $\mathcal{O}(m\widetilde{n}_t^2)$ and $\mathcal{O}(\widetilde{n}_t^2\widetilde{n}_{t-1})$ computational complexities, which are square and cubic with respect to the number of samples respectively. To decrease them, observed that $\mathbf{H}_t$ consists of $\{0,1\}$ and each column contains only one 1 as well as the columns are orthogonal, we have that $\mathbf{H}_t\mathbf{H}_t^\top$ is a diagonal matrix with elements as 0 and 1. Therefore, we have $\widetilde{\mathbf{Q}}_t\mathbf{H}_t\mathbf{H}_t^\top = \widetilde{\mathbf{Q}}_t \odot \mathbf{J}_t$ where $\mathbf{J}_t = \mathbf{1}_m \cdot [\sum_{j=1}^{n_t}[\mathbf{H}_t]_{1,j}, \sum_{j=1}^{n_t}[\mathbf{H}_t]_{2,j}, \cdots, \sum_{j=1}^{n_t}[\mathbf{H}_t]_{\widetilde{n}_t,j}]$. Similarly for $\widetilde{\mathbf{E}}_{t-1}$, we have $\widetilde{\mathbf{Q}}_t\widetilde{\mathbf{E}}_{t-1}\widetilde{\mathbf{E}}_{t-1}^\top$ is equal to $\widetilde{\mathbf{Q}}_t \odot \mathbf{M}_t$ where $\mathbf{M}_t = \mathbf{1}_m \cdot [\sum_{j=1}^{\widetilde{n}_{t-1}}[\widetilde{\mathbf{E}}_{t-1}]_{1,j}, \sum_{j=1}^{\widetilde{n}_{t-1}}[\widetilde{\mathbf{E}}_{t-1}]_{2,j}, \cdots, \sum_{j=1}^{\widetilde{n}_{t-1}}[\widetilde{\mathbf{E}}_{t-1}]_{\widetilde{n}_t,j}]$. Then, combined with $\widetilde{\mathbf{Q}}_t \in \mathbb{R}^{m \times \widetilde{n}_t}$, after this equivalent transformation, both of computational complexities are decreased to $\mathcal{O}(n)$.*

Then, combining the constraints about $\widetilde{\mathbf{Q}}_t$, $\widetilde{\mathbf{B}}_t$ and $\mathbf{G}_t$, we can update $\widetilde{\mathbf{Q}}_t$ on a column-by-column basis. Accordingly, we can deduce

$$\min_{[\widetilde{\mathbf{Q}}_t]_{:,j} \geq 0, [\widetilde{\mathbf{Q}}_t]_{:,j}^\top \mathbf{1}_m = 1} \left[\widetilde{\mathbf{Q}}_t\right]_{:,j}^\top \left[\widetilde{\mathbf{Q}}_t\right]_{:,j} - 2\left[\widetilde{\mathbf{Q}}_t\right]_{:,j}^\top \frac{\left[\widetilde{\mathbf{B}}_t^\top\mathbf{G}_t^\top\mathbf{D}_t\mathbf{H}_t^\top\right]_{:,j} + \beta\left[\widehat{\mathbf{S}}_{t-1}^\top\widetilde{\mathbf{Q}}_{t-1}\widetilde{\mathbf{E}}_{t-1}^\top\right]_{:,j}}{\sum_{l=1}^{n_t}[\mathbf{H}_t]_{j,l} + \beta\sum_{l=1}^{\widetilde{n}_{t-1}}\left[\widetilde{\mathbf{E}}_{t-1}\right]_{j,l}}.$$

(13)

**Remark 4.** *Note that $\mathbf{H}_t$ measures whether the samples are available on the t-th view and $\widetilde{\mathbf{E}}_{t-1}$ measures whether the samples are on all previous $t-1$ views. Consequently, we have $\sum_{l=1}^{n_t}[\mathbf{H}_t]_{j,l} + \beta\sum_{l=1}^{\widetilde{n}_{t-1}}[\widetilde{\mathbf{E}}_{t-1}]_{j,l}$ means that the sample $j$ must be in $t$ views. Thus, this coefficient must be non-zero.*

Further, we equivalently have

$$\min_{[\widetilde{\mathbf{Q}}_t]_{:,j} \geq 0, [\widetilde{\mathbf{Q}}_t]_{:,j}^\top \mathbf{1}_m = 1} \left\|\left[\widetilde{\mathbf{Q}}_t\right]_{:,j} - \mathbf{T}_{:,j}\right\|_F^2.$$

(14)

where $\mathbf{T}_{:,j} = \left(\left[\widetilde{\mathbf{B}}_t^\top\mathbf{G}_t^\top\mathbf{D}_t\mathbf{H}_t^\top + \beta\widehat{\mathbf{S}}_{t-1}^\top\widetilde{\mathbf{Q}}_{t-1}\widetilde{\mathbf{E}}_{t-1}^\top\right]_{:,j}\right) / \left(\sum_{l=1}^{n_t}[\mathbf{H}_t]_{j,l} + \beta\sum_{l=1}^{\widetilde{n}_{t-1}}\left[\widetilde{\mathbf{E}}_{t-1}\right]_{j,l}\right)$.

It has the following closed-form solution,

$$\left[\widetilde{\mathbf{Q}}_t\right]_{:,j} = \left(\mathbf{T}_{:,j} + \frac{\mathbf{1}_m - \mathbf{1}_m^\top\mathbf{T}_{:,j}\mathbf{1}_m}{m}\right)_+, \quad j = 1, 2, \cdots, \widetilde{n}_t.$$

(15)

**Remark 5.** *The computing overhead is mainly from the construction of $\mathbf{T}$, especially from the numerator term. Constructing the terms $\widetilde{\mathbf{B}}_t^\top\mathbf{G}_t^\top\mathbf{D}_t\mathbf{H}_t^\top$ and $\widehat{\mathbf{S}}_{t-1}^\top\widetilde{\mathbf{Q}}_{t-1}\widetilde{\mathbf{E}}_{t-1}^\top$ needs at least $\mathcal{O}(m^2d_t + md_tn_t + mn_t\widetilde{n}_t)$ and $\mathcal{O}(m^2\widetilde{n}_{t-1} + m\widetilde{n}_t\widetilde{n}_{t-1})$ cost respectively, both of which are square with respect to the number of samples. To reduce them, we utilize the $\mathbf{D}_t\mathbf{H}_t^\top$ built in the updating of $\mathbf{G}_t$ to construct $\widetilde{\mathbf{B}}_t^\top\mathbf{G}_t^\top\mathbf{D}_t\mathbf{H}_t^\top$, which takes $\mathcal{O}(md_t\widetilde{n}_t)$. For $\widehat{\mathbf{S}}_{t-1}^\top\widetilde{\mathbf{Q}}_{t-1}\widetilde{\mathbf{E}}_{t-1}^\top$, inspired by the $\{0,1\}$ characteristic and column orthogonality of $\widetilde{\mathbf{E}}_{t-1}$, we have that $\widetilde{\mathbf{Q}}_{t-1}\widetilde{\mathbf{E}}_{t-1}^\top$ means taking the columns of $\widetilde{\mathbf{Q}}_{t-1}$ to generate a matrix with the same size as $\widetilde{\mathbf{Q}}_t$. Therefore, we can establish $\widetilde{\mathbf{Q}}_{t-1}\widetilde{\mathbf{E}}_{t-1}^\top$ by first creating a zero matrix with the size of $m \times \widetilde{n}_t$ and then assigning its columns using the columns of $\widetilde{\mathbf{Q}}_{t-1}$ under corresponding indexes. Accordingly, constructing $\widehat{\mathbf{S}}_{t-1}^\top\widetilde{\mathbf{Q}}_{t-1}\widetilde{\mathbf{E}}_{t-1}^\top$ takes $\mathcal{O}(m^2\widetilde{n}_t)$.*

**Step-4** **Updating $\widehat{\mathbf{S}}_{t-1}$.** With $\mathbf{G}_t$, $\widetilde{\mathbf{B}}_t$ and $\widetilde{\mathbf{Q}}_t$ held constant, the objective (6) becomes

$$\min_{\widehat{\mathbf{S}}_{t-1} \in \{0,1\}, \widehat{\mathbf{S}}_{t-1}^\top\mathbf{1}_m = \mathbf{1}_m, \widehat{\mathbf{S}}_{t-1}\mathbf{1}_m = \mathbf{1}_m} \lambda\|\widetilde{\mathbf{B}}_t - \widetilde{\mathbf{B}}_{t-1}\widehat{\mathbf{S}}_{t-1}\|_F^2 + \beta\|\widetilde{\mathbf{Q}}_t\widetilde{\mathbf{E}}_{t-1} - \widehat{\mathbf{S}}_{t-1}^\top\widetilde{\mathbf{Q}}_{t-1}\|_F^2.$$

(16)

It can be further equivalently transformed as

$$\max_{\widehat{\mathbf{S}}_{t-1} \in \{0,1\}, \widehat{\mathbf{S}}_{t-1}^\top\mathbf{1}_m = \mathbf{1}_m, \widehat{\mathbf{S}}_{t-1}\mathbf{1}_m = \mathbf{1}_m} \mathrm{Tr}\left(\left(\lambda\widetilde{\mathbf{B}}_t^\top\widetilde{\mathbf{B}}_{t-1} + \beta\widetilde{\mathbf{Q}}_t\widetilde{\mathbf{E}}_{t-1}\widetilde{\mathbf{Q}}_{t-1}^\top\right)\widehat{\mathbf{S}}_{t-1}\right).$$

(17)

Via the matrix vectorization transformation $\mathrm{Vec}(\cdot)$, we have

$$\max_{\widehat{\mathbf{S}}_{t-1} \in \{0,1\}, \widehat{\mathbf{S}}_{t-1}^\top\mathbf{1}_m = \mathbf{1}_m, \widehat{\mathbf{S}}_{t-1}\mathbf{1}_m = \mathbf{1}_m} \left(\mathrm{Vec}\left(\left(\lambda\widetilde{\mathbf{B}}_t^\top\widetilde{\mathbf{B}}_{t-1} + \beta\widetilde{\mathbf{Q}}_t\widetilde{\mathbf{E}}_{t-1}\widetilde{\mathbf{Q}}_{t-1}^\top\right)^\top\right)\right)^\top \mathrm{Vec}\left(\widehat{\mathbf{S}}_{t-1}\right),$$

(18)

which is an integer linear programming problem and can be solved by existing software. Since $\widehat{\mathbf{S}}_{t-1}$ is with the size of $m \times m$ and $m$ is largely smaller than the number of samples, solving this problem requires minor computing cost.

**Remark 6.** *The computing cost about $\widehat{\mathbf{S}}_{t-1}$ is mainly from constructing $\lambda \widetilde{\mathbf{B}}_t^\top \widetilde{\mathbf{B}}_{t-1} + \beta \widetilde{\mathbf{Q}}_t \widetilde{\mathbf{E}}_{t-1} \widetilde{\mathbf{Q}}_{t-1}^\top$, which takes $\mathcal{O}(m^3 + m\widetilde{n}_t\widetilde{n}_{t-1} + m^2\widetilde{n}_{t-1})$. To reduce it to linear, noticed that there is only one 1 in each column of $\widetilde{\mathbf{E}}_{t-1}$ and other elements are all 0, we have that $\widetilde{\mathbf{Q}}_t\widetilde{\mathbf{E}}_{t-1}$ indicates picking out certain columns of $\widetilde{\mathbf{Q}}_t$ to build a smaller matrix. Thus, we establish an auxiliary matrix, and utilize the columns of $\widetilde{\mathbf{Q}}_t$ to do element assignment. Accordingly, the computing cost is reduced to $\mathcal{O}(m^2\widetilde{n}_{t-1})$.*

**Steps Under Initial View**. When facing only one view, the loss (6) degenerates into (3), which can be deemed as the special case of (6) with $\lambda$ and $\beta$ taking 0 and not involving $\widehat{\mathbf{S}}_{t-1}$. Therefore, $\mathbf{G}_t$ and $\widetilde{\mathbf{B}}_t$ can be updated via (8) and (10). Additionally, $\mathbf{H}_t$ (i.e., $\mathbf{H}_1$) is an identity matrix. Correspondingly, the term $\sum_{l=1}^{n_t} [\mathbf{H}_t]_{j,l} + \beta \sum_{l=1}^{\widetilde{n}_{t-1}} [\widetilde{\mathbf{E}}_{t-1}]_{j,l}$ in (13) is non-zero. Thus, $\widetilde{\mathbf{Q}}_t$ can still be updated via (15).

**Algorithm** 1 gives the overall procedure of BSTM where $f_{loss}(l)$ is the loss value at the $l$-th iteration.

**Remark 7.** *The overall computational complexity of the proposed BSTM is linear with respect to the number of samples, which accordingly renders it well-suited for large-scale implementations.*

**Remark 8.** *Updating $\mathbf{G}_t$, $\widetilde{\mathbf{B}}_t$, $\widetilde{\mathbf{Q}}_t$ and $\widehat{\mathbf{S}}_{t-1}$ takes $\mathcal{O}(d_t\widetilde{n}_t + d_t m)$, $\mathcal{O}(d_t\widetilde{n}_t + m\widetilde{n}_t + m^2)$, $\mathcal{O}(d_t\widetilde{n}_t + md_t + m\widetilde{n}_t)$, $\mathcal{O}(m^2 + m\widetilde{n}_{t-1})$ memory cost respectively. So, BSTM's space complexity is also linear.*

---

**Algorithm 1** Bit-swapping Oriented Twin-memory Lifelong Incomplete MVC (BSTM)

---

**Input**: Incomplete multi-view data $\mathbf{D}_t$ with index vector $\mathbf{w}_t$, hyper-parameters $\lambda$ and $\beta$;
**Initialize**: $\mathbf{G}_1, \widetilde{\mathbf{B}}_1, \widetilde{\mathbf{Q}}_1, \widehat{\mathbf{S}}_1$;
**Output**: Performing spectral partitioning on $\widetilde{\mathbf{Q}}_t$ to produce clustering results;

1: **for** $t = 1$ to $v$ **do**
2:    **if** $t == 1$ **then**
3:       Setting $\lambda$ and $\beta$ as 0.
4:    **end if**
5:    $l = 0$;
6:    **repeat**
7:       $l = l + 1$;
8:       Updating the guidance variable $\mathbf{G}_t$ via (8);
9:       Updating the unified anchor variable $\widetilde{\mathbf{B}}_t$ via (10);
10:      Updating the shared bipartite graph $\widetilde{\mathbf{Q}}_t$ via (15);
11:     **if** $t > 1$ **then**
12:       Updating the swapping variable $\widehat{\mathbf{S}}_{t-1}$ via (18);
13:     **end if**
14:    **until** $l > 1$ and $\frac{f_{loss}(l-1) - f_{loss}(l)}{f_{loss}(l-1)} \leq \epsilon$;
15: **end for**

---

## 6 Experiments

Table 1: Details of the Datasets Utilized in Experiments

| Dataset | NS | DD | NC | NV | Dataset | NS | DD | NC | NV |
|---|---|---|---|---|---|---|---|---|---|
| PROKARYO | 551 | 438/3/393 | 4 | 3 | PROTEINF | 694 | 27/27/.../27/27 | 27 | 12 |
| WIKIFEA | 2866 | 128/10 | 10 | 2 | NUSWIENE | 4095 | 128/73/144/225/64 | 33 | 5 |
| CALTEALL | 9144 | 48/40/254/512/928 | 102 | 5 | YOUTUTEN | 38654 | 944/576/512/640 | 10 | 4 |
| YOUTUTWE | 63896 | 944/576/512/640 | 20 | 4 | FAMNISIX | 70000 | 5376/512/5376/5376/1239/5376 | 10 | 6 |

We conduct experiments on 8 public datasets, with their key characteristics summarized in Table 1. DD is the data dimension. NS, NC and NV are the numbers of samples, clusters and views, respectively.

12 classical methods, GSRMC [26], FLSD [70], MVTSC [71], MVCBG [65], AGCMC [69], NGSPL [72], PIMVC [7], TCIMC [74], TMBSD [29], PSMVC [24], MKKMC [40] and LRGMV [6] are chosen as the baselines. Their descriptions are presented in Section F due to the space limit.

## 6.1 Results and Interpretations

Table 2: Comparative Evaluation of Clustering Method Performance

| Dataset | PROKARYO | | | | | | | | | PROTEINF | | | | | | | | |
|---|---|---|---|---|---|---|---|---|---|---|---|---|---|---|---|---|---|---|
| MR | 0.1 | | | 0.4 | | | 0.7 | | | 0.1 | | | 0.4 | | | 0.7 | | |
| Method | ACC | PUR | FSC | ACC | PUR | FSC | ACC | PUR | FSC | ACC | PUR | FSC | ACC | PUR | FSC | ACC | PUR | FSC |
| GSRMC | 27.37 | 56.81 | 30.42 | 39.79 | 57.03 | 35.97 | 45.59 | 64.24 | 40.48 | 32.02 | 36.19 | 17.39 | 31.34 | **38.70** | 16.95 | 29.53 | 34.03 | 12.13 |
| FLSD | 57.74 | 64.75 | 57.67 | 56.46 | 61.16 | 55.87 | **53.99** | 61.77 | **51.56** | 27.06 | 34.66 | 15.90 | 25.46 | 32.62 | 14.24 | 24.57 | 31.28 | 13.32 |
| MVTSC | 49.76 | 61.52 | 48.78 | 50.72 | 59.51 | 48.68 | 48.52 | 59.56 | 46.60 | 30.23 | 36.99 | 17.45 | 28.75 | 34.65 | 16.15 | 25.86 | 31.44 | 14.33 |
| MVCBG | 55.54 | 67.70 | 44.55 | 56.87 | 68.60 | 45.37 | 53.18 | 66.26 | 44.36 | 29.16 | 34.97 | 16.63 | 28.34 | 33.85 | 15.92 | 26.26 | 31.76 | 13.95 |
| AGCMC | 70.13 | 77.19 | 62.51 | 65.65 | 69.95 | 56.43 | 52.30 | 61.02 | 41.14 | 20.93 | 24.88 | 10.89 | 20.96 | 25.61 | 11.06 | 19.17 | 22.69 | 10.35 |
| NGSPL | 48.87 | 59.33 | 47.58 | 46.10 | 57.70 | 43.89 | 45.83 | 57.32 | 44.94 | 25.48 | 31.43 | 12.82 | 22.71 | 27.38 | 11.14 | 22.30 | 27.48 | 10.73 |
| PIMVC | 46.87 | 66.10 | 44.24 | 45.06 | 65.70 | 44.08 | 40.69 | 59.85 | 42.02 | 31.30 | 39.59 | 18.82 | 32.23 | 37.37 | 17.32 | 30.01 | **36.88** | 13.54 |
| TCIMC | 30.27 | 58.81 | 35.64 | 30.23 | 57.96 | 33.87 | 29.32 | 55.43 | 32.43 | 14.36 | 17.15 | 8.88 | 14.31 | 16.42 | 9.45 | 14.63 | 17.23 | 9.43 |
| TMBSD | 51.54 | 42.42 | 39.66 | 47.46 | 39.08 | 32.04 | 40.26 | 33.03 | 14.91 | 20.58 | 18.02 | 20.42 | 21.96 | 19.32 | 16.14 | 25.30 | 22.15 | 14.11 |
| PSMVC | 53.18 | 65.70 | 44.03 | 53.36 | 66.79 | 43.05 | 51.11 | 63.70 | 40.34 | 28.70 | 34.09 | 14.72 | 28.34 | 32.94 | 14.59 | 26.12 | 30.91 | 13.25 |
| MKKMC | 50.20 | 42.27 | 38.79 | 39.55 | 33.57 | 20.72 | 37.63 | 32.14 | 12.56 | 14.36 | 17.15 | 8.88 | 14.31 | 16.42 | 9.45 | 14.63 | 17.23 | 9.43 |
| LRGMV | 30.22 | 58.85 | 32.60 | 28.71 | 56.63 | 31.31 | 31.25 | 56.82 | 32.97 | 34.15 | 41.28 | 21.30 | 32.65 | 37.78 | 17.87 | 29.88 | 35.59 | 12.69 |
| Ours | **74.95** | **82.21** | **64.79** | **72.78** | **79.13** | **62.82** | 52.09 | **74.59** | 49.59 | **36.26** | **41.70** | **21.81** | 33.57 | 37.89 | **18.41** | **30.07** | 35.07 | **14.41** |

| Dataset | WIKIFEA | | | | | | | | | NUSWIENE | | | | | | | | |
|---|---|---|---|---|---|---|---|---|---|---|---|---|---|---|---|---|---|---|
| MR | 0.1 | | | 0.4 | | | 0.7 | | | 0.1 | | | 0.4 | | | 0.7 | | |
| Method | ACC | PUR | FSC | ACC | PUR | FSC | ACC | PUR | FSC | ACC | PUR | FSC | ACC | PUR | FSC | ACC | PUR | FSC |
| GSRMC | 47.09 | 50.48 | 35.00 | 38.22 | 42.73 | 25.61 | 35.96 | 39.61 | 23.54 | 10.18 | 32.19 | 6.74 | 11.42 | 32.33 | 7.32 | 11.10 | 31.56 | 7.40 |
| FLSD | 52.02 | 58.16 | 45.45 | 49.86 | 54.03 | 38.38 | 39.96 | 44.65 | 25.45 | 10.06 | 30.21 | 9.16 | 11.14 | 28.47 | 9.27 | 12.63 | 29.36 | 14.16 |
| MVTSC | 51.83 | 57.97 | 45.38 | 45.10 | 50.63 | 37.98 | 40.48 | 44.63 | 25.57 | 10.22 | 32.74 | 6.50 | 9.51 | **33.65** | 6.43 | 8.39 | 31.34 | 6.05 |
| MVCBG | 51.26 | 58.42 | 47.04 | 49.16 | 53.92 | 38.84 | 40.91 | 44.72 | **30.60** | 6.54 | 3.97 | 7.12 | 6.47 | 3.94 | 9.80 | 6.38 | 3.89 | 9.29 |
| AGCMC | 35.68 | 37.66 | 25.91 | 35.64 | 40.06 | 25.60 | 39.45 | 44.00 | 25.28 | 10.19 | 31.22 | 7.07 | 9.89 | 29.89 | 9.17 | 10.89 | 30.89 | 8.77 |
| NGSPL | 53.25 | 56.75 | 43.80 | 41.19 | 45.64 | 30.03 | 33.59 | 36.66 | 21.75 | 9.72 | 31.82 | 7.35 | 11.86 | 30.40 | 9.25 | 13.78 | 30.10 | 10.98 |
| PIMVC | 52.21 | 54.89 | 40.96 | 46.85 | 50.84 | 34.91 | 39.82 | 43.35 | 26.73 | 10.68 | 30.75 | 6.99 | 10.89 | 31.32 | 6.92 | 10.40 | 31.66 | 6.75 |
| TCIMC | 15.84 | 16.77 | 19.44 | 15.93 | 15.99 | 19.84 | 15.81 | 16.02 | 19.52 | 10.35 | 28.91 | 7.06 | 11.33 | 28.89 | 9.04 | 12.40 | 28.96 | 14.09 |
| TMBSD | 44.42 | 42.93 | 46.42 | 37.30 | 36.05 | 37.92 | 32.33 | 31.31 | 25.09 | 6.54 | 3.94 | **10.52** | 6.28 | 3.78 | 9.54 | 5.98 | 3.60 | 8.01 |
| PSMVC | 52.88 | 58.36 | 46.15 | 50.15 | 53.69 | 38.22 | 39.79 | 45.28 | 26.48 | 9.82 | 32.26 | 6.46 | 10.26 | 32.27 | 6.47 | 9.99 | 32.25 | 6.40 |
| MKKMC | **54.34** | 57.44 | 46.27 | 48.83 | 50.05 | 38.82 | 40.14 | 42.93 | 26.48 | 9.77 | 32.59 | 6.92 | 10.55 | 31.04 | 6.61 | 9.34 | 32.06 | 6.38 |
| LRGMV | 49.89 | 56.97 | 41.04 | 46.52 | 52.88 | 35.84 | 36.51 | 43.29 | 26.93 | 10.33 | 32.41 | 6.90 | 10.77 | 32.16 | 6.83 | 9.97 | 32.99 | 6.59 |
| Ours | 54.22 | **59.74** | **47.23** | **50.17** | **54.50** | **39.21** | **41.70** | **46.41** | 27.03 | **10.99** | **33.57** | 7.59 | **12.48** | 33.52 | **9.87** | **15.41** | 33.15 | **15.10** |

| Dataset | CALTEALL | | | | | | | | | YOUTUTEN | | | | | | | | |
|---|---|---|---|---|---|---|---|---|---|---|---|---|---|---|---|---|---|---|
| MR | 0.1 | | | 0.4 | | | 0.7 | | | 0.1 | | | 0.4 | | | 0.7 | | |
| Method | ACC | PUR | FSC | ACC | PUR | FSC | ACC | PUR | FSC | ACC | PUR | FSC | ACC | PUR | FSC | ACC | PUR | FSC |
| GSRMC | 22.67 | 31.52 | 16.69 | 21.29 | 31.46 | 13.98 | 19.66 | 28.07 | 9.68 | / | | | / | | | / | | |
| FLSD | 18.08 | 34.18 | 14.46 | 16.17 | 33.41 | 12.38 | 14.38 | 28.80 | 9.19 | 52.71 | 55.87 | 58.57 | 47.98 | 53.98 | 59.38 | 45.23 | 52.24 | 57.42 |
| MVTSC | 21.57 | 34.34 | 13.74 | 22.78 | 31.23 | 12.53 | **21.23** | 27.21 | 10.21 | / | | | / | | | / | | |
| MVCBG | 24.77 | 34.13 | 16.12 | **23.54** | 32.65 | 11.83 | 20.19 | 28.31 | 9.68 | **74.07** | 73.53 | 64.35 | 72.55 | 77.44 | 65.47 | 71.22 | 75.94 | 61.05 |
| AGCMC | 17.62 | 34.62 | 10.97 | 18.52 | 32.12 | 10.72 | 17.39 | 28.56 | **10.74** | / | | | / | | | / | | |
| NGSPL | 18.02 | 33.07 | 11.68 | 15.51 | 31.96 | 8.76 | 14.97 | 29.06 | 7.94 | / | | | / | | | / | | |
| PIMVC | / | | | / | | | / | | | 56.49 | 59.23 | 48.93 | 50.58 | 53.88 | 40.34 | 61.37 | 65.36 | 55.03 |
| TCIMC | / | | | / | | | / | | | / | | | / | | | / | | |
| TMBSD | 18.58 | 13.24 | 15.36 | 19.57 | 12.62 | 12.72 | 18.52 | 13.67 | 9.56 | / | | | / | | | / | | |
| PSMVC | 19.02 | 33.44 | 14.72 | 19.11 | 32.57 | 13.33 | 16.58 | 28.21 | 9.11 | 71.56 | 77.35 | **68.32** | 73.77 | 78.49 | 66.56 | 70.09 | **79.09** | 62.03 |
| MKKMC | 20.34 | 34.96 | 15.86 | 16.13 | 31.18 | 11.11 | 13.19 | 27.46 | 8.43 | / | | | / | | | / | | |
| LRGMV | 22.71 | 33.51 | 15.59 | 21.54 | 33.29 | 13.32 | 19.74 | 28.47 | 8.88 | / | | | / | | | / | | |
| Ours | **25.93** | **35.75** | **17.28** | 23.24 | **33.51** | **14.27** | 20.27 | **29.34** | 9.57 | 73.23 | **78.94** | 67.54 | **74.14** | **79.56** | **67.43** | **72.21** | 78.51 | **64.77** |

| Dataset | YOUTUTWE | | | | | | | | | FAMNISIX | | | | | | | | |
|---|---|---|---|---|---|---|---|---|---|---|---|---|---|---|---|---|---|---|
| MR | 0.1 | | | 0.4 | | | 0.7 | | | 0.1 | | | 0.4 | | | 0.7 | | |
| Method | ACC | PUR | FSC | ACC | PUR | FSC | ACC | PUR | FSC | ACC | PUR | FSC | ACC | PUR | FSC | ACC | PUR | FSC |
| GSRMC | / | | | / | | | / | | | / | | | / | | | / | | |
| FLSD | / | | | / | | | / | | | / | | | / | | | / | | |
| MVTSC | / | | | / | | | / | | | / | | | / | | | / | | |
| MVCBG | 68.79 | 69.23 | **66.85** | 68.41 | 68.85 | 61.82 | 68.55 | 69.17 | **63.26** | 56.23 | 56.89 | 44.82 | 54.63 | **57.49** | 44.56 | **55.08** | 56.45 | 43.70 |
| AGCMC | / | | | / | | | / | | | / | | | / | | | / | | |
| NGSPL | / | | | / | | | / | | | / | | | / | | | / | | |
| PIMVC | / | | | / | | | / | | | / | | | / | | | / | | |
| TCIMC | / | | | / | | | / | | | / | | | / | | | / | | |
| TMBSD | / | | | / | | | / | | | / | | | / | | | / | | |
| PSMVC | 65.44 | 73.34 | 61.67 | 71.85 | 76.75 | 61.77 | 67.67 | 74.17 | 60.63 | 49.56 | 54.45 | 42.87 | 49.64 | 53.87 | 41.68 | 48.05 | 51.86 | 39.93 |
| MKKMC | / | | | / | | | / | | | / | | | / | | | / | | |
| LRGMV | / | | | / | | | / | | | / | | | / | | | / | | |
| Ours | **71.23** | **76.86** | 64.74 | **73.86** | **79.51** | **62.06** | **68.62** | **76.91** | 62.90 | **57.97** | **58.35** | **47.84** | **56.34** | 57.27 | **46.89** | 54.52 | **58.43** | **45.93** |

To reflect the performance sufficiently, we record the clustering results under missing ratio as 0.1, 0.4 and 0.7 respectively, with three metrics, ACC, Purity (PUR), Fscore (FSC), as shown in Table 2. The notation '/' denotes cases where the method fails to process the dataset due to its excessive complexity or itself limitations. Combined with this table, we can acquire the following observations,

1. Our BSTM makes more favorable results in most cases. For instance, it is the best in PUR on PROKARYO, WIKIFEA, CALTEALL and YOUTUTWE. For PROTEINF, WIKIFEA, NUSWIENE, YOUTUTWE and FAMNISIX, there are only two sub-optimal cases on each dataset. The consistent performance confirms BSTM's effectiveness in tackling incomplete multi-view clustering problems.

2. The competitors GSRMC, FLSD, MVTSC, AGCMC, NGSPL, PIMVC, etc, are not able to handle YOUTUTWE and FAMNSIX. By contrast, we can operate normally in all scenarios while delivering competitive performance, which indicates that our BSTM possesses a wider range of applications.

3. For inferior results, such as on PROTEINF in PUR, possible reasons could be that we produce the clustering results by partitioning spectral embedding rather than directly by formulating the discrete cluster labels, which deteriorates the data diversity and accordingly harms the model performance.

## 6.2 Resource Usage and Component Ablation

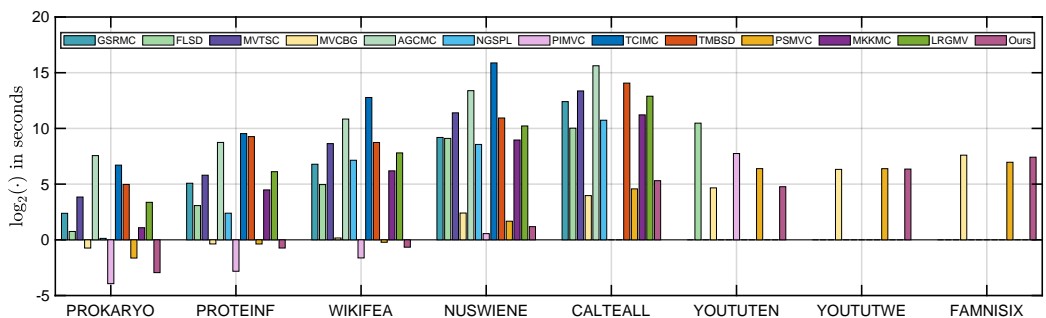

Figure 2: Execution Time Comparison ($\log_2(\cdot)$ in seconds)

To illustrate the execution efficiency, we summarize the running time in Fig. 2. As seen, owing to not involving view-data recalculating, our BSTM usually requires less time overhead. Methods PSMVC and PIMVC in some cases run faster, possibly because they do not integrate basis embedding patterns and also not employ permutation to rearrange anchors. Despite time-saving, combined with Table 2, they typically yield less favorable outcomes. Then, about the space usage, please refer to Section G.

Table 3: Ablation Results about Twin-memory and Bit-swapping on PROKARYO and PROTEINF

| Abla. | PROKARYO | | | | | | | | | PROTEINF | | | | | | | | |
| | 0.1 | | | 0.4 | | | 0.7 | | | 0.1 | | | 0.4 | | | 0.7 | | |
| | ACC | PUR | FSC | ACC | PUR | FSC | ACC | PUR | FSC | ACC | PUR | FSC | ACC | PUR | FSC | ACC | PUR | FSC |
|---|---|---|---|---|---|---|---|---|---|---|---|---|---|---|---|---|---|---|
| Ab-B | 51.54 | 72.60 | 46.61 | 43.99 | 56.81 | 42.66 | 44.61 | 56.98 | 41.62 | 36.49 | 41.92 | 21.67 | 31.99 | 36.53 | 17.08 | 28.65 | 33.17 | 12.51 |
| Ab-Q | 72.78 | 80.76 | 62.80 | 61.34 | 68.97 | 49.03 | 51.24 | 66.64 | 43.22 | 35.61 | 40.44 | 21.70 | 30.43 | 35.34 | 15.66 | 28.09 | 32.81 | 12.07 |
| Ab-bs | 51.24 | 72.41 | 46.39 | 42.83 | 56.64 | 42.69 | 44.11 | 56.61 | 41.51 | 35.17 | 40.66 | 21.27 | 29.97 | 35.70 | 16.49 | 27.31 | 31.74 | 12.28 |
| Ours | **74.95** | **82.21** | **64.79** | **72.78** | **79.13** | **62.82** | **52.09** | **74.59** | **49.59** | **36.26** | **41.70** | **21.81** | **33.57** | **37.89** | **18.41** | **30.07** | **35.07** | **14.41** |

Table 3 demonstrates the effectiveness of twin-memory and bit-swapping (bs). Ab-B and Ab-Q denote the ablation with memoryless $\widetilde{\mathbf{B}}_t$ and $\widetilde{\mathbf{Q}}_t$ respectively. Ab-bs denotes not employing our bs strategy. As seen, each component facilitates the performance. Besides, Fig. 3 visualizes the learned bs $\widehat{\mathbf{S}}_{t-1}$. Yellow denotes 1. As observed, the generated $\widehat{\mathbf{S}}_{t-1}$ is not an identity matrix, meaning that it rearranges anchor topology and graph structure. More ablations are in Section I.

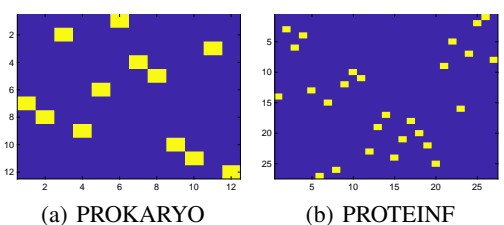

(a) PROKARYO        (b) PROTEINF

Figure 3: Visualization of Learned $\widehat{\mathbf{S}}_{t-1}$

## 7 Limitations

$\lambda$ and $\beta$ need extra efforts to do fine-tuning. So, parameter-free version will improve the practicality. Additionally, the paradigm formulating spectrum and then doing partitioning may degrade the data diversity. Directly producing cluster labels from original samples could further boost the performance.

## 8 Conclusion

This work provides a MVC method termed BSTM to address two limitations, basis embedding overlooking and value-distortion mapping. Rather than previous feature library mechanisms and per-view architectures, it designs a twin-memory paradigm to do knowledge transfer, not only incorporating underlying basis patterns but also eliminating redundant view-data recomputing and avoiding fusion. Further, it introduces a bit-swapping scheme that operates exclusively on binary space to adaptively transform anchor topologies and graph structures while well maintaining numerical consistency. All parts are jointly learned in an end-to-end manner, enabling mutually reinforcing interactions. Then, an efficient four-step solver is developed to minimize the resulting loss, and experiments on several datasets with varying incomplete percentages validate BSTM's effectiveness.

## Acknowledgment

This work was supported by the National Natural Science Foundation of China under Grant No. 62406329, 62476280, 62441618, 62325604, 62276271.

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

# Appendices

## A    Solution Details

Due to the joint non-convexity, we adopt the alternating optimization strategy to minimize the original objective loss (6).

### A.1    Updating the Guidance Variable $\mathbf{G}_t$

Under given $\widetilde{\mathbf{B}}_t$, $\widetilde{\mathbf{Q}}_t$ and $\widehat{\mathbf{S}}_{t-1}$, the original objective loss (6) with respect to $\mathbf{G}_t$ can be equivalently expressed as

$$\min_{\mathbf{G}_t} \|\mathbf{D}_t - \mathbf{G}_t\widetilde{\mathbf{B}}_t\widetilde{\mathbf{Q}}_t\mathbf{H}_t\|_F^2$$
$$\text{s.t. } \mathbf{G}_t^\top \mathbf{G}_t = \mathbf{I}_m. \tag{19}$$

Unfolding $F$-norm through trace operation, we can have

$$\|\mathbf{D}_t - \mathbf{G}_t\widetilde{\mathbf{B}}_t\widetilde{\mathbf{Q}}_t\mathbf{H}_t\|_F^2 \Leftrightarrow \mathrm{Tr}\left(\mathbf{D}_t\mathbf{D}_t^\top - 2\mathbf{D}_t\mathbf{H}_t^\top\widetilde{\mathbf{Q}}_t^\top\widetilde{\mathbf{B}}_t^\top\mathbf{G}_t^\top + \mathbf{G}_t\widetilde{\mathbf{B}}_t\widetilde{\mathbf{Q}}_t\mathbf{H}_t\mathbf{H}_t^\top\widetilde{\mathbf{Q}}_t^\top\widetilde{\mathbf{B}}_t^\top\mathbf{G}_t^\top\right). \tag{20}$$

According to the fact that the data matrix $\mathbf{D}_t$ is irrelevant to $\mathbf{G}_t$ and that the feasible region is orthogonal, we further have

$$\min_{\mathbf{G}_t} \|\mathbf{D}_t - \mathbf{G}_t\widetilde{\mathbf{B}}_t\widetilde{\mathbf{Q}}_t\mathbf{H}_t\|_F^2 \Leftrightarrow \max_{\mathbf{G}_t} \mathrm{Tr}\left(\mathbf{G}_t^\top\mathbf{D}_t\mathbf{H}_t^\top\widetilde{\mathbf{Q}}_t^\top\widetilde{\mathbf{B}}_t^\top\right). \tag{21}$$

Due to $\mathbf{D}_t \in \mathbb{R}^{d_t \times n_t}$, $\mathbf{H}_t \in \mathbb{R}^{\widetilde{n}_t \times n_t}$, $\widetilde{\mathbf{Q}}_t \in \mathbb{R}^{m \times \widetilde{n}_t}$ and $\widetilde{\mathbf{B}}_t \in \mathbb{R}^{m \times m}$, constructing the term $\mathbf{D}_t\mathbf{H}_t^\top\widetilde{\mathbf{Q}}_t^\top\widetilde{\mathbf{B}}_t^\top$ will take $\mathcal{O}(d_tn_t\widetilde{n}_t + d_t\widetilde{n}_tm + d_tm^2)$ computational complexity, which is dominated by $d_tn_t\widetilde{n}_t$ since $m$ is generally smaller than $n_t$ and $\widetilde{n}_t$. When the number of samples observed on the $t$-th view is larger, i.e., the missing ratio is smaller, as time goes on, the complexity will approach $\mathcal{O}(n^2)$, resulting in its unsuitability for large-scale applications.

In order to achieve linear scaling, note that $\widetilde{n}_t$ is greater than $n_t$ and that $\mathbf{H}_t$ is composed of 1 and 0 and there is only one non-zero element in each column. Accordingly, the meaning of $\mathbf{D}_t\mathbf{H}_t^\top$ is to construct a bigger matrix with the size of $d_t \times \widetilde{n}_t$ using the columns of $\mathbf{D}_t$. The non-zero columns of this bigger matrix correspond to the indexes of samples **observed** on the $t$-th view. As a result, rather than directly calculating $\mathbf{D}_t\mathbf{H}_t^\top$, we can firstly create a zero matrix with the size of $d_t \times \widetilde{n}_t$, and then assign its column elements using the columns of $\mathbf{D}_t$ via corresponding indexes. Since it does not involve any element calculating operations and can be constructed before optimization, this will take a constant-order complexity. Consequently, the construction of $\mathbf{D}_t\mathbf{H}_t^\top\widetilde{\mathbf{Q}}_t^\top\widetilde{\mathbf{B}}_t^\top$ requires only $\mathcal{O}(d_t\widetilde{n}_tm)$, linear to the sample size.

Let $\mathbf{U}$, $\mathbf{V}$ and $\boldsymbol{\Sigma}$ denote the singular matrices of $\mathbf{D}_t\mathbf{H}_t^\top\widetilde{\mathbf{Q}}_t^\top\widetilde{\mathbf{B}}_t^\top$, and then we have

$$\max_{\mathbf{G}_t} \mathrm{Tr}\left(\mathbf{G}_t^\top\mathbf{D}_t\mathbf{H}_t^\top\widetilde{\mathbf{Q}}_t^\top\widetilde{\mathbf{B}}_t^\top\right) \Leftrightarrow \max_{\mathbf{G}_t} \mathrm{Tr}\left(\mathbf{V}^\top\mathbf{G}_t^\top\mathbf{U}\boldsymbol{\Sigma}\right). \tag{22}$$

Combined with the orthogonality of $\mathbf{G}_t$, we have that the $\mathbf{V}^\top\mathbf{G}_t^\top\mathbf{U}$ is also an orthogonal matrix. Further, together with the non-negative property of $\boldsymbol{\Sigma}$, we can obtain that $\mathrm{Tr}\left(\mathbf{V}^\top\mathbf{G}_t^\top\mathbf{U}\boldsymbol{\Sigma}\right) \le \mathrm{Tr}\left(\boldsymbol{\Sigma}\right)$. Therefore, we have that when $\mathbf{V}^\top\mathbf{G}_t^\top\mathbf{U} = \mathbf{I}$, the equality holds. Accordingly, the optimal solution of $\mathbf{G}_t$ is $\mathbf{U}\mathbf{V}^\top$.

### A.2    Updating the Unified Anchor Variable $\widetilde{\mathbf{B}}_t$

Under given $\mathbf{G}_t$, $\widetilde{\mathbf{Q}}_t$ and $\widehat{\mathbf{S}}_{t-1}$, the original objective loss (6) with respect to $\widetilde{\mathbf{B}}_t$ can be equivalently expressed as

$$\min_{\widetilde{\mathbf{B}}_t} \|\mathbf{D}_t - \mathbf{G}_t\widetilde{\mathbf{B}}_t\widetilde{\mathbf{Q}}_t\mathbf{H}_t\|_F^2 + \lambda\|\widetilde{\mathbf{B}}_t - \widetilde{\mathbf{B}}_{t-1}\widehat{\mathbf{S}}_{t-1}\|_F^2$$
$$\text{s.t. } \widetilde{\mathbf{B}}_t\widetilde{\mathbf{B}}_t^\top = \mathbf{I}_m. \tag{23}$$

Combined with (20) and the orthogonality of $\widetilde{\mathbf{B}}_t$, we can get

$$\min_{\widetilde{\mathbf{B}}_t} \|\mathbf{D}_t - \mathbf{G}_t\widetilde{\mathbf{B}}_t\widetilde{\mathbf{Q}}_t\mathbf{H}_t\|_F^2 \Leftrightarrow \max_{\widetilde{\mathbf{B}}_t} \operatorname{Tr}\left(\mathbf{G}_t^\top\mathbf{D}_t\mathbf{H}_t^\top\widetilde{\mathbf{Q}}_t^\top\widetilde{\mathbf{B}}_t^\top\right). \tag{24}$$

For the second term in (23), after $F$-norm expanding, we have

$$\|\widetilde{\mathbf{B}}_t - \widetilde{\mathbf{B}}_{t-1}\widehat{\mathbf{S}}_{t-1}\|_F^2 = \operatorname{Tr}\left(\widetilde{\mathbf{B}}_t\widetilde{\mathbf{B}}_t^\top - 2\widetilde{\mathbf{B}}_t^\top\widetilde{\mathbf{B}}_{t-1}\widehat{\mathbf{S}}_{t-1} + \widetilde{\mathbf{B}}_{t-1}\widehat{\mathbf{S}}_{t-1}\widehat{\mathbf{S}}_{t-1}^\top\widetilde{\mathbf{B}}_{t-1}^\top\right). \tag{25}$$

Due to $\widetilde{\mathbf{B}}_t\widetilde{\mathbf{B}}_t^\top = \mathbf{I}_m$ and $\widetilde{\mathbf{B}}_{t-1}\widehat{\mathbf{S}}_{t-1}$ being irrelevant to the optimization variable $\widetilde{\mathbf{B}}_t$, we can derive

$$\min_{\widetilde{\mathbf{B}}_t} \|\widetilde{\mathbf{B}}_t - \widetilde{\mathbf{B}}_{t-1}\widehat{\mathbf{S}}_{t-1}\|_F^2 \Leftrightarrow \max_{\widetilde{\mathbf{B}}_t} \operatorname{Tr}\left(\widetilde{\mathbf{B}}_t^\top\widetilde{\mathbf{B}}_{t-1}\widehat{\mathbf{S}}_{t-1}\right). \tag{26}$$

Combing (24) and (26), we can derive that the problem (23) is equivalently transformed as

$$\max_{\widetilde{\mathbf{B}}_t} \operatorname{Tr}\left(\left(\mathbf{G}_t^\top\mathbf{D}_t\mathbf{H}_t^\top\widetilde{\mathbf{Q}}_t^\top + \lambda\widetilde{\mathbf{B}}_{t-1}\widehat{\mathbf{S}}_{t-1}\right)\widetilde{\mathbf{B}}_t^\top\right)$$
$$\text{s.t. } \widetilde{\mathbf{B}}_t\widetilde{\mathbf{B}}_t^\top = \mathbf{I}_m. \tag{27}$$

Therefore, the optimal solution of $\widetilde{\mathbf{B}}_t$ is $\mathbf{U}\mathbf{V}^\top$, in which $\mathbf{U}$ and $\mathbf{V}$ come from the SVD results of $\mathbf{G}_t^\top\mathbf{D}_t\mathbf{H}_t^\top\widetilde{\mathbf{Q}}_t^\top + \lambda\widetilde{\mathbf{B}}_{t-1}\widehat{\mathbf{S}}_{t-1}$.

Directly calculating $\mathbf{G}_t^\top\mathbf{D}_t\mathbf{H}_t^\top\widetilde{\mathbf{Q}}_t^\top + \lambda\widetilde{\mathbf{B}}_{t-1}\widehat{\mathbf{S}}_{t-1}$ will take $\mathcal{O}(md_tn_t + mn_t\widetilde{n}_t + m^2\widetilde{n}_t + m^3)$ computational cost. Evidently, it will be square with respect to $n$ over time, causing difficulties in tackling large-scale problems. Inspired by the strategy handling $\mathbf{D}_t\mathbf{H}_t^\top$ in the optimization procedure of $\mathbf{G}_t$, we first equivalently calculate $\mathbf{D}_t\mathbf{H}_t^\top$ via a bigger temporary matrix, and then construct the term $\mathbf{G}_t^\top\mathbf{D}_t\mathbf{H}_t^\top\widetilde{\mathbf{Q}}_t^\top$. Accordingly, the complexity is reduced to $\mathcal{O}(md_t\widetilde{n}_t + m^2\widetilde{n}_t + m^3)$. Due to $m$ generally being smaller than $d_t$ and $\widetilde{n}_t$, the computational complexity about $\widetilde{\mathbf{B}}_t$ can be formulated as $\mathcal{O}(md_t\widetilde{n}_t)$, which is linear to the sample size.

### A.3  Updating the Shared Bipartite Graph $\widetilde{\mathbf{Q}}_t$

Under given $\mathbf{G}_t$, $\widetilde{\mathbf{B}}_t$ and $\widehat{\mathbf{S}}_{t-1}$, the original objective loss (6) with respect to $\widetilde{\mathbf{Q}}_t$ can be equivalently expressed as

$$\min_{\widetilde{\mathbf{Q}}_t} \|\mathbf{D}_t - \mathbf{G}_t\widetilde{\mathbf{B}}_t\widetilde{\mathbf{Q}}_t\mathbf{H}_t\|_F^2 + \beta\|\widetilde{\mathbf{Q}}_t\widetilde{\mathbf{E}}_{t-1} - \widehat{\mathbf{S}}_{t-1}^\top\widetilde{\mathbf{Q}}_{t-1}\|_F^2$$
$$\text{s.t. } \widetilde{\mathbf{Q}}_t \geq 0, \widetilde{\mathbf{Q}}_t^\top\mathbf{1}_m = \mathbf{1}_{\widetilde{n}_t}. \tag{28}$$

Together with (20), we can further obtain

$$\min_{\widetilde{\mathbf{Q}}_t} \|\mathbf{D}_t - \mathbf{G}_t\widetilde{\mathbf{B}}_t\widetilde{\mathbf{Q}}_t\mathbf{H}_t\|_F^2 \Leftrightarrow \min_{\widetilde{\mathbf{Q}}_t} \operatorname{Tr}\left(\widetilde{\mathbf{Q}}_t^\top\widetilde{\mathbf{B}}_t^\top\mathbf{G}_t^\top\mathbf{G}_t\widetilde{\mathbf{B}}_t\widetilde{\mathbf{Q}}_t\mathbf{H}_t\mathbf{H}_t^\top - 2\widetilde{\mathbf{Q}}_t^\top\widetilde{\mathbf{B}}_t^\top\mathbf{G}_t^\top\mathbf{D}_t\mathbf{H}_t^\top\right). \tag{29}$$

According to the characteristics of feasible region, we can optimize $\widetilde{\mathbf{Q}}_t$ by column. Specially, we have

$$\min_{\widetilde{\mathbf{Q}}_t} \operatorname{Tr}\left(\widetilde{\mathbf{Q}}_t^\top\widetilde{\mathbf{B}}_t^\top\mathbf{G}_t^\top\mathbf{G}_t\widetilde{\mathbf{B}}_t\widetilde{\mathbf{Q}}_t\mathbf{H}_t\mathbf{H}_t^\top\right) \Leftrightarrow \min_{[\widetilde{\mathbf{Q}}_t]_{:,j}} \left[\widetilde{\mathbf{Q}}_t\right]_{:,j}^\top\widetilde{\mathbf{B}}_t^\top\mathbf{G}_t^\top\mathbf{G}_t\widetilde{\mathbf{B}}_t\left[\widetilde{\mathbf{Q}}_t\mathbf{H}_t\mathbf{H}_t^\top\right]_{:,j}. \tag{30}$$

Considering that these is only one non-zero element in each column of $\mathbf{H}_t$ and that all columns are orthogonal, $\mathbf{H}_t\mathbf{H}_t^\top \in \mathbb{R}^{\widetilde{n}_t\times\widetilde{n}_t}$ will be a diagonal matrix with element 0 and 1. Directly calculating $\widetilde{\mathbf{Q}}_t\mathbf{H}_t\mathbf{H}_t^\top$ will need $\mathcal{O}(m\widetilde{n}_t^2)$ computational complexity. As time goes on, it will be quadratic with respect to $n$, and consequently is not applicable to large-scale scenes. To decrease the complexity, by virtue of the diagonal property, we can derive $\widetilde{\mathbf{Q}}_t\mathbf{H}_t\mathbf{H}_t^\top = \widetilde{\mathbf{Q}}_t \odot \mathbf{J}_t$ where $\mathbf{J}_t = \mathbf{1}_m \cdot [\sum_{j=1}^{n_t}[\mathbf{H}_t]_{1,j}, \sum_{j=1}^{n_t}[\mathbf{H}_t]_{2,j}, \cdots, \sum_{j=1}^{n_t}[\mathbf{H}_t]_{\widetilde{n}_t,j}]$. Further, we have

$$\left[\widetilde{\mathbf{Q}}_t\mathbf{H}_t\mathbf{H}_t^\top\right]_{:,j} = \sum_{l=1}^{n_t}[\mathbf{H}_t]_{j,l}\left[\widetilde{\mathbf{Q}}_t\right]_{:,j}. \tag{31}$$

Combined with (30), we can equivalently obtain

$$\min_{\widetilde{\mathbf{Q}}_t} \operatorname{Tr}\left(\widetilde{\mathbf{Q}}_t^\top \widetilde{\mathbf{B}}_t^\top \mathbf{G}_t^\top \mathbf{G}_t \widetilde{\mathbf{B}}_t \widetilde{\mathbf{Q}}_t \mathbf{H}_t \mathbf{H}_t^\top\right) \Leftrightarrow \min_{[\widetilde{\mathbf{Q}}_t]_{:,j}} \left[\widetilde{\mathbf{Q}}_t\right]_{:,j}^\top \sum_{l=1}^{n_t} [\mathbf{H}_t]_{j,l}\, \widetilde{\mathbf{B}}_t^\top \mathbf{G}_t^\top \mathbf{G}_t \widetilde{\mathbf{B}}_t \left[\widetilde{\mathbf{Q}}_t\right]_{:,j}. \quad (32)$$

Therefore, for (29), we have the following expression holds,

$$\min_{\widetilde{\mathbf{Q}}_t} \|\mathbf{D}_t - \mathbf{G}_t \widetilde{\mathbf{B}}_t \widetilde{\mathbf{Q}}_t \mathbf{H}_t\|_F^2 \Leftrightarrow \min_{[\widetilde{\mathbf{Q}}_t]_{:,j}} \left[\widetilde{\mathbf{Q}}_t\right]_{:,j}^\top \sum_{l=1}^{n_t} [\mathbf{H}_t]_{j,l}\, \widetilde{\mathbf{B}}_t^\top \mathbf{G}_t^\top \mathbf{G}_t \widetilde{\mathbf{B}}_t \left[\widetilde{\mathbf{Q}}_t\right]_{:,j}$$
$$-2 \left[\widetilde{\mathbf{Q}}_t\right]_{:,j}^\top \left[\widetilde{\mathbf{B}}_t^\top \mathbf{G}_t^\top \mathbf{D}_t \mathbf{H}_t^\top\right]_{:,j}. \quad (33)$$

Then, for the second term in (28), after expanding and deleting irrelevant items, we have

$$\min_{\widetilde{\mathbf{Q}}_t} \|\widetilde{\mathbf{Q}}_t \widetilde{\mathbf{E}}_{t-1} - \widehat{\mathbf{S}}_{t-1}^\top \widetilde{\mathbf{Q}}_{t-1}\|_F^2 \Leftrightarrow \min_{\widetilde{\mathbf{Q}}_t} \operatorname{Tr}\left(\widetilde{\mathbf{Q}}_t^\top \widetilde{\mathbf{Q}}_t \widetilde{\mathbf{E}}_{t-1} \widetilde{\mathbf{E}}_{t-1}^\top - 2\widetilde{\mathbf{Q}}_t^\top \widehat{\mathbf{S}}_{t-1}^\top \widetilde{\mathbf{Q}}_{t-1} \widetilde{\mathbf{E}}_{t-1}^\top\right). \quad (34)$$

Note that $\widetilde{\mathbf{E}}_{t-1} \in \mathbb{R}^{\widetilde{n}_t \times \widetilde{n}_{t-1}}$, the computing of $\widetilde{\mathbf{E}}_{t-1}\widetilde{\mathbf{E}}_{t-1}^\top$ will need $\mathcal{O}(\widetilde{n}_t^2 \widetilde{n}_{t-1})$ complexity. Over time, this will be cubic with respect to $n$, severely harming the ability to tackle large-scale tasks. Inspired by the scheme disposing of $\mathbf{H}_t$, we observe that $\widetilde{\mathbf{E}}_{t-1}$ is a skinny matrix with binary elements, the columns of which are mutually orthogonal. Consequently, $\widetilde{\mathbf{E}}_{t-1}\widetilde{\mathbf{E}}_{t-1}^\top$ is a diagonal matrix with elements as the row sums of $\widetilde{\mathbf{E}}_{t-1}$ respectively. Further, we have that $\widetilde{\mathbf{Q}}_t\widetilde{\mathbf{E}}_{t-1}\widetilde{\mathbf{E}}_{t-1}^\top$ is equal to $\widetilde{\mathbf{Q}}_t \odot \mathbf{M}_t$. $\mathbf{M}_t = \mathbf{1}_m \cdot [\sum_{j=1}^{\widetilde{n}_{t-1}} [\widetilde{\mathbf{E}}_{t-1}]_{1,j}, \sum_{j=1}^{\widetilde{n}_{t-1}} [\widetilde{\mathbf{E}}_{t-1}]_{2,j}, \cdots, \sum_{j=1}^{\widetilde{n}_{t-1}} [\widetilde{\mathbf{E}}_{t-1}]_{\widetilde{n}_t,j}]$. Then, by column-wise decomposition, we have

$$\min_{\widetilde{\mathbf{Q}}_t} \operatorname{Tr}\left(\widetilde{\mathbf{Q}}_t^\top \widetilde{\mathbf{Q}}_t \widetilde{\mathbf{E}}_{t-1} \widetilde{\mathbf{E}}_{t-1}^\top\right) \Leftrightarrow \min_{[\widetilde{\mathbf{Q}}_t]_{:,j}} \left[\widetilde{\mathbf{Q}}_t\right]_{:,j}^\top \sum_{l=1}^{\widetilde{n}_{t-1}} \left[\widetilde{\mathbf{E}}_{t-1}\right]_{j,l} \left[\widetilde{\mathbf{Q}}_t\right]_{:,j} \quad (35)$$

and

$$\min_{\widetilde{\mathbf{Q}}_t} \operatorname{Tr}\left(\widetilde{\mathbf{Q}}_t^\top \widehat{\mathbf{S}}_{t-1}^\top \widetilde{\mathbf{Q}}_{t-1} \widetilde{\mathbf{E}}_{t-1}^\top\right) \Leftrightarrow \min_{[\widetilde{\mathbf{Q}}_t]_{:,j}} \left[\widetilde{\mathbf{Q}}_t\right]_{:,j}^\top \left[\widehat{\mathbf{S}}_{t-1}^\top \widetilde{\mathbf{Q}}_{t-1} \widetilde{\mathbf{E}}_{t-1}^\top\right]_{:,j}. \quad (36)$$

Together with (33), (34) (35) and (36), we can equivalently simplify (28) as

$$\min_{[\widetilde{\mathbf{Q}}_t]_{:,j}} \left[\widetilde{\mathbf{Q}}_t\right]_{:,j}^\top \left(\sum_{l=1}^{n_t} [\mathbf{H}_t]_{j,l}\, \widetilde{\mathbf{B}}_t^\top \mathbf{G}_t^\top \mathbf{G}_t \widetilde{\mathbf{B}}_t + \beta \sum_{l=1}^{\widetilde{n}_{t-1}} \left[\widetilde{\mathbf{E}}_{t-1}\right]_{j,l} \mathbf{I}_m\right) \left[\widetilde{\mathbf{Q}}_t\right]_{:,j}$$
$$-2\left[\widetilde{\mathbf{Q}}_t\right]_{:,j}^\top \left(\left[\widetilde{\mathbf{B}}_t^\top \mathbf{G}_t^\top \mathbf{D}_t \mathbf{H}_t^\top\right]_{:,j} + \beta \left[\widehat{\mathbf{S}}_{t-1}^\top \widetilde{\mathbf{Q}}_{t-1} \widetilde{\mathbf{E}}_{t-1}^\top\right]_{:,j}\right) \quad (37)$$
$$\text{s.t. } \left[\widetilde{\mathbf{Q}}_t\right]_{:,j} \geq 0, \left[\widetilde{\mathbf{Q}}_t\right]_{:,j}^\top \mathbf{1}_m = 1,\ j = 1, 2, \cdots, \widetilde{n}_t.$$

In conjunction with the orthogonality of $\mathbf{G}_t$ and $\widetilde{\mathbf{B}}_t$, we can further have the following equivalent optimization problem,

$$\min_{[\widetilde{\mathbf{Q}}_t]_{:,j}} \left(\sum_{l=1}^{n_t} [\mathbf{H}_t]_{j,l} + \beta \sum_{l=1}^{\widetilde{n}_{t-1}} \left[\widetilde{\mathbf{E}}_{t-1}\right]_{j,l}\right) \left[\widetilde{\mathbf{Q}}_t\right]_{:,j}^\top \left[\widetilde{\mathbf{Q}}_t\right]_{:,j}$$
$$-2\left[\widetilde{\mathbf{Q}}_t\right]_{:,j}^\top \left(\left[\widetilde{\mathbf{B}}_t^\top \mathbf{G}_t^\top \mathbf{D}_t \mathbf{H}_t^\top\right]_{:,j} + \beta \left[\widehat{\mathbf{S}}_{t-1}^\top \widetilde{\mathbf{Q}}_{t-1} \widetilde{\mathbf{E}}_{t-1}^\top\right]_{:,j}\right) \quad (38)$$
$$\text{s.t. } \left[\widetilde{\mathbf{Q}}_t\right]_{:,j} \geq 0, \left[\widetilde{\mathbf{Q}}_t\right]_{:,j}^\top \mathbf{1}_m = 1,\ j = 1, 2, \cdots, \widetilde{n}_t.$$

According to the meaning of $\mathbf{H}_t$ and $\widetilde{\mathbf{E}}_{t-1}$ that $\mathbf{H}_t$ measures whether the samples are on the $t$-th view and $\widetilde{\mathbf{E}}_{t-1}$ measures whether the samples are on all previous $t-1$ views, we can obtain that $\sum_{l=1}^{n_t} [\mathbf{H}_t]_{j,l} + \sum_{l=1}^{\widetilde{n}_{t-1}} [\widetilde{\mathbf{E}}_{t-1}]_{j,l}$ indicates the sample $j$ must be in all $t$ views. Therefore, we can

derive that the coefficient $\sum_{l=1}^{n_t} [\mathbf{H}_t]_{j,l} + \beta \sum_{l=1}^{\widetilde{n}_{t-1}} [\widetilde{\mathbf{E}}_{t-1}]_{j,l}$ must be non-zero. On the basis of this characteristic, we can further transform (38) as

$$
\min_{[\widetilde{\mathbf{Q}}_t]_{:,j}} \left\| \left[\widetilde{\mathbf{Q}}_t\right]_{:,j} - \frac{\left[\widetilde{\mathbf{B}}_t^\top \mathbf{G}_t^\top \mathbf{D}_t \mathbf{H}_t^\top\right]_{:,j} + \beta \left[\widehat{\mathbf{S}}_{t-1}^\top \widetilde{\mathbf{Q}}_{t-1} \widetilde{\mathbf{E}}_{t-1}^\top\right]_{:,j}}{\sum_{l=1}^{n_t} [\mathbf{H}_t]_{j,l} + \beta \sum_{l=1}^{\widetilde{n}_{t-1}} \left[\widetilde{\mathbf{E}}_{t-1}\right]_{j,l}} \right\|_F^2 \tag{39}
$$

$$
\text{s.t.} \ \left[\widetilde{\mathbf{Q}}_t\right]_{:,j} \geq 0, \ \left[\widetilde{\mathbf{Q}}_t\right]_{:,j}^\top \mathbf{1}_m = 1, \ j = 1, 2, \cdots, \widetilde{n}_t.
$$

Its Lagrangian function can be expressed as

$$
\mathcal{J}([\widetilde{\mathbf{Q}}_t]_{:,j}, \psi, \tau) = \min_{[\widetilde{\mathbf{Q}}_t]_{:,j}} \left\| \left[\widetilde{\mathbf{Q}}_t\right]_{:,j} - \mathbf{T}_{:,j} \right\|_F^2 - \psi^\top \left[\widetilde{\mathbf{Q}}_t\right]_{:,j} - \tau \left( \left[\widetilde{\mathbf{Q}}_t\right]_{:,j}^\top \mathbf{1}_m - 1 \right), \tag{40}
$$

where

$$
\mathbf{T}_{:,j} = \left( \left[\widetilde{\mathbf{B}}_t^\top \mathbf{G}_t^\top \mathbf{D}_t \mathbf{H}_t^\top + \beta \widehat{\mathbf{S}}_{t-1}^\top \widetilde{\mathbf{Q}}_{t-1} \widetilde{\mathbf{E}}_{t-1}^\top\right]_{:,j} \right) / \left( \sum_{l=1}^{n_t} [\mathbf{H}_t]_{j,l} + \beta \sum_{l=1}^{\widetilde{n}_{t-1}} \left[\widetilde{\mathbf{E}}_{t-1}\right]_{j,l} \right), \tag{41}
$$

$\psi \in \mathbb{R}^{m \times 1} \geq 0$ and $\tau \in \mathbb{R}^{1 \times 1}$ are the Lagrange multipliers.

By virtue of KTT conditions, we have

$$
\left[\widetilde{\mathbf{Q}}_t\right]_{:,j} - \mathbf{T}_{:,j} - \tau \mathbf{1}_m = \psi \tag{42}
$$

and

$$
\psi_i \left[\widetilde{\mathbf{Q}}_t\right]_{i,j} = 0, \ i = 1, 2, 3, \cdots, m. \tag{43}
$$

According to the feasible region characteristic that the column sum is equal to 1, we can further obtain

$$
\tau = \frac{1 - \mathbf{1}_m^\top \mathbf{T}_{:,j}}{m}. \tag{44}
$$

Therefore, we have

$$
\left[\widetilde{\mathbf{Q}}_t\right]_{i,j} = \left[ \mathbf{T}_{:,j} + \frac{\mathbf{1}_m - \mathbf{1}_m^\top \mathbf{T}_{:,j} \mathbf{1}_m}{m} \right]_i \tag{45}
$$

if $\left[ \frac{\mathbf{1}_m - \mathbf{1}_m^\top \mathbf{T}_{:,j} \mathbf{1}_m}{m} \right]_i \geq -\mathbf{T}_{i,j}$ otherwise 0, $i = 1, 2, \cdots, m; j = 1, 2, \cdots, \widetilde{n}_t$.

Since it exists closed-form solution, the computational overhead mainly comes from the construction of $\mathbf{T}_{:,j}$, especially from $\widetilde{\mathbf{B}}_t^\top \mathbf{G}_t^\top \mathbf{D}_t \mathbf{H}_t^\top + \beta \widehat{\mathbf{S}}_{t-1}^\top \widetilde{\mathbf{Q}}_{t-1} \widetilde{\mathbf{E}}_{t-1}^\top$. Combined with the matrix size, we have that constructing $\widetilde{\mathbf{B}}_t^\top \mathbf{G}_t^\top \mathbf{D}_t \mathbf{H}_t^\top$ and $\widehat{\mathbf{S}}_{t-1}^\top \widetilde{\mathbf{Q}}_{t-1} \widetilde{\mathbf{E}}_{t-1}^\top$ will take $\mathcal{O}(m^2 d_t + m d_t n_t + m n_t \widetilde{n}_t)$ and $\mathcal{O}(m^2 \widetilde{n}_{t-1} + m \widetilde{n}_t \widetilde{n}_{t-1})$ computing cost respectively, which are both square with respect to the sample size.

To reduce it, we first construct $\mathbf{D}_t \mathbf{H}_t^\top$ via a zero-matrix with larger size and then assign elements as done in the optimization about $\mathbf{G}_t$. Consequently, generating $\widetilde{\mathbf{B}}_t^\top \mathbf{G}_t^\top \mathbf{D}_t \mathbf{H}_t^\top$ takes $\mathcal{O}(m d_t \widetilde{n}_t)$ complexity. Subsequently, for $\widehat{\mathbf{S}}_{t-1}^\top \widetilde{\mathbf{Q}}_{t-1} \widetilde{\mathbf{E}}_{t-1}^\top$, due to $\widetilde{\mathbf{Q}}_{t-1} \in \mathbb{R}^{m \times \widetilde{n}_{t-1}}$ and $\widetilde{\mathbf{E}}_{t-1}^\top \in \mathbb{R}^{\widetilde{n}_{t-1} \times \widetilde{n}_t}$ as well as $\widetilde{n}_t$ being greater than or equal to $\widetilde{n}_{t-1}$, combined with the 0-1 property and column orthogonality of $\widetilde{\mathbf{E}}_{t-1}$, we have that $\widetilde{\mathbf{Q}}_{t-1} \widetilde{\mathbf{E}}_{t-1}^\top$ indicates taking the columns of $\widetilde{\mathbf{Q}}_{t-1}$ to construct a larger matrix with the size of $m \times \widetilde{n}_t$ (i.e., the same as the size of $\widetilde{\mathbf{Q}}_t$). Therefore, instead of directly computing $\widetilde{\mathbf{Q}}_{t-1} \widetilde{\mathbf{E}}_{t-1}^\top$, we establish a bigger zero-matrix with the size of $m \times \widetilde{n}_t$, and then assign its columns using the columns of $\widetilde{\mathbf{Q}}_{t-1}$ under corresponding indexes. As a result, constructing $\widehat{\mathbf{S}}_{t-1}^\top \widetilde{\mathbf{Q}}_{t-1} \widetilde{\mathbf{E}}_{t-1}^\top$ takes $\mathcal{O}(m^2 \widetilde{n}_t)$ computing overhead.

Therefore, we have that the computational complexity of constructing the term $\widetilde{\mathbf{B}}_t^\top \mathbf{G}_t^\top \mathbf{D}_t \mathbf{H}_t^\top + \beta \widehat{\mathbf{S}}_{t-1}^\top \widetilde{\mathbf{Q}}_{t-1} \widetilde{\mathbf{E}}_{t-1}^\top$ is $\mathcal{O}(m d_t \widetilde{n}_t + m^2 \widetilde{n}_t)$. Correspondingly, optimizing $\widetilde{\mathbf{Q}}_t$ will require $\mathcal{O}(m d_t \widetilde{n}_t + m^2 \widetilde{n}_t)$ computing cost.

## A.4 Updating the Swapping Variable $\widehat{\mathbf{S}}_{t-1}$

Under given $\mathbf{G}_t$, $\widetilde{\mathbf{B}}_t$ and $\widetilde{\mathbf{Q}}_t$, the original objective loss (6) with respect to $\widehat{\mathbf{S}}_{t-1}$ can be equivalently expressed as

$$\min_{\widehat{\mathbf{S}}_{t-1}} \ \lambda\|\widetilde{\mathbf{B}}_t - \widetilde{\mathbf{B}}_{t-1}\widehat{\mathbf{S}}_{t-1}\|_F^2 + \beta\|\widetilde{\mathbf{Q}}_t\widetilde{\mathbf{E}}_{t-1} - \widehat{\mathbf{S}}_{t-1}^\top\widetilde{\mathbf{Q}}_{t-1}\|_F^2$$
$$\text{s.t. } \widehat{\mathbf{S}}_{t-1} \in \{0,1\}, \widehat{\mathbf{S}}_{t-1}^\top\mathbf{1}_m = \mathbf{1}_m, \widehat{\mathbf{S}}_{t-1}\mathbf{1}_m = \mathbf{1}_m. \tag{46}$$

On the basis of (25), for the term $\|\widetilde{\mathbf{B}}_t - \widetilde{\mathbf{B}}_{t-1}\widehat{\mathbf{S}}_{t-1}\|_F^2$, we have

$$\min_{\widehat{\mathbf{S}}_{t-1}} \|\widetilde{\mathbf{B}}_t - \widetilde{\mathbf{B}}_{t-1}\widehat{\mathbf{S}}_{t-1}\|_F^2 \Leftrightarrow \min_{\widehat{\mathbf{S}}_{t-1}} \ \mathrm{Tr}\left(\widehat{\mathbf{S}}_{t-1}\widehat{\mathbf{S}}_{t-1}^\top\widetilde{\mathbf{B}}_{t-1}^\top\widetilde{\mathbf{B}}_{t-1} - 2\widetilde{\mathbf{B}}_t^\top\widetilde{\mathbf{B}}_{t-1}\widehat{\mathbf{S}}_{t-1}\right). \tag{47}$$

Combined with the characteristics of feasible region, i.e., $\widehat{\mathbf{S}}_{t-1}$ is a square matrix with elements as 0 and 1, and there is only one 1 in every column and every row, we can derive

$$\widehat{\mathbf{S}}_{t-1}\widehat{\mathbf{S}}_{t-1}^\top = \mathbf{I}_m. \tag{48}$$

Therefore, we have

$$\min_{\widehat{\mathbf{S}}_{t-1}} \|\widetilde{\mathbf{B}}_t - \widetilde{\mathbf{B}}_{t-1}\widehat{\mathbf{S}}_{t-1}\|_F^2 \Leftrightarrow \max_{\widehat{\mathbf{S}}_{t-1}} \ \mathrm{Tr}\left(\widetilde{\mathbf{B}}_t^\top\widetilde{\mathbf{B}}_{t-1}\widehat{\mathbf{S}}_{t-1}\right). \tag{49}$$

For the term $\|\widetilde{\mathbf{Q}}_t\widetilde{\mathbf{E}}_{t-1} - \widehat{\mathbf{S}}_{t-1}^\top\widetilde{\mathbf{Q}}_{t-1}\|_F^2$ in (46), after expanding and removing irrelevant items, we can obtain

$$\min_{\widehat{\mathbf{S}}_{t-1}} \|\widetilde{\mathbf{Q}}_t\widetilde{\mathbf{E}}_{t-1} - \widehat{\mathbf{S}}_{t-1}^\top\widetilde{\mathbf{Q}}_{t-1}\|_F^2 \Leftrightarrow \min_{\widehat{\mathbf{S}}_{t-1}} \ \mathrm{Tr}\left(\widehat{\mathbf{S}}_{t-1}^\top\widetilde{\mathbf{Q}}_{t-1}\widetilde{\mathbf{Q}}_{t-1}^\top\widehat{\mathbf{S}}_{t-1} - 2\widetilde{\mathbf{Q}}_t\widetilde{\mathbf{E}}_{t-1}\widetilde{\mathbf{Q}}_{t-1}^\top\widehat{\mathbf{S}}_{t-1}\right). \tag{50}$$

Through trace cycle, accordingly we have

$$\min_{\widehat{\mathbf{S}}_{t-1}} \|\widetilde{\mathbf{Q}}_t\widetilde{\mathbf{E}}_{t-1} - \widehat{\mathbf{S}}_{t-1}^\top\widetilde{\mathbf{Q}}_{t-1}\|_F^2 \Leftrightarrow \max_{\widehat{\mathbf{S}}_{t-1}} \ \mathrm{Tr}\left(\widetilde{\mathbf{Q}}_t\widetilde{\mathbf{E}}_{t-1}\widetilde{\mathbf{Q}}_{t-1}^\top\widehat{\mathbf{S}}_{t-1}\right). \tag{51}$$

Together with (49) and (51), we have that the problem (46) is equivalently simplified as

$$\max_{\widehat{\mathbf{S}}_{t-1}} \ \mathrm{Tr}\left(\left(\lambda\widetilde{\mathbf{B}}_t^\top\widetilde{\mathbf{B}}_{t-1} + \beta\widetilde{\mathbf{Q}}_t\widetilde{\mathbf{E}}_{t-1}\widetilde{\mathbf{Q}}_{t-1}^\top\right)\widehat{\mathbf{S}}_{t-1}\right)$$
$$\text{s.t. } \widehat{\mathbf{S}}_{t-1} \in \{0,1\}, \widehat{\mathbf{S}}_{t-1}^\top\mathbf{1}_m = \mathbf{1}_m, \widehat{\mathbf{S}}_{t-1}\mathbf{1}_m = \mathbf{1}_m. \tag{52}$$

In virtue of the matrix vectorization transformation $\mathrm{Vec}(\cdot)$, we can transform (52) as the following equivalent optimization problem,

$$\max_{\widehat{\mathbf{S}}_{t-1}} \ \left(\mathrm{Vec}\left(\left(\lambda\widetilde{\mathbf{B}}_t^\top\widetilde{\mathbf{B}}_{t-1} + \beta\widetilde{\mathbf{Q}}_t\widetilde{\mathbf{E}}_{t-1}\widetilde{\mathbf{Q}}_{t-1}^\top\right)^\top\right)\right)^\top \mathrm{Vec}\left(\widehat{\mathbf{S}}_{t-1}\right)$$
$$\text{s.t. } \widehat{\mathbf{S}}_{t-1} \in \{0,1\}, \widehat{\mathbf{S}}_{t-1}^\top\mathbf{1}_m = \mathbf{1}_m, \widehat{\mathbf{S}}_{t-1}\mathbf{1}_m = \mathbf{1}_m. \tag{53}$$

This is an integer linear programming problem, and can be solved within cubic computational complexity with respect to the number of variables via existing software. Kindly note that $\widehat{\mathbf{S}}_{t-1}$ is in $\mathbb{R}^{m \times m}$ and $m$ is greatly smaller than the number of samples. Accordingly, this requires minor computational overhead.

Subsequently, it needs to construct the term $\lambda\widetilde{\mathbf{B}}_t^\top\widetilde{\mathbf{B}}_{t-1} + \beta\widetilde{\mathbf{Q}}_t\widetilde{\mathbf{E}}_{t-1}\widetilde{\mathbf{Q}}_{t-1}^\top$. Direct calculation takes $\mathcal{O}(m^3 + m\widetilde{n}_t\widetilde{n}_{t-1} + m^2\widetilde{n}_{t-1})$ cost, which is almost close to the square with respect to the sample size $n$. To decrease it, combined with the fact that there is only one 1 in each column of $\widetilde{\mathbf{E}}_{t-1}$ and other elements are all 0, we have that the product between $\widetilde{\mathbf{Q}}_t$ and $\widetilde{\mathbf{E}}_{t-1}$ means picking out some columns of $\widetilde{\mathbf{Q}}_t$ to generate a smaller matrix. Therefore, rather than direct calculation, we establish an auxiliary matrix with the size of $m \times \widetilde{n}_{t-1}$, and then utilize the columns of $\widetilde{\mathbf{Q}}_t$ to do assignment for it via the sample indexes observed on all $t-1$ views. Then, we utilize this auxiliary matrix to construct $\widetilde{\mathbf{Q}}_t\widetilde{\mathbf{E}}_{t-1}\widetilde{\mathbf{Q}}_{t-1}^\top$. Accordingly, building $\lambda\widetilde{\mathbf{B}}_t^\top\widetilde{\mathbf{B}}_{t-1} + \beta\widetilde{\mathbf{Q}}_t\widetilde{\mathbf{E}}_{t-1}\widetilde{\mathbf{Q}}_{t-1}^\top$ will take $\mathcal{O}(m^3 + m^2\widetilde{n}_{t-1})$ computational cost.

In conjunction with the fact that $m$ is generally greatly smaller than $\widetilde{n}_{t-1}$, we can obtain that updating $\widehat{\mathbf{S}}_{t-1}$ will take $\mathcal{O}(m^2\widetilde{n}_{t-1})$ cost, which is linear to the number of samples and consequently does not damage the large-scale ability.

## A.5 Updating Variables under Initial View

Under $t = 1$, the swapping variable $\widehat{\mathbf{S}}_{t-1}$ is not required, and accordingly the objective loss equivalently becomes

$$
\min_{\mathbf{G}_t, \widetilde{\mathbf{B}}_t, \widetilde{\mathbf{Q}}_t} \|\mathbf{D}_t - \mathbf{G}_t \widetilde{\mathbf{B}}_t \widetilde{\mathbf{Q}}_t \mathbf{H}_t\|_F^2
$$

$$
\text{s.t. } \mathbf{G}_t^\top \mathbf{G}_t = \mathbf{I}_m, \widetilde{\mathbf{B}}_t \widetilde{\mathbf{B}}_t^\top = \mathbf{I}_m, \widetilde{\mathbf{Q}}_t \geq 0, \widetilde{\mathbf{Q}}_t^\top \mathbf{1}_m = \mathbf{1}_{\widetilde{n}_t},
$$

(54)

which can be seen as a special case of (6) when both $\lambda$ and $\beta$ take 0, and only three variables need to be optimized.

When updating $\mathbf{G}_t$, with given $\widetilde{\mathbf{B}}_t$ and $\widetilde{\mathbf{Q}}_t$, the problem (54) is equivalent to (19). Therefore, we can update $\mathbf{G}_t$ by solving (21) as $t \geq 2$ does. When updating the variable $\widetilde{\mathbf{B}}_t$, with given $\mathbf{G}_t$ and $\widetilde{\mathbf{Q}}_t$, the problem (54) is equivalent to (23) with $\lambda$ taking 0. Therefore, we can update $\widetilde{\mathbf{B}}_t$ by solving (27) as previous $t \geq 2$ does with $\lambda$ taking 0. When updating $\widetilde{\mathbf{Q}}_t$, with given $\mathbf{G}_t$ and $\widetilde{\mathbf{B}}_t$. (54) is equivalent to (28) with $\beta$ taking 0. Consequently, we can update $\widetilde{\mathbf{Q}}_t$ via (45) with $\beta$ taking 0.

In summary, under the scenario $t = 1$, we can optimize $\mathbf{G}_t$ and $\widetilde{\mathbf{B}}_t$ as well as $\widetilde{\mathbf{Q}}_t$ using the updating paradigm in $t \geq 2$ with $\lambda$ and $\beta$ taking 0.

# B Complexity Analysis

## B.1 Computational Complexity

The computational expenditure is mainly composed of updating $\mathbf{G}_t$, $\widetilde{\mathbf{B}}_t$, $\widetilde{\mathbf{Q}}_t$ and $\widehat{\mathbf{S}}_{t-1}$. When updating $\mathbf{G}_t$, it involves constructing $\mathbf{D}_t \mathbf{H}_t^\top \widetilde{\mathbf{Q}}_t^\top \widetilde{\mathbf{B}}_t^\top$ and performing SVD on it, which takes $\mathcal{O}(d_t \widetilde{n}_t m)$ and $\mathcal{O}(d_t m^2)$ respectively. $m$ is largely smaller than $\widetilde{n}_t$, and therefore updating $\mathbf{G}_t$ totally takes $\mathcal{O}(d_t \widetilde{n}_t m)$ computational cost. When updating $\widetilde{\mathbf{B}}_t$, it involves constructing $\mathbf{G}_t^\top \mathbf{D}_t \mathbf{H}_t^\top \widetilde{\mathbf{Q}}_t^\top + \lambda \widetilde{\mathbf{B}}_{t-1} \widehat{\mathbf{S}}_{t-1}$ and performing SVD on it, which takes $\mathcal{O}(m d_t \widetilde{n}_t)$ and $\mathcal{O}(m^3)$ respectively. Therefore, updating $\widetilde{\mathbf{B}}_t$ totally takes $\mathcal{O}(m d_t \widetilde{n}_t)$. When updating $\widetilde{\mathbf{Q}}_t$, owing to the closed-form solution, it only needs to construct $\widetilde{\mathbf{B}}_t^\top \mathbf{G}_t^\top \mathbf{D}_t \mathbf{H}_t^\top + \beta \widehat{\mathbf{S}}_{t-1}^\top \widetilde{\mathbf{Q}}_{t-1} \widetilde{\mathbf{E}}_{t-1}^\top$, which takes $\mathcal{O}(m d_t \widetilde{n}_t + m^2 \widetilde{n}_t)$. When updating $\widehat{\mathbf{S}}_{t-1}$, it needs to construct $\lambda \widetilde{\mathbf{B}}_t^\top \widetilde{\mathbf{B}}_{t-1} + \beta \widetilde{\mathbf{Q}}_t \widetilde{\mathbf{E}}_{t-1} \widetilde{\mathbf{Q}}_{t-1}^\top$, taking $\mathcal{O}(m^3 + m^2 \widetilde{n}_{t-1})$. Based on the above analysis, accordingly, the overall computing cost is $\mathcal{O}(d_t \widetilde{n}_t m + m^2 \widetilde{n}_t + m^2 \widetilde{n}_{t-1} + m^3)$. As time goes on, both $\widetilde{n}_t$ and $\widetilde{n}_{t-1}$ approach to $n$. Moreover, the data dimension $d_t$ is not relevant to the number of samples $n$. So, the computational complexity of proposed algorithm is $\mathcal{O}(n)$, i.e, linear with respect to the number of samples $n$.

## B.2 Space Complexity

The space expenditure is mainly composed of the memory parts required for optimizing $\mathbf{G}_t$, $\widetilde{\mathbf{B}}_t$, $\widetilde{\mathbf{Q}}_t$ and $\widehat{\mathbf{S}}_{t-1}$. During optimizing $\mathbf{G}_t$, due to $\mathbf{D}_t \in \mathbb{R}^{d_t \times n_t}$, $\widetilde{\mathbf{H}}_t \in \mathbb{R}^{\widetilde{n}_t \times n_t}$, $\widetilde{\mathbf{Q}}_t \in \mathbb{R}^{m \times \widetilde{n}_t}$ and $\widetilde{\mathbf{B}}_t \in \mathbb{R}^{m \times m}$, it will require $\mathcal{O}(d_t \widetilde{n}_t + d_t m)$ memory cost to store $\mathbf{D}_t \mathbf{H}_t^\top \widetilde{\mathbf{Q}}_t^\top \widetilde{\mathbf{Q}}_t$. Combined with $\mathbf{G}_t \in \mathbb{R}^{d_t \times m}$ and the fact that $m$ is largely smaller than $\widetilde{n}_t$, therefore, optimizing $\mathbf{G}_t$ needs $\mathcal{O}(d_t \widetilde{n}_t)$ memory overhead. During optimizing $\widetilde{\mathbf{B}}_t$, due to $\widetilde{\mathbf{B}}_{t-1} \in \mathbb{R}^{m \times m}$ and $\widehat{\mathbf{S}}_{t-1} \in \mathbb{R}^{m \times m}$, it will require $\mathcal{O}(d_t \widetilde{n}_t + m \widetilde{n}_t + m^2)$ cost to store $\mathbf{G}_t^\top \mathbf{D}_t \mathbf{H}_t^\top \widetilde{\mathbf{Q}}_t^\top + \lambda \widetilde{\mathbf{B}}_{t-1} \widehat{\mathbf{S}}_{t-1}$. Therefore, optimizing $\widetilde{\mathbf{B}}_t$ takes $\mathcal{O}(d_t \widetilde{n}_t)$ memory overhead. During optimizing $\widetilde{\mathbf{Q}}_t$, it will require $\mathcal{O}(d_t \widetilde{n}_t + m d_t + m \widetilde{n}_t)$ cost to store $\widetilde{\mathbf{B}}_t^\top \mathbf{G}_t^\top \mathbf{D}_t \mathbf{H}_t^\top + \beta \widehat{\mathbf{S}}_{t-1}^\top \widetilde{\mathbf{Q}}_{t-1} \widetilde{\mathbf{E}}_{t-1}^\top$. Accordingly, optimizing $\widetilde{\mathbf{Q}}_t$ takes $\mathcal{O}(d_t \widetilde{n}_t)$ memory overhead. During optimizing $\widehat{\mathbf{S}}_{t-1}$, it will require $\mathcal{O}(m^2 + m \widetilde{n}_{t-1})$ cost to store $\lambda \widetilde{\mathbf{B}}_t^\top \widetilde{\mathbf{B}}_{t-1} + \beta \widetilde{\mathbf{Q}}_t \widetilde{\mathbf{E}}_{t-1} \widetilde{\mathbf{Q}}_{t-1}^\top$. Therefore, optimizing $\widehat{\mathbf{S}}_{t-1}$ takes $\mathcal{O}(m \widetilde{n}_{t-1})$ memory overhead. Additionally, $\widetilde{n}_{t-1}$ is less than or equal to $\widetilde{n}_t$. Based on the above analysis, consequently, we have that the overall memory overhead is $\mathcal{O}(d_t \widetilde{n}_t)$. In conjunction with the fact that the data dimension $d_t$ is irrelevant to the number of samples, we can obtain that the space complexity of proposed algorithm is $\mathcal{O}(n)$.

# C Additional Related Work

To effectively cluster multi-view data, Cui *et al.* [6] introduce adaptive embedding learning to construct low-rank graph affinities, and impute the missing components by exploiting the latent cross-view correlations to generate complete similarity representations. Fu *et al.* [11] establish bipartite graphs through a generative modeling approach to capture potential global structure distributions, and employ a dynamic weighted fusion to mitigate the adverse effects of data incompleteness. Rather than the full graph, Li *et al.* [26] allocate dedicated sub-graphs to individual views to alleviate the biased error induced by incompleteness, and utilize the local graph structure refined by tensor means to circumvent explicit feature recovery. Gu *et al.* [13] build anchor graphs via dictionary learning to improve the robustness against missing data, and concurrently extract local representation and high-order correlations by Gaussian error rank minimization with Laplacian manifold regularization. Zhao *et al.* [93] develop a low-rank representation framework based on intrinsic graphs to infer missing samples, and establish inter-graph correlations through between-view structural consistency to formulate a robust consensus representation. Li *et al.* [20] employ the view-wise prototype graph learned via a dual-stream framework to model similarity, and conduct data recovery by simultaneously investigating instance-level commonality and view versatility to preserve essential cluster structures. In contrast to separate representation learning, Wen *et al.* [68] formulate unified representation by combining spectral clustering and graph learning to achieve partial-view balance, and derive low-dimensional embedding using a co-regularizer to reveal inherent structure among data. He *et al.* [14] utilize an anchor-inferred graph to directly produce asymmetric intra-view similarity, and resolve information deficiency in incomplete data through a paired anchor bridging mechanism.

Different from graph technique for clustering tasks, Liu *et al.* [36] leverage the matrix factorization paradigm to mitigate the imbalance factor in incomplete learning, and apply a sparse regularizer to capture low-dimensional individual features and construct compact local embedding. Wen *et al.* [70] maintain local geometric similarities via a efficient-weighted factorization model to facilitate common representation learning, and leverage adaptive view importance assignment to alleviate the bias caused by unbalanced incomplete views. Li *et al.* [22] explicitly model the between-view relationships through orthogonal factorization to explore the within-view spatial organization, and employ a tensor regularization as rank approximation to capture complementary information. In comparison to individual factorization mechanism, Chen *et al.* [3] unify coefficient expression and base learning via concept factorization to model the cluster-sample relations, and build explicit view connections using the projecting learning to preserve semantic correlations. Li *et al.* [30] jointly factorize multiple affinities in a tensor manner to maintain high-order geometrical characteristics, and impose a hyper-Laplacian constraint on neighbor samples to complement the consistency for incomplete view completion. Wen *et al.* [67] model the characteristic of each view through feature-aware factorization to mitigate the adverse effects caused by redundancy and noise interference, and boost the feature distinctiveness via an adaptive weighting scheme to produce structured unified representations. Unlike single-layer factorization, Zhang *et al.* [88] utilize multi-stage factorization paradigm to progressively explore partition-level representations, and aggregate all view information through late fusion to generate common grouping results. Li *et al.* [23] decrease the between-view errors by orthogonal and low-rank factorization to construct indicators without post-processing, and employ tensor norm regularization and soft labels jointly to exploit data complementary.

Unlike graph and matrix factorization means, subspace scheme is recently introduced into the grouping of multi-view data. Deng *et al.* [7] perform learning in a low-dimensional subspace to effectively integrate multi-view information, and utilize an ensemble of projection mappings to process new samples with maintained balance. Wang *et al.* [64] map original data through convolutional encoders into a subspace to construct potential representations, and learn soft labels via a self-expressive layer to reveal hierarchical feature structures. Huang *et al.* [17] collaboratively examine multilevel diversity patterns in random subspaces to form metric pairs, and adopt an entropy-driven mechanism to ensure comprehensive preservation about cluster wise variations in ensemble structures. Kang *et al.* [18] derive partition-level features across multiple subspaces to mitigate intra-view noise and inter-feature inconsistencies, and adaptively assign higher weights to partitions stronger consensus clustering agreement. Lv *et al.* [44] weight the reconstruction loss via pairwise similarity measures to preserve local structural relationships, and leverage pseudo-labeling to iteratively refine uncertain knowledge derived from subspace learning. Contrary to plain subspace, Qin *et al.* [48] construct the subspace endowed with both sparse and low-rank properties to mitigate errors in projectors, and take advantages of the block diagonal structure to amplify the

feature discriminability. Chen *et al.* [4] model data distribution through essential basis selection to derive cluster partitions in orthogonal subspace, and employ a mutual enhancement mechanism to jointly optimize cluster indicators and data representations. Wang *et al.* [63] maximize cross-view correlations for consistent subspace learning by canonical analysis, and enforce sample-wise mutual exclusivity via an $\ell_{1,2}$ norm constraint to preserve underlying data structures.

Kernel strategy is also generally deemed as a effective means to handle multi-view data. Liu *et al.* [38] leverage cluster assignments to refine incomplete kernel recovery, and integrate the kernel imputation and clustering through alignment maximization to achieve superior data grouping performance. Zhang *et al.* [90] simultaneously address kernel imputation and space partitioning to reconstruct latent nonlinear view structures, and seamlessly generate low-rank tensor embedding and affinity matrices in an unified framework. Wang *et al.* [57] transform the concatenated view into a kernel space to formulate nonlinear view relationships, and employ a kernelized smoothness regularizer to explicitly maintain locality properties in the feature space. Orthogonal to equally treating kernels, Li *et al.* [25] explicitly model both diversity and complementarity among base kernels to effectively capture inter-kernel correlations, and incorporate a matrix-induced regularization to probabilistically favor selection of kernel pairs with optimal dissimilarity. Zhang *et al.* [89] jointly improve kernel completion and low-rank tensor learning through a co-optimization scheme, and integrate kernel techniques into an incomplete subspace to reliably uncover multi-view structures. Liang *et al.* [31] approximate the spectral decomposition corresponding to the convex combination of base kernels to generate low-dimensional out-of-sample embedding, and establish rigorous theoretical guarantees for its stability. Liu *et al.* [39] emphasize learned unified clustering matrix to fill incomplete base matrices rather than operating directly on kernel spaces, and regularize clustering matrix by introducing prior knowledge to improve the performance. Li *et al* [27] adaptively determine the dimension of unified partition in a potential embedding domain to overcome the limitations of fixed-partition dimension, and leverage the latent consensus structure to model distributions among base kernels. Another algorithms, for instance, [32, 61, 59, 15, 10, 87], are also well-researched.

## D  Symbol Summary

Table 4 provides a summary of the main mathematical symbols and their corresponding meanings utilized in this study.

Table 4: Symbol and Meaning

| Symbol | Meaning |
|---|---|
| $\mathbf{D}_t \in \mathbb{R}^{d_t \times n_t}$ | the data matrix on the $t$-th view; |
| $\mathbf{G}_t \in \mathbb{R}^{d_t \times m}$ | the guidance matrix on the $t$-th view; |
| $\widetilde{\mathbf{B}}_t \in \mathbb{R}^{m \times m}$ | the unified anchor matrix on all $t$ views; |
| $\widetilde{\mathbf{Q}}_t \in \mathbb{R}^{m \times \widetilde{n}_t}$ | the shared bipartite graph on all $t$ views; |
| $\mathbf{H}_t \in \mathbb{R}^{\widetilde{n}_t \times n_t}$ | the indicator matrix on the $t$-th view; |
| $\widehat{\mathbf{S}}_{t-1} \in \mathbb{R}^{m \times m}$ | the swapping matrix about $\widetilde{\mathbf{B}}_{t-1}$ (and $\widetilde{\mathbf{Q}}_{t-1}$) |
| $\widetilde{\mathbf{E}}_{t-1} \in \mathbb{R}^{\widetilde{n}_t \times \widetilde{n}_{t-1}}$ | the indicator matrix on all $t-1$ views; |
| $\mathbf{I}_m \in \mathbb{R}^{m \times m}$ | the identity matrix with the size of $m \times m$; |
| $\mathbf{1}_m \in \mathbb{R}^{m \times 1}$ | the vector with the size of $m \times 1$; |
| $\mathbf{w}_t \in \mathbb{R}^{n_t \times 1}$ | the index vector on the $t$-th view; |
| $\widetilde{\mathbf{w}}_t \in \mathbb{R}^{\widetilde{n}_t \times 1}$ | the index vector union on all $t$ views; |
| $n_t$ | the number of samples observed on the $t$-th view; |
| $\widetilde{n}_t$ | the number of samples observed on all $t$ views; |
| $n$ | the number of samples; |
| $m$ | the number of anchors; |
| $d_t$ | the data dimension on the $t$-th view; |
| $\lambda$ and $\beta$ | the hyper-parameters; |

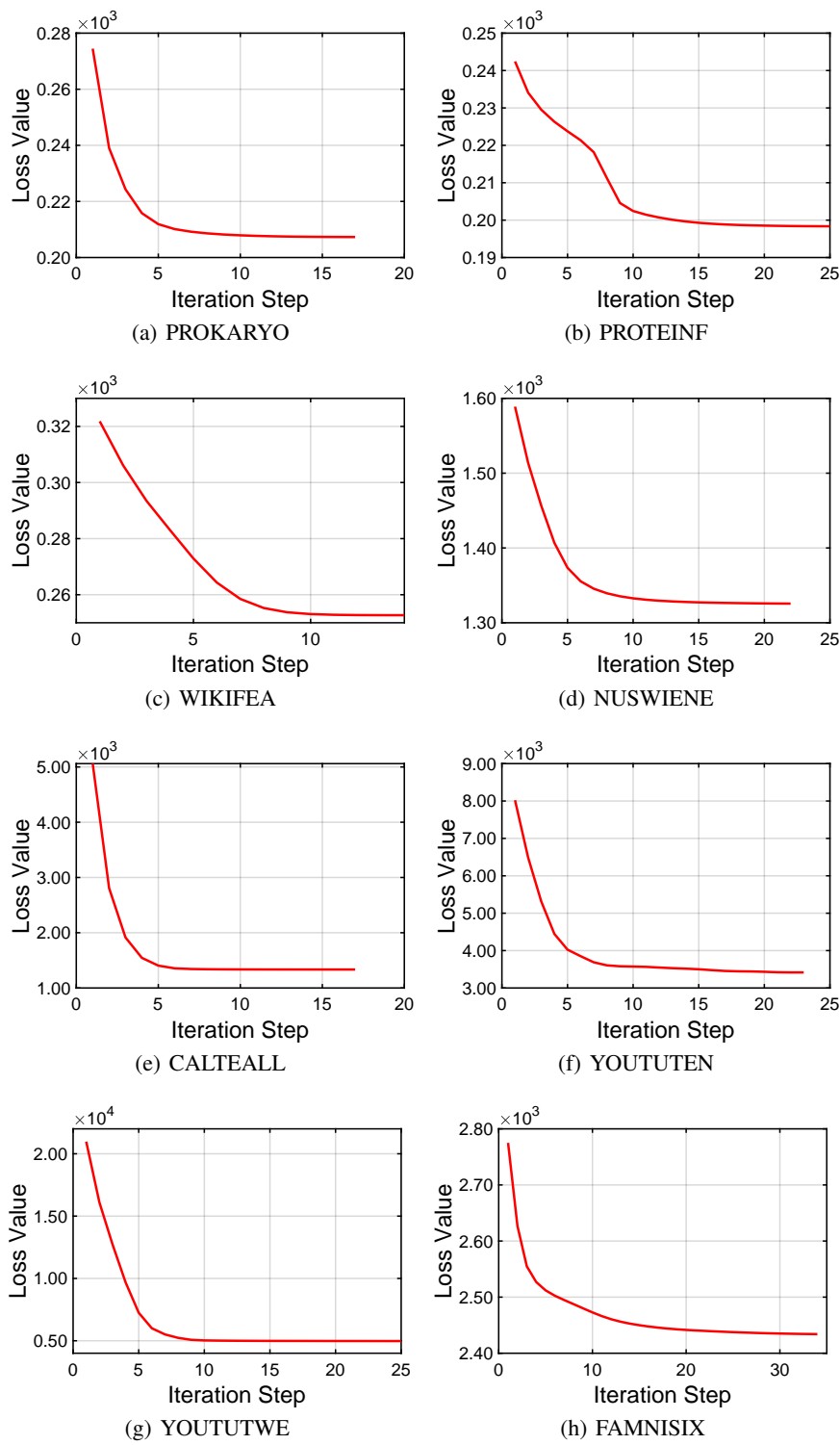

Figure 4: Evolution of the Loss Value with Respect to Iteration Step

# E   Experimental Setting

In experiments, we initialize the variables $\mathbf{G}_1$ and $\widehat{\mathbf{S}}_1$ using a random orthogonal matrix and an identify matrix, respectively. For $\widetilde{\mathbf{B}}_1$, we firstly construct a unified space by minimizing the reconstruction error, and then employ $k$-means within this space to generate initial anchors serving as the assignment of $\widetilde{\mathbf{B}}_1$. For $\widetilde{\mathbf{Q}}_1$, we utilize one-hot vectors to randomly assign its columns, guaranteeing that the sum of its each column equals to 1. For hyper-parameters $\lambda$ and $\beta$, we fine-tune them in $[10^{-1}, 10^0, 10^1, 10^2, 10^3]$ respectively. For the stopping error $\epsilon$, we set it as $10^{-4}$. Then, we execute 20 times and summarize the average clustering results.

# F   Description of Benchmark Methods

This section provides brief description of relevant benchmark methods.

**GSRMC**: It utilizes precomputed sub-graphs on each view instead of feature recovering to balance boundary samples, and separates the graph structure by means of nuclear norm to reduce biased error.

**FLSD**: It develops a graph regularizer based on factorization mechanism to maintain local geometries, and utilizes semantic consistency to homeoregulate the discriminativeness of incomplete observations.

**MVTSC**: It combines manifold space and feature space to restore missing parts, and explores inter-view information by tensor low-rank and graph consistency to formulate expressive representations.

**MVCBG**: It constructs a potential space via data compression to decreases the noisy impact, and utilizes view information complementarity based on random walk principle to establish the affinity.

**AGCMC**: It integrates consensus learning and graph completion to exploit hidden features of missing instances, and employs scale vectors to decrease the imbalance influence induced by incompleteness.

**NGSPL**: It builds up the relations between neighbor group and single sample pair to formulate representations, and designs a structure constraint based on neighbors to improve the graph quality.

**PIMVC**: It extracts sample features in a subspace instead of original space to balance information among different views, and devises a scatter matrices based graph learning to capture data structures.

**TCIMC**: It adopts a tensor norm mechanism to mine inter-view structures and complementary features, and utilizes graph connectivity constraint to alleviate post-processing grouping operations.

**TMBSD**: It introduces the membership learning by view consistency to maintain multiple spectral embeddings, and employs a tensor norm based regularizer to enhance the block-diagonal structure.

**PSMVC**: It constructs the prototype graph rather than pair-wise graph to exploit data similarity relations, and introduces a group of heterogeneous projections to preserve consistent cluster structures.

**MKKMC**: It integrates clustering and imputation to get rid of the limitations of base kernel completeness, and encourages kernel matrices via joint learning to adaptively populate each other mutually.

**LRGMV**: It unifies inter-view and intra-view related information to recover latent relationships across views, and gives a low-rank regularizer to explicitly extract global structures between views.

Table 5: Space Usage Comparison

| Dataset | GSRMC | FLSD | MVTSC | MVCBG | AGCMC | NGSPL | PIMVC | TCIMC | TMBSD | PSMVC | MKKMC | LRGMV | Ours |
|---|---|---|---|---|---|---|---|---|---|---|---|---|---|
| PROKARYO | 0.12 | 0.05 | 0.08 | 0.04 | 0.10 | 0.07 | 0.05 | 0.17 | 0.08 | 0.03 | 0.05 | 0.04 | **0.03** |
| PROTEINF | 0.83 | 0.11 | 0.53 | 0.06 | 0.20 | 0.25 | 0.06 | 0.90 | 0.57 | 0.06 | 0.24 | 0.29 | **0.03** |
| WIKIFEA | 2.59 | 0.48 | 1.57 | 0.07 | 1.13 | 1.95 | 0.40 | 2.70 | 1.77 | 0.08 | 1.22 | 1.15 | **0.05** |
| NUSWIENE | 12.31 | 1.79 | 7.86 | 0.16 | 3.31 | 5.82 | 7.79 | 9.80 | 8.45 | 0.13 | 3.53 | 4.51 | **0.11** |
| CALTEALL | 60.92 | 8.96 | 36.16 | 0.79 | 15.67 | 29.10 | / | / | 42.47 | 0.71 | 17.71 | 22.54 | **0.64** |
| YOUTUTEN | / | 73.14 | / | 4.29 | / | / | 68.91 | / | / | 3.77 | / | / | **2.84** |
| YOUTUTWE | / | / | / | 6.52 | / | / | / | / | / | 5.98 | / | / | **4.44** |
| FAMNISIX | / | / | / | 37.32 | / | / | / | / | / | 43.07 | / | / | **34.41** |

# G   Space Usage

In addition to linear time complexity, the proposed BSTM is also with linear space complexity. Section 6.2 has demonstrated the execution efficiency. Here, to illustrate its space-friendly characteristic,

we count the memory overhead, as shown in Table 5 where all results are expressed in GB units. One can see that we consistently receive less memory overhead, which mainly benefits from the fact that we circumvent view-data recomputing and successfully leverage only one bipartite graph with small size, rather than multiple self-expression affinities with full size, to characterize overall similarity.

## H    Convergence

To illustrate the convergence of our BSTM, we visualize the loss values, as suggested in Fig. 4. One can observe that the loss value is consistently decreasing and gradually reaches stability within forty iterations, which indicates that the presented BSTM is convergent.

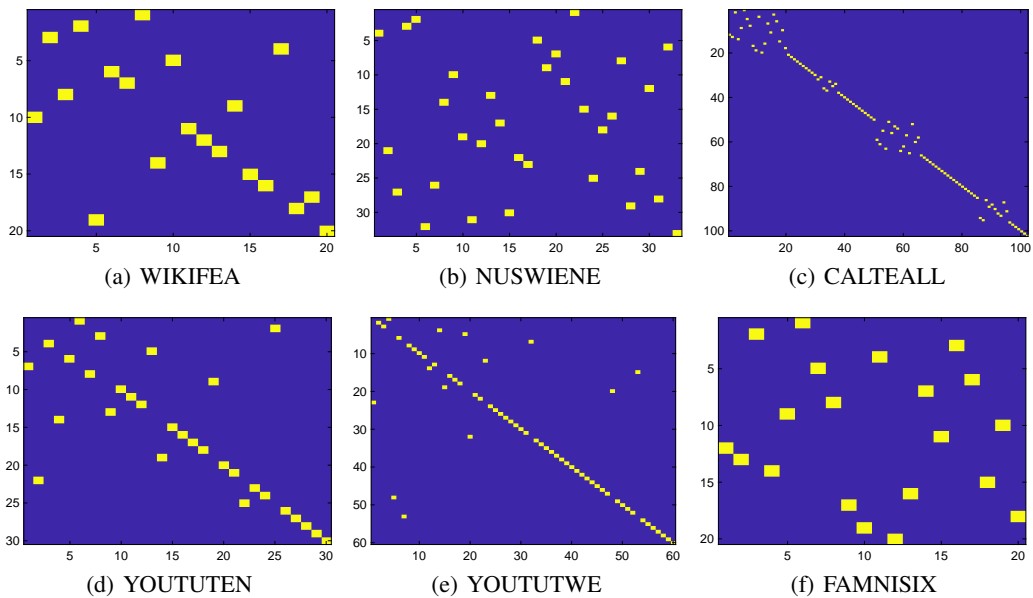

(a) WIKIFEA            (b) NUSWIENE            (c) CALTEALL

(d) YOUTUTEN            (e) YOUTUTWE            (f) FAMNISIX

Figure 5: Visualization of $\mathbf{S}_{t-1}$ Learned on WIKIFEA $\sim$ FAMNISIX

Table 6: Ablation Results about Twin-memory and Bit-swapping on WIKIFEA $\sim$ FAMNISIX

| | WIKIFEA | | | | | | | | | NUSWIENE | | | | | | | | |
|---|---|---|---|---|---|---|---|---|---|---|---|---|---|---|---|---|---|---|
| Abla. | 0.1 | | | 0.4 | | | 0.7 | | | 0.1 | | | 0.4 | | | 0.7 | | |
| | ACC | PUR | FSC | ACC | PUR | FSC | ACC | PUR | FSC | ACC | PUR | FSC | ACC | PUR | FSC | ACC | PUR | FSC |
| Ab-B | 50.98 | 57.08 | 44.74 | 49.07 | 54.22 | 38.13 | 41.13 | 45.65 | 26.05 | 10.74 | 33.34 | 7.32 | 12.10 | 30.53 | 9.67 | 11.79 | 30.96 | 9.33 |
| Ab-Q | 52.52 | 59.17 | 45.12 | 48.22 | 53.22 | 37.96 | 40.62 | 44.10 | 25.19 | 10.10 | 33.28 | 7.11 | 12.37 | 30.53 | 9.74 | 15.19 | 30.28 | 14.92 |
| Ab-bs | 50.49 | 57.80 | 44.52 | 47.81 | 53.00 | 37.88 | 39.60 | 43.72 | 24.49 | 9.86 | 32.84 | 6.95 | 12.14 | 30.90 | 8.99 |
| Ours | **54.22** | **59.74** | **47.23** | **50.17** | **54.50** | **39.21** | **41.70** | **46.41** | **27.03** | **10.99** | **33.57** | **7.59** | **12.48** | **33.52** | **9.87** | **15.41** | **33.15** | **15.10** |
| | CALTEALL | | | | | | | | | YOUTUTEN | | | | | | | | |
| Ab-B | 21.97 | 32.11 | 15.23 | 18.38 | 31.00 | 12.56 | 16.12 | 26.93 | 8.58 | 70.97 | 76.93 | 62.67 | 70.97 | 77.36 | 63.42 | 71.31 | 76.36 | 62.86 |
| Ab-Q | 23.46 | 32.46 | 16.56 | 21.75 | 32.46 | 11.24 | 19.38 | 28.79 | 7.51 | 71.47 | 77.12 | 62.95 | 71.76 | 77.87 | 62.56 | 71.43 | 76.58 | 61.98 |
| Ab-bs | 21.78 | 30.72 | 14.62 | 15.59 | 26.57 | 10.35 | 16.96 | 27.45 | 9.21 | 71.13 | 76.67 | 61.98 | 70.67 | 78.23 | 63.26 | 70.95 | 75.89 | 62.31 |
| Ours | **25.93** | **35.75** | **17.28** | **23.24** | **33.51** | **14.27** | **20.27** | **29.34** | **9.57** | **73.23** | **78.94** | **67.54** | **74.14** | **79.56** | **67.43** | **72.21** | **78.51** | **64.77** |
| | YOUTUTWE | | | | | | | | | FAMNISIX | | | | | | | | |
| Ab-B | 67.21 | 68.32 | 60.47 | 68.49 | 76.21 | 59.84 | 66.76 | 75.05 | 59.23 | 51.23 | 51.64 | 42.79 | 50.94 | 51.43 | 42.14 | 51.52 | 52.02 | 41.28 |
| Ab-Q | 67.47 | 70.43 | 62.98 | 70.87 | 76.63 | 60.62 | 67.23 | 75.87 | 60.18 | 51.08 | 51.73 | 44.86 | 51.78 | 51.76 | 40.78 | 51.23 | 52.24 | 41.17 |
| Ab-bs | 66.97 | 68.85 | 60.42 | 68.93 | 75.47 | 59.78 | 66.13 | 76.03 | 59.78 | 50.89 | 52.18 | 43.56 | 52.02 | 52.16 | 43.37 | 50.97 | 52.21 | 42.12 |
| Ours | **71.23** | **76.86** | **64.74** | **73.86** | **79.51** | **62.06** | **68.62** | **76.91** | **62.90** | **57.97** | **58.35** | **47.84** | **56.34** | **57.27** | **46.89** | **54.52** | **58.43** | **45.93** |

## I    Additional Ablation Results

Table 6 summarizes the ablation results on dataset WIKIFEA $\sim$ FAMNISIX about the effectiveness of twin-memory and bit-swapping strategies. One can see that we receive consistently superior results than other counterparts. Combined with Table 6 and Table 3, we can obtain that the devised twin-memory strategy and bit-swapping strategy indeed bring performance enhancement.

Moreover, we also plot the learned bit-swapping $\widehat{\mathbf{S}}_{t-1}$, as shown in Fig. 5. Yellow denotes 1 while blue denotes 0. As seen, the produced $\widehat{\mathbf{S}}_{t-1}$ conforms to permutation characteristics, constitutes a non-identity transformation matrix, and accordingly reorders anchor topology and graph structure.

Next, we provide the ablation results in which $\widehat{\mathbf{S}}_{t-1}$ only acts on $\widetilde{\mathbf{B}}_t$, $\widehat{\mathbf{S}}_{t-1}$ only acts on $\widetilde{\mathbf{Q}}_t$, and our bit-swapping transformation is replaced by the value-altering mapping.

## I.1 Ablation for $\widehat{\mathbf{S}}_{t-1}$ Only Acting on $\widetilde{\mathbf{B}}_t$

In our model, the bit-swapping transformation $\widehat{\mathbf{S}}_{t-1}$ is simultaneously applied to both $\widetilde{\mathbf{B}}_t$ and $\widetilde{\mathbf{Q}}_t$ to rearrange anchor topology and graph structure. Accordingly, we organize comparative experiments to display the results in which $\widehat{\mathbf{S}}_{t-1}$ only acts on one of $\widetilde{\mathbf{B}}_t$ and $\widetilde{\mathbf{Q}}_t$.

Table 7 presents the ablation results for $\widehat{\mathbf{S}}_{t-1}$ only acting on $\widetilde{\mathbf{B}}_t$ (SonB). As seen, ours is the better in most cases, which means that applying $\widehat{\mathbf{S}}_{t-1}$ on $\widetilde{\mathbf{Q}}_t$ is effective and can promote clustering performance.

Table 7: Ablation Results for $\widehat{\mathbf{S}}_{t-1}$ Only Acting on $\widetilde{\mathbf{B}}_t$

| Abla. | PROKARYO | | | | | | | | | PROTEINF | | | | | | | | |
| | 0.1 | | | 0.4 | | | 0.7 | | | 0.1 | | | 0.4 | | | 0.7 | | |
| | ACC | PUR | FSC | ACC | PUR | FSC | ACC | PUR | FSC | ACC | PUR | FSC | ACC | PUR | FSC | ACC | PUR | FSC |
| SonB | 51.54 | 72.41 | 46.39 | 43.99 | 56.81 | 42.67 | 44.61 | 56.81 | 41.62 | 35.68 | 40.97 | 21.31 | 29.64 | 35.37 | 16.14 | 26.57 | 31.17 | 12.25 |
| Ours | **74.95** | **82.21** | **64.79** | **72.78** | **79.13** | **62.82** | **52.09** | **74.59** | **49.59** | **36.26** | **41.70** | **21.81** | **33.57** | **37.89** | **18.41** | **30.07** | **35.07** | **14.41** |
| | WIKIFEA | | | | | | | | | NUSWIENE | | | | | | | | |
| SonB | 53.24 | 59.71 | 45.92 | 48.94 | 53.13 | 38.01 | 39.70 | 44.80 | 25.59 | 10.29 | 33.13 | 7.05 | 12.21 | 30.60 | 9.66 | 11.53 | 30.90 | 8.99 |
| Ours | **54.22** | **59.74** | **47.23** | **50.17** | **54.50** | **39.21** | **41.70** | **46.41** | **27.03** | **10.99** | **33.57** | **7.59** | **12.48** | **33.52** | **9.87** | **15.41** | **33.15** | **15.10** |
| | CALTEALL | | | | | | | | | YOUTUTEN | | | | | | | | |
| SonB | 22.97 | 33.14 | 16.85 | 15.59 | 26.57 | 10.35 | 16.96 | 27.45 | 9.14 | 72.52 | 78.50 | 62.75 | 70.55 | 77.30 | 60.71 | 71.45 | 76.58 | 62.22 |
| Ours | **25.93** | **35.75** | **17.28** | **23.24** | **33.51** | **14.27** | **20.27** | **29.34** | **9.57** | **73.23** | **78.94** | **67.54** | **74.14** | **78.94** | **67.43** | **72.21** | **78.51** | **64.77** |
| | YOUTUTWE | | | | | | | | | FAMNISIX | | | | | | | | |
| SonB | 68.79 | 72.41 | 61.65 | 68.49 | 76.63 | 59.62 | **68.76** | **77.05** | 59.18 | 51.57 | 52.18 | 40.14 | 52.09 | 52.36 | 40.37 | 52.52 | 53.11 | 40.28 |
| Ours | **71.23** | **76.86** | **64.74** | **73.86** | **79.51** | **62.06** | 68.62 | 76.91 | **62.90** | **57.97** | **58.35** | **47.84** | **56.34** | **57.27** | **46.89** | **54.52** | **58.43** | **45.93** |

## I.2 Ablation for $\widehat{\mathbf{S}}_{t-1}$ Only Acting on $\widetilde{\mathbf{Q}}_t$

Table 8 provides the ablation results for $\widehat{\mathbf{S}}_{t-1}$ only acting on $\widetilde{\mathbf{Q}}_t$ (SonQ). As observed, we receive better results, indicating that applying $\widehat{\mathbf{S}}_{t-1}$ on $\widetilde{\mathbf{B}}_t$ is beneficial for the clustering performance gain.

Table 8: Ablation Results for $\widehat{\mathbf{S}}_{t-1}$ Only Acting on $\widetilde{\mathbf{Q}}_t$

| Abla. | PROKARYO | | | | | | | | | PROTEINF | | | | | | | | |
| | 0.1 | | | 0.4 | | | 0.7 | | | 0.1 | | | 0.4 | | | 0.7 | | |
| | ACC | PUR | FSC | ACC | PUR | FSC | ACC | PUR | FSC | ACC | PUR | FSC | ACC | PUR | FSC | ACC | PUR | FSC |
| SonQ | 51.73 | 72.56 | 47.74 | 43.42 | 57.92 | 44.36 | 45.82 | 58.12 | 40.21 | 35.12 | 40.48 | 21.04 | 30.26 | 36.11 | 16.30 | 26.47 | 31.33 | 12.02 |
| Ours | **74.95** | **82.21** | **64.79** | **72.78** | **79.13** | **62.82** | **52.09** | **74.59** | **49.59** | **36.26** | **41.70** | **21.81** | **33.57** | **37.89** | **18.41** | **30.07** | **35.07** | **14.41** |
| | WIKIFEA | | | | | | | | | NUSWIENE | | | | | | | | |
| SonQ | 50.49 | 58.80 | 44.52 | 48.81 | 53.00 | 37.88 | 39.60 | 44.72 | 25.49 | 9.87 | 32.84 | 6.98 | 12.01 | 30.48 | 9.48 | 11.23 | 30.92 | 9.23 |
| Ours | **54.22** | **59.74** | **47.23** | **50.17** | **54.50** | **39.21** | **41.70** | **46.41** | **27.03** | **10.99** | **33.57** | **7.59** | **12.48** | **33.52** | **9.87** | **15.41** | **33.15** | **15.10** |
| | CALTEALL | | | | | | | | | YOUTUTEN | | | | | | | | |
| SonQ | 23.86 | 33.85 | 16.23 | 15.78 | 26.21 | 11.57 | 17.26 | 27.02 | **9.72** | 71.23 | 78.12 | 63.42 | 70.17 | 76.12 | 60.31 | 70.45 | 75.28 | 61.43 |
| Ours | **25.93** | **35.75** | **17.28** | **23.24** | **33.51** | **14.27** | **20.27** | **29.34** | 9.57 | **73.23** | **78.94** | **67.54** | **74.14** | **79.56** | **67.43** | **72.21** | **78.51** | **64.77** |
| | YOUTUTWE | | | | | | | | | FAMNISIX | | | | | | | | |
| SonQ | 67.92 | 73.26 | 62.13 | 69.67 | 77.22 | 58.23 | 67.83 | 76.86 | 58.18 | 53.67 | 51.43 | 41.37 | 43.69 | 53.67 | 42.36 | 51.47 | 55.63 | 41.36 |
| Ours | **71.23** | **76.86** | **64.74** | **73.86** | **79.51** | **62.06** | **68.62** | **76.91** | **62.90** | **57.97** | **58.35** | **47.84** | **56.34** | **57.27** | **46.89** | **54.52** | **58.43** | **45.93** |

## I.3 Ablation for Value-altering Mapping

Our bit-swapping transformation is with the ability to maintain numerical-consistency. That is, it only reorganizes the topology of anchors and the structure of graph while it does not alter their value. To illustrate its strengths, we utilize previous orthogonal permutation transformation (VaM) to replace it, and the ablation results are presented in Table 9. One can observe that we produce more

desirable clustering results than VaM in most cases, which means that our bit-swapping strategy is more worth-having.

Table 9: Ablation Results for Value-altering Mapping

| Abla. | PROKARYO | | | | | | | | | PROTEINF | | | | | | | | |
|---|---|---|---|---|---|---|---|---|---|---|---|---|---|---|---|---|---|---|
| | 0.1 | | | 0.4 | | | 0.7 | | | 0.1 | | | 0.4 | | | 0.7 | | |
| | ACC | PUR | FSC | ACC | PUR | FSC | ACC | PUR | FSC | ACC | PUR | FSC | ACC | PUR | FSC | ACC | PUR | FSC |
| VaM | 72.12 | 80.42 | 62.14 | 72.23 | 79.01 | 61.37 | **52.13** | 72.42 | 47.12 | 34.89 | 40.16 | 20.46 | 31.98 | 37.26 | **18.47** | 28.45 | 34.63 | 14.12 |
| Ours | **74.95** | **82.21** | **64.79** | **72.78** | **79.13** | **62.82** | 52.09 | **74.59** | **49.59** | **36.26** | **41.70** | **21.81** | **33.57** | **37.89** | 18.41 | **30.07** | **35.07** | **14.41** |
| | WIKIFEA | | | | | | | | | NUSWIENE | | | | | | | | |
| VaM | 53.63 | 57.84 | 46.72 | 50.02 | 53.12 | 38.63 | 40.86 | 46.21 | 26.73 | 10.63 | 33.12 | 7.13 | 12.12 | 32.34 | 9.62 | 14.63 | 32.56 | 14.42 |
| Ours | **54.22** | **59.74** | **47.23** | **50.17** | **54.50** | **39.21** | **41.70** | **46.41** | **27.03** | **10.99** | **33.57** | **7.59** | **12.48** | **33.52** | **9.87** | **15.41** | **33.15** | **15.10** |
| | CALTEALL | | | | | | | | | YOUTUTEN | | | | | | | | |
| VaM | 25.24 | 34.24 | 16.96 | 23.13 | 32.89 | 13.43 | 19.78 | 28.75 | 9.36 | 72.36 | 77.23 | 66.62 | 73.24 | 78.24 | 66.24 | 70.98 | 77.37 | **65.42** |
| Ours | **25.93** | **35.75** | **17.28** | **23.24** | **33.51** | **14.27** | **20.27** | **29.34** | **9.57** | **73.23** | **78.94** | **67.54** | **74.14** | **79.56** | **67.43** | **72.21** | **78.51** | 64.77 |
| | YOUTUTWE | | | | | | | | | FAMNISIX | | | | | | | | |
| VaM | 70.89 | 75.47 | 63.52 | 73.18 | 79.23 | **62.31** | 66.74 | 75.31 | 61.24 | **58.23** | 58.01 | 47.21 | 55.13 | 56.35 | 45.23 | 52.37 | 57.64 | 44.32 |
| Ours | **71.23** | **76.86** | **64.74** | **73.86** | **79.51** | 62.06 | **68.62** | **76.91** | **62.90** | 57.97 | **58.35** | **47.84** | **56.34** | **57.27** | **46.89** | **54.52** | **58.43** | **45.93** |

# J  Steadiness

In addition to the average value of clustering results, we count the standard deviation to demonstrate the steadiness of our model, as suggested in Table 10. According to these results, one can obtain that the presented model is relatively steady and can produce robust clustering results.

Table 10: Standard Deviation of Clustering Results (%)

| PROKARYO | | | | | | | | | PROTEINF | | | | | | | | |
|---|---|---|---|---|---|---|---|---|---|---|---|---|---|---|---|---|---|
| 0.1 | | | 0.4 | | | 0.7 | | | 0.1 | | | 0.4 | | | 0.7 | | |
| ACC | PUR | FSC | ACC | PUR | FSC | ACC | PUR | FSC | ACC | PUR | FSC | ACC | PUR | FSC | ACC | PUR | FSC |
| 0.00 | 0.00 | 0.00 | 0.00 | 0.00 | 0.00 | 0.00 | 0.00 | 0.00 | 0.87 | 0.89 | 0.63 | 0.76 | 0.58 | 0.63 | 0.75 | 0.78 | 0.51 |
| WIKIFEA | | | | | | | | | NUSWIENE | | | | | | | | |
| 0.01 | 0.01 | 0.01 | 0.00 | 0.00 | 0.00 | 0.00 | 0.00 | 0.00 | 0.37 | 0.19 | 0.21 | 0.16 | 0.15 | 0.07 | 0.19 | 0.15 | 0.10 |
| CALTEALL | | | | | | | | | YOUTUTEN | | | | | | | | |
| 0.50 | 0.15 | 0.42 | 0.35 | 0.25 | 1.10 | 0.45 | 0.36 | 0.37 | 2.34 | 1.42 | 0.00 | 2.00 | 1.25 | 0.00 | 2.47 | 1.50 | 0.00 |
| YOUTUTWE | | | | | | | | | FAMNISIX | | | | | | | | |
| 0.57 | 0.42 | 0.48 | 0.81 | 0.50 | 0.53 | 0.71 | 0.38 | 0.54 | 0.04 | 0.04 | 0.00 | 0.08 | 0.07 | 0.00 | 0.02 | 0.02 | 0.00 |

# K  Parameter Sensitivity

Parameters $\lambda$ and $\beta$ play a role in regulating $\mathcal{L}_2$ and $\mathcal{L}_3$ respectively to formulate favorable clustering representations. To explore the parameter sensitivity, we plot the results under multiple groups of $\lambda$s and $\beta$s, as shown in Fig. 6 and Fig. 7. One can observe that the clustering performance exhibits no significant fluctuations. Therefore, our model is relatively robust with respect to parameters $\lambda$ and $\beta$.

# L  Influence of Anchor Number

To investigate the influence of anchor number, we count the clustering performance under diverse anchor numbers, as illustrated in Fig. 8 where $m_1 \sim m_5$ equal to $1k \sim 5k$ respectively. $k$ is the number of clusters. As seen, the performance curve is relatively smooth with respect to anchor number, demonstrating that our proposed model is not significantly affected by the anchor number.

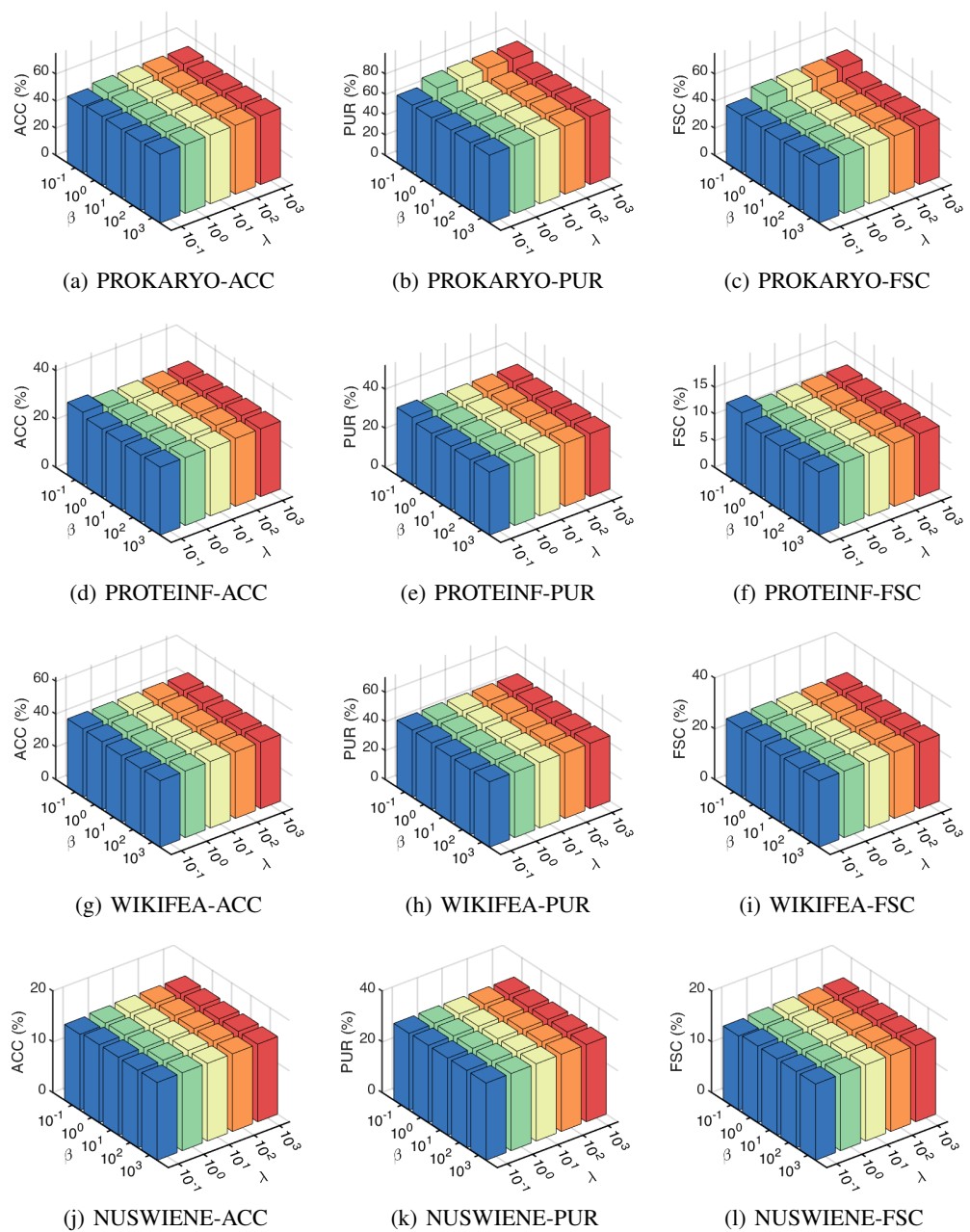

Figure 6: Parameter Sensitivity on PROKARYO ∼ NUSWIENE

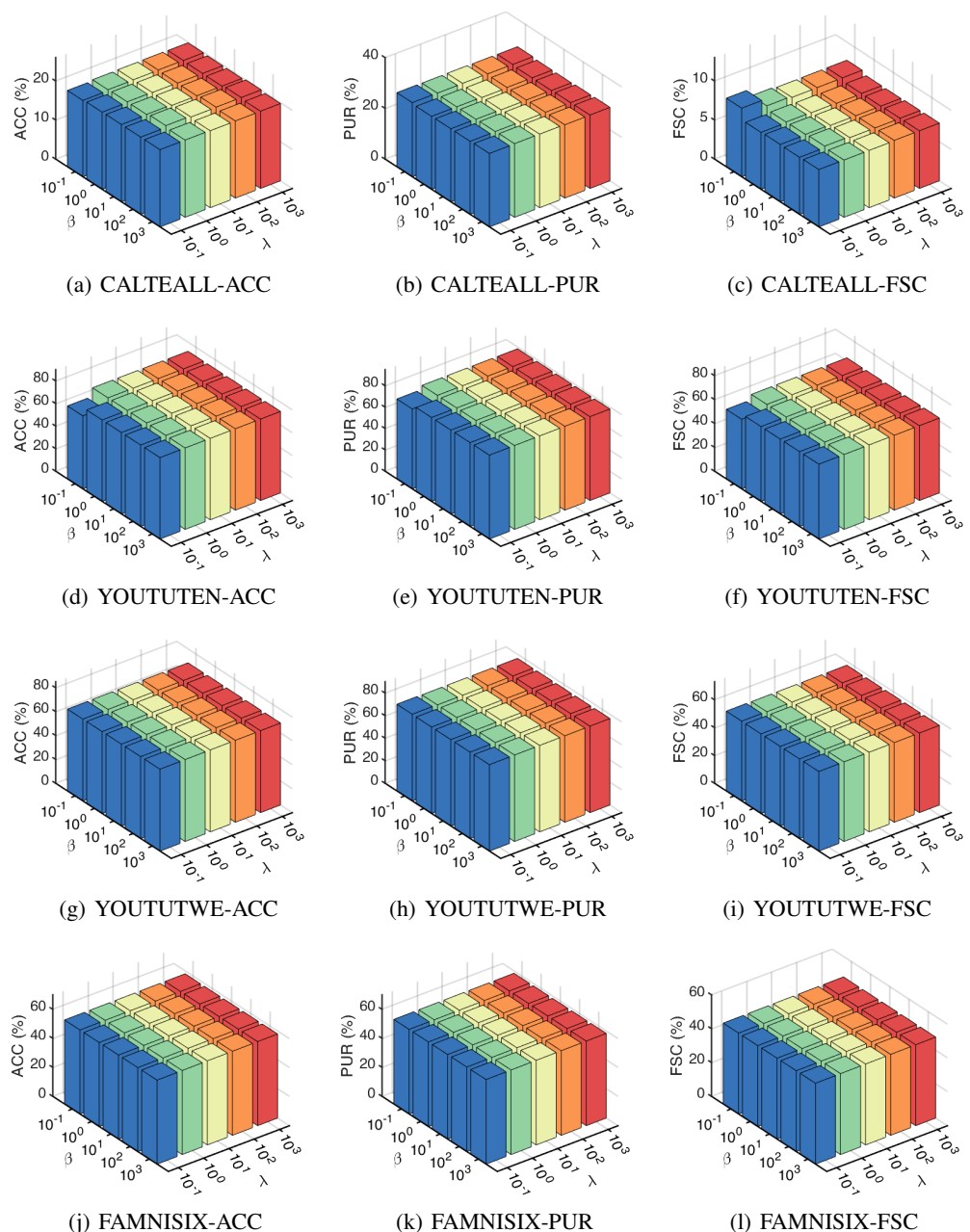

Figure 7: Parameter Sensitivity on CALTEALL ∼ FAMNISIX

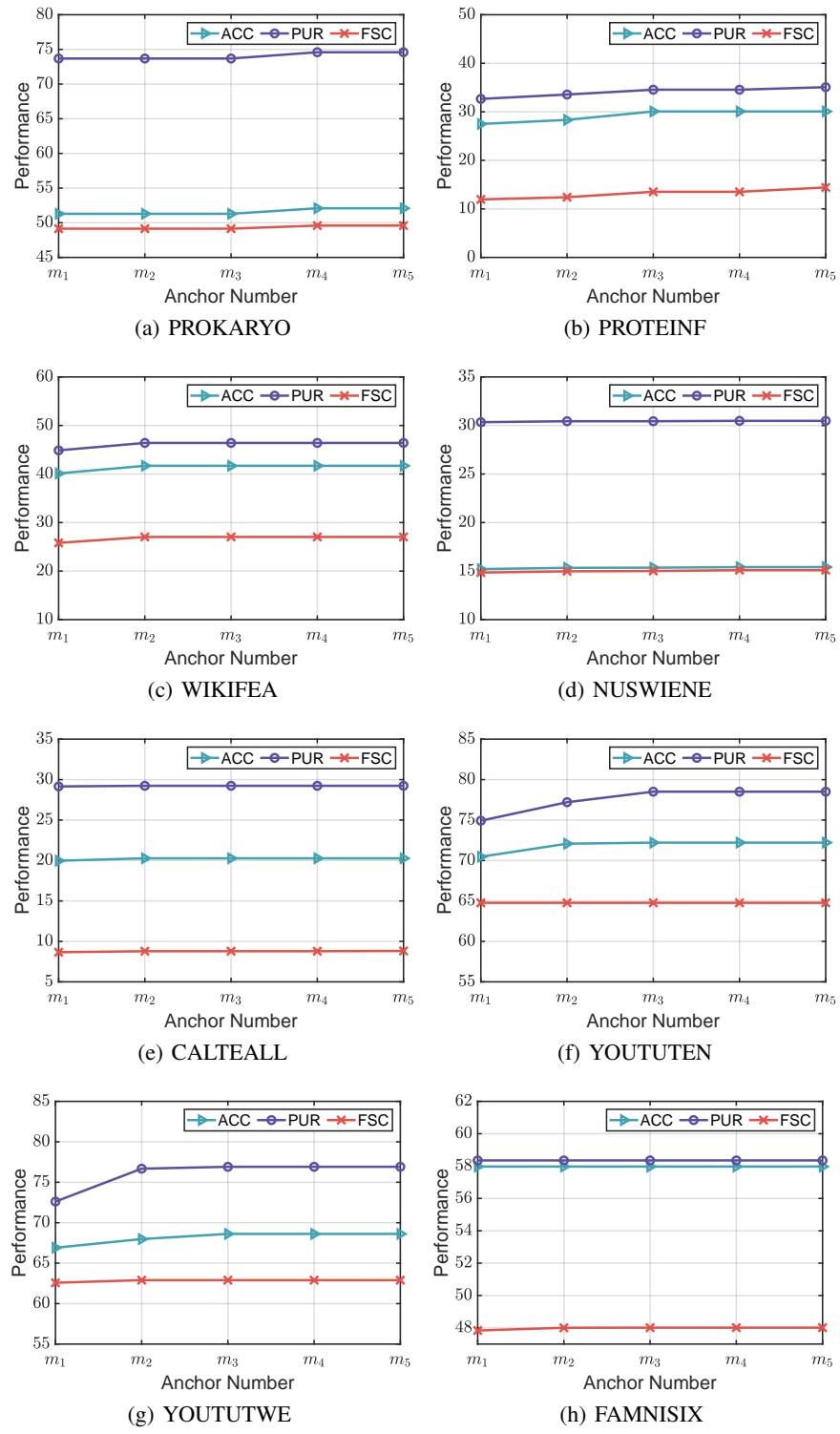

Figure 8: Clustering Performance under Different Anchor Numbers

