# OpenReview forum: "Bit-swapping Oriented Twin-memory Multi-view Clustering in Lifelong Incomplete Scenarios"
_NeurIPS.cc/2025/Conference — NeurIPS 2025 poster_

### Official Review · Reviewer_o1Vu · 2025-06-23

**Clarity:** 3
**Significance:** 4
**Originality:** 4
**Rating:** 4
**Confidence:** 4

**Summary:**

This paper addresses the challenges of incomplete multi-view clustering (MVC) by proposing the BSTM algorithm. It introduces a Twin-memory paradigm and a Bit-swapping strategy. The Twin-memory paradigm, consisting of a Unified Anchor Matrix (UAM) and a Shared Bipartite Graph (SBG), enables knowledge transfer to avoid redundant computation and fusion distortion. The Bit-swapping strategy reconstructs the anchor topology and graph structure based on binary operations. Experiments on 8 public datasets show that BSTM outperforms 12 baseline methods in most cases, especially in PUR and FSC metrics, and has linear complexity suitable for large-scale scenarios. However, there are still areas for improvement in theoretical analysis, experimental comparison, and expression.

**Questions:**

1.Theoretical Rigor: The theoretical explanation of the cooperation mechanism between the core components (UAM and SBG) is insufficient, and the assumption conditions and limitations of the model are not clearly stated.
2.In formula (6), the hyper-parameters  and  are merely stated to "balance the reconstruction error and the fusion term". It is suggested to supplement: ablation experiments investigating the influence of different (,) combinations on the convergence behavior should be included in the appendix.
3.This work lacks an analysis of the interactions between core components (UAM and SBG), particularly failing to investigate the performance degradation when UAM and SBG are used individually. It is recommended to conduct additional ablation experiments to quantitatively analyze the performance contributions of UAM and SBG in isolation, such as evaluating the metrics when only UAM or SBG is employed.
4.While the paper mentions a complexity analysis, it does not provide an explicit comparison of computational complexity with other methods. Please supplement a comparative analysis of time complexity and computational complexity between BSTM and state-of-the-art methods (e.g., MVTSC, AGCMC).

**Ethical Concerns:**

["NO or VERY MINOR ethics concerns only"]

**Final Justification:**

The authors have provided a well-prepared rebuttal that addresses most of the concerns raised in the initial review. Their clarifications and additional experimental results have improved the overall clarity and completeness of the paper. I appreciate the effort the authors have made in responding to the feedback. After considering the rebuttal, I have decided to maintain my original score.

**Limitations:**

Yes.

**Paper Formatting Concerns:**

No.

**Quality:**

3

**Strengths And Weaknesses:**

The proposed Twin-memory paradigm and Bit-swapping strategy are innovative. The UAM and SBG effectively address the issues of ignoring basic embedding information and mapping distortion in existing MVC methods, with a self-consistent logical framework. Adequate Experimentation: Comprehensive experiments on multiple datasets, comparing with a large number of baseline methods, and testing under different missing rates, which can well verify the performance of the algorithm. The linear complexity analysis also shows its potential for large-scale data. Clearly points out the problems in existing MVC methods and proposes corresponding solutions, which has certain guiding significance for the development of the field. However, theoretical derivation details, extreme scenario analysis, and clarity of expression still leave much to be desired. The supplementation of absent comparative experiments and mechanistic explanations is expected to further enhance its academic influence and universality in practical applications.

---

> ### Author Rebuttal · Authors · 2025-07-31
>
> **Q1:** Theoretical explanation on the cooperation mechanism between  UAM and SBG;  the assumption conditions; the limitations.
>
> **A1:** Thanks! UAM and SBG collectively transform knowledge via one common learnable framework. UAM aims to do propagation at embedding level while SBG does propagation at similarity level. They seamlessly integrate representations from historical views and newly-arrived data to facilitate clustering. Kindly note that they can cooperate with each other via $\mathcal{L}_1$, or via $\mathcal{L}_2$ and $\mathcal{L}_3$ using the swapping matrix, or via both.
>
>
> About the assumption conditions, we assume that views are independent of each other. This is rational since in practice, view data is usually collected from diverse sensors. Based on this assumption, we customize a guidance for each view to harmonize disparate dimensions  and thereby construct transferable anchors and bipartite graph.
>
>
> About the limitations, there are two hyper-parameters in our model, which could to some extent weaken its practicality. The paradigm that firstly formulates spectrum and then conducts partitioning may degenerate view data diversity. Besides, we at this point do not learn to adjust the representation contributions between newly-arrived data and historical views. As data collection progresses, inaccessible historical views will account for an increasing proportion compared to new view. Therefore, balancing them will further boost the model performance.
>
>
> **Q2:** The influence of different hyper-parameter combinations  on the convergence behavior.
>
> **A2:** We employ an alternating optimization strategy to minimize the formulated loss function that is lower-bounded by zero. Under different hyper-parameter combinations, the loss still can be lower-bounded by zero. Additionally, the solutions of all variables still can be obtained by Eqs.(8), (10), (15) and (18) respectively, even both of hyper-parameters as zero (as illustrated in Steps Under Initial View). Therefore, during updating each variable, it is monotonically decreasing. Combined with monotonicity and boundedness, consequently, our algorithm is theoretically convergent for different hyper-parameter combinations. We will add the convergence curves under diverse hyper-parameter combinations in the next version.
>
>
>
> **Q3:** 	The interactions between UAM and SBG; the performance contributions of UAM and SBG in isolation.
>
> **A3:**  UAM and SBG work within one integrated framework, and they negotiate with each other during the learning process. Owing to the joint-optimization, UAM and SBG establish a tightly-coupled relationship through bidirectional collaborative mechanisms. Specially, UAM guides SBG in capturing global clustering structures, while SBG refines UAM through its connection patterns to eliminate view-specific biases. This synergistic interaction enhances clustering robustness, particularly when handling incomplete views, where the memory function of UAM and the topological elasticity of SBG effectively preserve clustering performance.
>
>
>
> About the performance contributions, the following table gives the ablation results. As seen, UAM and SBG collaboratively improve the clustering performance.
>
>
> |  | PROKARYO |  |  |  |  |  |  |  |  |
> |:---:|:---:|:---:|:---:|:---:|:---:|:---:|:---:|:---:|:---:|
> |  | 0.1 |  |  | 0.4 |  |  | 0.7 |  |  |
> |  | ACC | PUR | FSC | ACC | PUR | FSC | ACC | PUR | FSC |
> | SBG | 51.54 | 72.60 | 46.61 | 43.99 | 56.81 | 42.66 | 44.61 | 56.98 | 41.62 |
> | UAM | 72.78 | 80.76 | 62.80 | 61.34 | 68.97 | 49.03 | 51.24 | 66.64 | 43.22 |
> | Both | **74.95** | **82.21** | **64.79** | **72.78** | **79.13** | **62.82** | **52.09** | **74.59** | **49.59** |
> |  | PROTEINF |  |  |  |  |  |  |  |  |
> | SBG | 36.49 | 41.92 | 21.67 | 31.99 | 36.53 | 17.08 | 28.65 | 33.17 | 12.51 |
> | UAM | 35.61 | 40.44 | 21.70 | 30.43 | 35.34 | 15.66 | 28.09 | 32.81 | 12.07 |
> | Both | **36.26** | **41.70** | **21.81** | **33.57** | **37.89** | **18.41** | **30.07** | **35.07** | **14.41** |
> |  | WIKIFEA |  |  |  |  |  |  |  |  |
> | SBG | 50.98 | 57.08 | 44.74 | 49.07 | 54.22 | 38.13 | 41.13 | 45.65 | 26.05 |
> | UAM | 52.52 | 59.17 | 45.12 | 48.22 | 53.22 | 37.96 | 40.62 | 44.10 | 25.19 |
> | Both | **54.22** | **59.74** | **47.23** | **50.17** | **54.50** | **39.21** | **41.70** | **46.41** | **27.03** |
> |  | NUSWIENE |  |  |  |  |  |  |  |  |
> | SBG | 10.74 | 33.34 | 7.32 | 12.10 | 30.53 | 9.67 | 11.79 | 30.96 | 9.33 |
> | UAM | 10.10 | 33.28 | 7.11 | 12.37 | 30.53 | 9.74 | 15.19 | 30.28 | 14.92 |
> | Both | **10.99** | **33.57** | **7.59** | **12.48** | **33.52** | **9.87** | **15.41** | **33.15** | **15.10** |
> |  | CALTEALL |  |  |  |  |  |  |  |  |
> | SBG | 21.97 | 32.11 | 15.23 | 18.38 | 31.00 | 12.56 | 16.12 | 26.93 | 8.58 |
> | UAM | 23.46 | 32.46 | 16.56 | 21.75 | 32.46 | 11.24 | 19.38 | 28.79 | 7.51 |
> | Both | **25.93** | **35.75** | **17.28** | **23.24** | **33.51** | **14.27** | **20.27** | **29.34** | **9.57** |
> |  | YOUTUTEN |  |  |  |  |  |  |  |  |
> | SBG | 70.97 | 76.93 | 62.67 | 70.97 | 77.36 | 63.42 | 71.31 | 76.36 | 62.86 |
> | UAM | 71.47 | 77.12 | 62.95 | 71.76 | 77.87 | 62.56 | 71.43 | 76.58 | 61.98 |
> | Both | **73.23** | **78.94** | **67.54** | **74.14** | **79.56** | **67.43** | **72.21** | **78.51** | **64.77** |
> |  | YOUTUTWE |  |  |  |  |  |  |  |  |
> | SBG | 67.21 | 68.32 | 60.47 | 68.49 | 76.21 | 59.84 | 66.76 | 75.05 | 59.23 |
> | UAM | 67.47 | 70.43 | 62.98 | 70.87 | 76.63 | 60.62 | 67.23 | 75.87 | 60.18 |
> | Both | **71.23** | **76.86** | **64.74** | **73.86** | **79.51** | **62.06** | **68.62** | **76.91** | **62.90** |
> |  | FAMNISIX |  |  |  |  |  |  |  |  |
> | SBG | 51.23 | 51.64 | 42.79 | 50.94 | 51.43 | 42.14 | 51.52 | 52.02 | 41.28 |
> | UAM | 51.08 | 51.73 | 44.86 | 51.78 | 51.76 | 40.78 | 51.23 | 52.24 | 41.17 |
> | Both | **57.97** | **58.35** | **47.84** | **56.34** | **57.27** | **46.89** | **54.52** | **58.43** | **45.93** |
>
>
>
> **Q4:** Complexity comparison between BSTM and comparison methods.
>
> **A4:** Thanks! We summarize the complexity comparison in the following table. Please check it. We will add this in the next version.
>
> |  | Time Complexity | Space Complexity |
> |:---:|:---:|:---:|
> | GSRMC | $\mathcal{O}(vn^2(v+\log(n)))$ | $\mathcal{O}(vn^2)$ |
> | FLSD | $\mathcal{O}(nd^2)$  | $\mathcal{O}(n^2v+dnk)$ |
> | MVTSC | $\mathcal{O}(vn^3+vn^2\log(n)+v^2n^2)$ | $\mathcal{O}(mn^2+mnk)$ |
> | MVCBG | $\mathcal{O}(ndk+ndm+dmk)$ | $\mathcal{O}(dk+mk+mn)$ |
> | AGCMC | $\mathcal{O}(n^2k)$ | $\mathcal{O}(vn^2+nk+v^2)$ |
> | NGSPL | $\mathcal{O}(n^2k)$ | $\mathcal{O}(mn^2+(m+1)nk)$ |
> | PIMVC | $\mathcal{O}(n^3+dk^2)$ | $\mathcal{O}(mn^2+dk^2)$ |
> | TCIMC | $\mathcal{O}(n^3+n^2v\log(n))$  | $\mathcal{O}(n^3+mn^2)$  |
> | TMBSD | $\mathcal{O}(nk^2+k^3+n^3)$ | $\mathcal{O}(mn^2+mnk+k^3)$  |
> | PSMVC | $\mathcal{O}(dm^2+nm^2+dmn)$ | $\mathcal{O}(nd+mn+m^2)$ |
> | MKKMC | $\mathcal{O}(n^3v)$ | $\mathcal{O}(n^2v+nkv)$ |
> | LRGMV | $\mathcal{O}(nk^2)$ | $\mathcal{O}(mn^2+nk)$ |
> | Ours | $\mathcal{O}(dnm + m^2n + m^3)$ | $\mathcal{O}(dn)$ |

---

> ### Comment · Reviewer_o1Vu · 2025-08-05
>
> Thanks for the detailed response. It has clarified most of my concerns. I would maintain my score.

---

### Official Review · Reviewer_e6Ai · 2025-06-29

**Clarity:** 3
**Significance:** 3
**Originality:** 3
**Rating:** 5
**Confidence:** 5

**Summary:**

This work proposes a twin-memory paradigm to overcome the loss of basis patterns during the process of propagating knowledge from historical views to new views, and a bit-swapping mechanism to correspond the topologies of anchors and graphs under numerical consistency. It specializes a group of guidances to synchronize distinct dimensions, and captures intrinsic patterns via transferable anchors.  The collective graph structure is utilized to quantify similarity, eliminating the data-recalculation requirement and fusion distortion. By formulating topology transformation on binary space, it reorders anchor and graph structure without altering value consistency to reach appropriate matching. Subsequently, a four-step alternating updating scheme is designed to minimize the associated loss function.

**Questions:**

The questions,

- The specialized guidance mechanism serves to mitigate dimensional discrepancies for perspective- invariant feature extraction. Through the tight integration with the anchor framework, it helps anchor representation transfer across modalities.  Can this generalize to raw data? If applicable, would such implementation yield measurable gains? If not, what are the limitations?

- The built indicator matrices play a role in assisting establishing similarity and transferring similarity from previous views to new views. It seems that they have a complementary relation. Is it possible to build one from the other?  This helps achieve additional space savings. Besides, over time, the indicator matrix for similarity transferring is close to the full size. Are there additional means available to decrease it?

- Along with the ongoing propagation of view information, there will lead to continuous accumulation of historical views.  As a result, compared to the new views, the relative proportion of historical views will continue to increase. However, in lifelong learning scenarios constrained by both privacy regulations and memory limitations, these historical views become inaccessible over time. This raises a critical question: How can we calibrate the contributions between incoming new views and unavailable historical views in the learning process?

**Ethical Concerns:**

["NO or VERY MINOR ethics concerns only"]

**Final Justification:**

The authors have well solved my concerns and I decide to raise the rating.

**Limitations:**

Extra efforts are required for hyper-parameters. The spectrum partitioning may degrade the data diversity.

**Quality:**

3

**Strengths And Weaknesses:**

The strengths,
- The motivation and the overall framework are clearly explained.

- The designed linear complexity facilitates the model’s large-scale deployment capacity.

- The algorithm performance is analysed through multiple perspectives.

The weaknesses,

- The topology transformation may complicate the model, as it requires that each column is composed of one-hot encoder and meanwhile the rows also need to satisfy sum constraints. Moreover, the discrete optimization is usually more challenging to solve compared with continuous counterpart.

- The need for parameter searching may weaken practical performance, as the model requires manual adjustment of two hyper-parameters. This may result in suboptimal performance when processing data with poorly-defined clusters.

- In addition to the memory graph, the model involves the memory anchor, which may increase the space cost, especially on resource-limited platforms. Moreover, in propagation, the anchor quality may vary with the arrival of incoming data.  The degenerated anchors will restrict the representation expression ability of the proposed model.

---

> ### Author Rebuttal · Authors · 2025-07-31
>
> **Q1:**  The topology transformation and discrete optimization may complicate the model.
>
> **A1:**  Thanks! The topology transformation aims at reorganizing anchor order to alleviate the mismatching dilemma while the discrete optimization is to preserve the value consistency. They collaboratively support the anchor knowledge transfer from historical views to current view.  The topology transformation and discrete optimization are equivalently converted to a linear programming, and can be easily solved by off-the-shelf software. Besides, the theoretical complexity demonstrates that they are with a linear overhead and do not bring about severe computing and storing workload.
>
>
> **Q2:**  The need for parameter searching may weaken practical performance.
>
> **A2:** The hyper-parameters aim at regularizing anchor residual and graph residual to formulate desirable similarity. Although it needs parameter searching, combined with sensitivity figures, we have that the model performance is relatively stable under diverse parameter values on multiple data scenes. Therefore, we can deploy it on different scenes with the given guidelines. In the future, we will strive to devise a parameter-free version to further enhance its practicality.
>
>
> **Q3:** The memory anchor may increase the space cost.
>
> **A3:** The memory anchor matrix plays a role in propagating basis embedding from historical views to incoming view. Although it is with memory function, kindly than it has the size of $m \times m$. In general, $m$ is greatly smaller than $n$. Thus, the space cost caused by the memory anchor matrix is almost negligible compared to the overall cost.
>
>
> **Q4:** Can the specialized guidance generalize to raw data?
>
> **A4:** Yes. The specialized guidance is associated with anchor matrix (AAM) in the proposed model, and it is mainly to alleviate dimension disparity to build transferable space. When generalizing to raw data (GRD), although this is also able to construct transferable space via dimension reduction, it generally degenerates dimension diversity and brings about information loss of view data, accordingly weakening the overall clustering performance. To validate this, we organize relevant comparison experiments, as shown in the following table. One can see that GRD is suboptimal compared to AAM.
>
>
>
> |  | PROKARYO |  |  |  |  |  |  |  |  |
> |:---:|:---:|:---:|:---:|:---:|:---:|:---:|:---:|:---:|:---:|
> |  | 0.1 |  |  | 0.4 |  |  | 0.7 |  |  |
> |  | ACC | PUR | FSC | ACC | PUR | FSC | ACC | PUR | FSC |
> | GRD | 62.12 | 74.32 | 50.32 | 52.17 | 58.72 | 51.32 | 43.21 | 58.32 | 42.83 |
> | AAM | **74.95** | **82.21** | **64.79** | **72.78** | **79.13** | **62.82** | **52.09** | **74.59** | **49.59** |
> |  | PROTEINF |  |  |  |  |  |  |  |  |
> | GRD | 33.42 | 39.52 | 20.56 | 30.82 | 35.83 | 16.21 | 26.52 | 31.47 | 11.84 |
> | AAM | **36.26** | **41.70** | **21.81** | **33.57** | **37.89** | **18.41** | **30.07** | **35.07** | **14.41** |
> |  | WIKIFEA |  |  |  |  |  |  |  |  |
> | GRD | 51.35 | 56.73 | 44.57 | 48.65 | 53.11 | 38.23 | 39.78 | 44.67 | 25.89 |
> | AAM | **54.22** | **59.74** | **47.23** | **50.17** | **54.50** | **39.21** | **41.70** | **46.41** | **27.03** |
> |  | NUSWIENE |  |  |  |  |  |  |  |  |
> | GRD | 10.42 | 31.67 | 7.34 | 11.56 | 31.21 | 8.23 | 13.11 | 31.23 | 10.21 |
> | AAM | **10.99** | **33.57** | **7.59** | **12.48** | **33.52** | **9.87** | **15.41** | **33.15** | **15.10** |
> |  | CALTEALL |  |  |  |  |  |  |  |  |
> | GRD | 22.35 | 33.27 | 15.74 | 21.42 | 32.26 | 12.72 | 18.53 | 28.12 | 8.52 |
> | AAM | **25.93** | **35.75** | **17.28** | **23.24** | **33.51** | **14.27** | **20.27** | **29.34** | **9.57** |
> |  | YOUTUTEN |  |  |  |  |  |  |  |  |
> | GRD | 71.34 | 76.83 | 63.62 | 71.87 | 76.86 | 64.53 | 70.26 | 75.43 | 63.67 |
> | AAM | **73.23** | **78.94** | **67.54** | **74.14** | **79.56** | **67.43** | **72.21** | **78.51** | **64.77** |
> |  | YOUTUTWE |  |  |  |  |  |  |  |  |
> | GRD | 69.83 | 76.12 | 58.73 | 69.16 | 75.72 | 58.92 | 66.21 | 74.52 | 58.43 |
> | AAM | **71.23** | **76.86** | **64.74** | **73.86** | **79.51** | **62.06** | **68.62** | **76.91** | **62.90** |
> |  | FAMNISIX |  |  |  |  |  |  |  |  |
> | GRD | 54.24 | 54.76 | 44.52 | 52.31 | 53.52 | 42.73 | 51.32 | 53.32 | 43.78 |
> | AAM | **57.97** | **58.35** | **47.84** | **56.34** | **57.27** | **46.89** | **54.52** | **58.43** | **45.93** |
>
>
> **Q5:**  Is it possible to build one indicator matrix from the other?  Are there means available to decrease its overhead?
>
> **A5:**  Yes. The construction of indicator matrices is based on the union set of observed sample indexes. Therefore, the indicator matrix about current view can be built through that about historical views, vice versa. Besides, although the indicator matrix about historical views is close to full size over time, kindly note that we do not explicitly build and calculate these indicator matrices, and only utilize them for illustration. In practical, we select certain columns from the shared graph via corresponding indexes to establish similarity for current view and do graph propagation for historical views. All these operations are proven to only require linear complexity.
>
>
> **Q6:** How to calibrate the contributions between incoming new views and unavailable historical views?
>
> **A6:** Due to privacy protection or memory limitation, the historical views become inaccessible over time. Consequently, it can not directly weight historical views and incoming new views. To calibrate the contributions, note that during propagation, it inherently involves anchor transfer and graph transfer from historical views to new views. These components hierarchically characterize view data representation. Therefore,  instead of view data, we can calibrate anchor and graph respectively.  Specially, we can adjust anchor importance  in $\mathcal{L}_2$ and graph importance in $\mathcal{L}_3$ to balance view contributions. This dual-calibration mechanism ensures equitable weight allocation across views while preserving structural hierarchies.

---

### Official Review · Reviewer_TZU6 · 2025-07-01

**Clarity:** 2
**Significance:** 3
**Originality:** 3
**Rating:** 4
**Confidence:** 4

**Summary:**

This paper tackles multi-view clustering by proposing a unified framework that simultaneously learns basis embeddings and value-preserving mappings. Distinct feature spaces are first harmonised through dedicated projectors; the model then constructs shared anchors to capture intrinsic patterns and generates bipartite graphs to quantify cross-view similarity, thereby avoiding repeated view-by-view recomputation and fusion distortion. To remain compatible with incoming views, the authors introduce a transformation based on one-hot–encoded row and column attributes of historical views. Anchors and graphs are jointly reordered topologically so that their structures align while their values remain unchanged, and all components are optimised together with overall linear time complexity.

**Questions:**

- Could the authors elaborate on the mathematical or clustering rationale behind enforcing exactly one ‘1’ in each row and column of $\widehat{\mathbf{S}}_t$? What would be the implications—on anchor assignment, similarity computation, and convergence behavior—if this constraint were relaxed to allow multiple or no selections per row?

- What are the key advantages or characteristics of the proposed twin-memory architecture compared to conventional per-view designs, particularly in terms of computational workload, feature extraction, and view-wise information propagation?

- The anchor-alignment transformation can vary from an identity mapping to a complex permutation. Under which data distributions or hyperparameter settings does this transformation collapse to the identity? How does this affect clustering quality and training stability?

- If the guidance mechanism is allowed to relax the one-hot column constraint, does performance on clustering metrics (e.g., NMI, ARI) degrade significantly? Please discuss any observed failure modes such as mode collapse or anchor duplication, and consider including sensitivity curves to illustrate the impact.

- The rationale behind hyperparameter choices is not clearly explained. Since different values can substantially affect performance, a discussion of how these settings were determined would help clarify the method’s underlying principles.

- What is the motivation for introducing matrix vectorization at the specified point in the algorithm? Are there any computational trade-offs or penalties associated with this choice?

- The graph transfer process relies on shared sample observations across views. How does the method behave when a new incoming view has no intersection with historical views? Is anchor transmission or graph transfer still valid in such cases?

**Ethical Concerns:**

["NO or VERY MINOR ethics concerns only"]

**Final Justification:**

Thank you for the response and additional results, which address most of my concerns. After considering the other reviewers’ comments, I am leaning positive to the submission

**Limitations:**

Authors have mentioned the limitations in the work. Parameter-free learning and direct cluster label learning could further boost the practicality and performance.

**Quality:**

3

**Strengths And Weaknesses:**

**Strengths**

+The bit-swapping strategy and twin-memory paradigm address common pitfalls of view misalignment and information loss.

+Related work is thoroughly covered, and the optimisation procedure is explained in a step-by-step, easy-to-follow manner.

**Weaknesses**

- As the overall loss function consists of three components, an explicit summary of the contribution of each item is beneficial for highlighting the entire method’s design.

- Giving the inspiration about the tailored indicator matrices is encouraging. Since the main purpose of these matrices is to pick out certain graph columns to do similarity comparison, the behind inspiration will facilitate fundamental understanding of structural similarity.

---

> ### Author Rebuttal · Authors · 2025-07-31
>
> **Q1:** A summary on the contribution of each item.
>
> **A1:** Thanks! $\mathcal{L}_1$ aims at formulating similarity and extracting basis patterns via reconstruction error minimization. $\mathcal{L}_2$ renders anchors via current view and historical views while preserving value consistency to reach matching configuration. $\mathcal{L}_3$ guarantees  graph scale to be compatible and meanwhile reorganizes graph topology. $\mathcal{L}_2$ and $\mathcal{L}_3$ collaboratively transfer knowledge at similarity level and embedding level. We will add this summary in the next version.
>
>
>
> **Q2:**  The inspiration about tailored indicator matrices.
>
> **A2:** Due to the incompleteness, the graph size is diverse and consequently it is impossible to construct the graph shared for newly-arrived view and previous views. Besides, it also can not do knowledge transfer on graph. To overcome these difficulties, we utilize $\mathbf{H} _t$ to pick out some columns of the shared graph  for current view to build similarity, and utilize $\widetilde{\mathbf{E}} _{t-1}$ to form a smaller graph to compare with previous views. Kindly note that the size of shared graph is dynamically changing as view data is collected.
>
>
> **Q3:** The underlying principle for $\widehat{\mathbf{S}}_t$.
>
> **A3:** This essentially guarantees that anchors are merely rearranged without altering their values. It mainly leverages one-hot characteristics to reorganize anchors. If violated, the learned anchors will be disturbed, potentially destabilizing the entire graph topology.
>
>
>
> **Q4:** The advantages for twin-memory paradigm.
>
> **A4:** Rather than extracting features by view, it builds only one graph across views, and so the computational workload is smaller than current per-view architectures. Besides, it effectively gathers view information during forming features instead of via fusion. Accordingly, the distortion induced by fusion is effectively avoided.
>
>
>
> **Q5:** Could the transformation be a unit matrix?
>
> **A5:** Yes. When the anchors learned on current view is the same order as those on previous views, topological rendering of anchors becomes unnecessary during transformation. Accordingly, it degenerates into a unit matrix.
>
>
>
> **Q6:** Performance under guidance without column constraints.
>
> **A6:** These constraints aims at discriminating projectors. Without them (WO), the dimension space could be redundant and accordingly the model performance will degrade. The following table gives the comparison results.
>
>
> |   | PROKARYO |  |  |  |  |  |  |  |  |
> |:---:|:---:|:---:|:---:|:---:|:---:|:---:|:---:|:---:|:---:|
> |  | 0.1 |  |  | 0.4 |  |  | 0.7 |  |  |
> |  | ACC | PUR | FSC | ACC | PUR | FSC | ACC | PUR | FSC |
> | WO | 68.36 | 77.24 | 60.62 | 67.47 | 77.02 | 58.56 | 48.32 | 66.32 | 41.21 |
> | Ours | **74.95** | **82.21** | **64.79** | **72.78** | **79.13** | **62.82** | **52.09** | **74.59** | **49.59** |
> |  | PROTEINF |  |  |  |  |  |  |  |  |
> | WO | 32.42 | 38.24 | 19.32 | 30.36 | 35.23 | 16.12 | 26.32 | 32.27 | 13.34 |
> | Ours | **36.26** | **41.70** | **21.81** | **33.57** | **37.89** | **18.41** | **30.07** | **35.07** | **14.41** |
> |  | WIKIFEA |  |  |  |  |  |  |  |  |
> | WO | 52.12 | 54.37 | 45.35 | 48.53 | 52.23 | 37.34 | 37.42 | 43.42 | 24.36 |
> | Ours | **54.22** | **59.74** | **47.23** | **50.17** | **54.50** | **39.21** | **41.70** | **46.41** | **27.03** |
> |  | NUSWIENE |  |  |  |  |  |  |  |  |
> | WO | 9.83 | 32.32 | 6.53 | 11.23 | 31.47 | 9.11 | 13.26 | 30.12 | 14.21 |
> | Ours | **10.99** | **33.57** | **7.59** | **12.48** | **33.52** | **9.87** | **15.41** | **33.15** | **15.10** |
> |  | CALTEALL |  |  |  |  |  |  |  |  |
> | WO | 23.28 | 33.43 | 16.27 | 21.27 | 31.56 | 12.89 | 19.84 | 28.83 | 8.47 |
> | Ours | **25.93** | **35.75** | **17.28** | **23.24** | **33.51** | **14.27** | **20.27** | **29.34** | **9.57** |
>
>
>
> **Q7:** The reasons for hyper-parameter setting.
>
> **A7:** $\lambda$ plays a role in regulating anchor residual while $\beta$ regulates graph residual. Because of the consensus structure, these residuals are generally in the same low magnitude. Besides, in experiments, we found that the reconstruction error is usually slightly larger than these residuals. Thus, we search $\lambda$ and $\beta$ in a slightly larger same scale range.
>
>
>
> **Q8:** The rationale for matrix vectorization.
>
> **A8:** It primarily transforms the intractable trace problem into an element-wise maximization linear programming problem, thereby enabling solutions via existing packages.
>
>
>
> **Q9:** Is the transfer still valid without view intersection?
>
> **A9:** Yes. The devised transfer mechanism is only related to the union between indexes. Even without view intersection, it still operates properly as long as  the union is not empty.

---

### Official Review · Reviewer_hkM5 · 2025-07-01

**Clarity:** 3
**Significance:** 3
**Originality:** 3
**Rating:** 5
**Confidence:** 4

**Summary:**

This paper designs an algorithm named BSTM to overcome two limitations in multi-view clustering:  the ignoring of basis embedding and the value-distorting mapping. Unlike traditional feature library mechanisms, it harmonizes disparate dimensions through specialized view-specific projector, and incorporates potential basis patterns during knowledge transfer. This avoids view data recalculating and alleviates fusion-induced distortion. The bit-swapping strategy presented re-render anchor topology and affinity structure by virtue of the one-hot encoded characteristics while preserving numerical consistency. All components are co-optimized in an end-to-end manner, and the updating scheme with linear complexity guarantees the overall efficiency.

**Questions:**

1. Due to the presence of incompleteness, the graph scale is diverse for different views. How to ensure that it is compatible for history data and newly arrived views?
2. In the process of updating the bipartite graph variable, as illustrated, it will induce a third-order overhead to the sample number. So, could you please provide more detailed procedures about how it reduces to linear?
3. It claims that the coefficient in (13) consisting of $\mathbf{H}_t$ in (3) and $\widetilde{\mathbf{E}} _{t-1}$  is non-zero. How to guarantee this point? or How to derive this conclusion?
4. The proposed twin-memory paradigm transfers knowledge collectively, so, rather than the employed learning strategy for anchors, how does the current model perform under heuristic search?

**Ethical Concerns:**

["NO or VERY MINOR ethics concerns only"]

**Final Justification:**

Thank you for the detailed response from the author, which has answered most of my questions. I have decided to maintain my rating.

**Limitations:**

As stated, the designed model involves two hyper-parameters, which requires extra efforts to do fine-tuning. Parameter-free version is the future goal.

**Paper Formatting Concerns:**

The paper formatting is right.

**Quality:**

4

**Strengths And Weaknesses:**

**Strengths**
1. The structure is well-organized and presented logically.
2. Experiments conducted are extensive, and validate the method from multiple aspects.
3. The idea of twin-memory architecture is interesting, and its collaborative knowledge propagating facilitates the discovery of patterns in newly arrived incomplete views.

**Weaknesses**
1.  Associating IN-1 and IN-A with the matrix $\mathbf{H}_t$ in (3) and $\widetilde{\mathbf{E}} _{t-1}$ in (5) respectively helps grasp the contributions  of customized indicators better.
2.  Although harmonizing disparate dimensions, the projectors could bring about extra computational overhead and accordingly reduce efficiency.
3. The illustration about the topology rendering that consists of column sum and row sum is slightly concise, and more explanations could better enhance its significance.

---

> ### Author Rebuttal · Authors · 2025-07-31
>
> **Q1:**  Associating IN-1 and IN-A with $\mathbf{H}_t$ in (3) and $\widetilde{\mathbf{E}} _{t-1}$ in (5).
>
> **A1:** Thanks! We will associate these in the next version.
>
>
> **Q2:** The projectors could bring about extra overhead.
>
> **A2:** According to their size $d_t \times m$, we have that storing them will take $\mathcal{O}(dm)$ memory overhead. Besides, combined with Remark 1, we have that updating them will take $\mathcal{O}(n)$ computing overhead. Since $d$ is a constant and $m$ is largely smaller than $n$, we can derive that these projectors do not impact the overall complexity.
>
>
> **Q3:**  More illustration about topology rendering.
>
> **A3:** The topology rendering mechanism requires that in each column and each row, there is only one element equal to 1, while all other elements are zero. These features guarantee that it merely alters anchor order while anchor value remains unchanged. This makes learned anchors well maintain the characteristics of original data.
>
>
> **Q4:**   How to guarantee compatible graph scale?
>
> **A4:**  To derive compatible graph, we construct indicator matrices to pick out certain columns of the common graph for current view to do similarity calculation. Afterwards, to make this graph transferable, we utilize the indicator matrix to construct corresponding small similarity and then compare it with that constructed on all previous views. Kindly note that we do not explicitly establish these tailored indicator matrices, and use them solely for illustrative purposes. Remarkably, we adopt the element-wise value extraction operation via corresponding index to do similarity construction and comparison.
>
>
> **Q5:** How to reduce the cost of bipartite graph to linear?
>
>
> **A5:** Along with the view collection, the size of $\widetilde{\mathbf{E}} _{t-1}$ will be close to $n \times n$. Accordingly, computing $\widetilde{\mathbf{E}} _{t-1} \widetilde{\mathbf{E}} _{t-1}^{\top}$ will require a third-order cost. To reduce it to linear, note that $\widetilde{\mathbf{E}} _{t-1}$ consists of binary elements, and $\widetilde{\mathbf{E}} _{t-1} \widetilde{\mathbf{E}} _{t-1}^{\top}$ is a diagonal matrix. The diagonal elements are each row sum of $\widetilde{\mathbf{E}} _{t-1}$. Thus, instead of explicit calculation, we can directly construct the diagonal elements, and then multiply it by the shared graph with the size of $m \times n$. Consequently, the computing cost is linear.
>
>
>
> **Q6:** How to guarantee the coefficient in (13) non-zero?
>
> **A6:** $\mathbf{H}_t$ aims at measuring whether the samples are available on current view, while  $\widetilde{\mathbf{E}} _{t-1}$ measures whether the samples are on historical views. Note that the current sample either appears in the current view, the historical view, or both. Therefore, we have that each row sum of $\mathbf{H} _t$ + $\widetilde{\mathbf{E}} _{t-1}$ must be non-zero.  Accordingly, we can derive that this coefficient is non-zero.
>
> **Q7:** The performance under heuristic search.
>
> **A7:** The following table presents the performance comparison where HS and LS denote the results based on heuristic search and learning strategy respectively. As seen, HS usually is sub-optimal, possibly because the heuristic search is unable to effectively characterize original data due to the separation from graph generation.
>
>
> |  | PROKARYO |  |  |  |  |  |  |  |  |
> |:---:|:---:|:---:|:---:|:---:|:---:|:---:|:---:|:---:|:---:|
> |  | 0.1 |  |  | 0.4 |  |  | 0.7 |  |  |
> |  | ACC | PUR | FSC | ACC | PUR | FSC | ACC | PUR | FSC |
> | HS | 53.54 | 72.23 | 46.74 | 44.10 | 56.81 | 43.37 | 42.86 | 56.13 | 41.41 |
> | LS | **74.95** | **82.21** | **64.79** | **72.78** | **79.13** | **62.82** | **52.09** | **74.59** | **49.59** |
> |  | PROTEINF |  |  |  |  |  |  |  |  |
> | HS | 35.41 | 40.95 | 21.35 | 31.64 | 36.44 | 17.33 | 28.42 | 33.63 | 12.77 |
> | LS | **36.26** | **41.70** | **21.81** | **33.57** | **37.89** | **18.41** | **30.07** | **35.07** | **14.41** |
> |  | WIKIFEA |  |  |  |  |  |  |  |  |
> | HS | 53.98 | 58.06 | 46.45 | 49.21 | 54.22 | 39.13 | 41.66 | 45.11 | 26.02 |
> | LS | **54.22** | **59.74** | **47.23** | **50.17** | **54.50** | **39.21** | **41.70** | **46.41** | **27.03** |
> |  | NUSWIENE |  |  |  |  |  |  |  |  |
> | HS | 10.85 | 33.26 | 7.41 | 12.42 | 30.45 | 9.79 | 13.02 | 31.07 | 10.98 |
> | LS | **10.99** | **33.57** | **7.59** | **12.48** | **33.52** | **9.87** | **15.41** | **33.15** | **15.10** |
> |  | CALTEALL |  |  |  |  |  |  |  |  |
> | HS | 21.37 | **35.85** | 15.16 | 18.33 | 31.11 | 12.67 | 17.60 | 29.24 | **9.64** |
> | LS | **25.93** | 35.75 | **17.28** | **23.24** | **33.51** | **14.27** | **20.27** | **29.34** | 9.57 |
> |  | YOUTUTEN |  |  |  |  |  |  |  |  |
> | HS | 72.75 | 78.62 | 62.97 | 73.80 | 79.37 | 64.34 | **72.23** | 76.36 | 64.51 |
> | LS | **73.23** | **78.94** | **67.54** | **74.14** | **79.56** | **67.43** | 72.21 | **78.51** | **64.77** |
> |  | YOUTUTWE |  |  |  |  |  |  |  |  |
> | HS | 70.06 | 75.51 | 55.73 | 68.28 | 76.79 | 56.56 | 68.53 | 76.86 | 58.27 |
> | LS | **71.23** | **76.86** | **64.74** | **73.86** | **79.51** | **62.06** | **68.62** | **76.91** | **62.90** |
> |  | FAMNISIX |  |  |  |  |  |  |  |  |
> | HS | 52.16 | 55.72 | 41.26 | 50.14 | 53.70 | 39.34 | 43.00 | 43.11 | 25.66 |
> | LS | **57.97** | **58.35** | **47.84** | **56.34** | **57.27** | **46.89** | **54.52** | **58.43** | **45.93** |

---

### Decision · Program_Chairs · 2025-09-17

**Decision:**

Accept (poster)

**Comment:**

This paper proposes a novel Bit-swapping Oriented Twin-memory (BSTM) framework for lifelong incomplete multi-view clustering, featuring a twin-memory architecture for cross-view alignment, a bit-swapping mechanism to harmonize current and historical views, and a linear-complexity update strategy for scalability. Reviewers praised its originality, clear methodology, and comprehensive experimental validation across benchmarks, noting its effectiveness in addressing view misalignment and preserving knowledge. Concerns centered on the justification of one-hot constraints, reliance on hyperparameters, and the need for clearer explanations of certain components and parameters. The rebuttal addressed most issues with additional experiments, complexity analysis, and design rationale, leading two reviewers to recommend acceptance and two to lean borderline accept due to clarity. Overall, the contribution is technically sound and well-supported, warranting acceptance.